# NEURAL+SYMBOLIC ACTOR-CRITIC FRAMEWORK FOR INTERPRETABLE REINFORCEMENT LEARNING

**Yue Yang**[1,2*†]**, Fan Yang**[1*]**, Yu Bai**[1]**, Hao Wang**[1†]
[1] Faculty of IT, Monash University, Australia    [2] Maincode, Australia
`{yue.yang1,fan.yang1,yu.bai3,hao.wang2}@monash.edu`

## ABSTRACT

The integration of neural networks into actor-critic frameworks has been pivotal in advancing the field of reinforcement learning, enabling agents to perform complex tasks with greater efficiency and adaptability. However, neural network-based actor-critic models remain opaque "black boxes," concealing their decision-making processes and hindering their use in critical applications where transparent and explainable reasoning is essential. This work introduces an innovative adaptation of the actor-critic framework that unites neural networks with rule ensembles to tackle key challenges in reinforcement learning. We harness the computational power, scalability, and adaptability of neural networks to model the critic, while integrating a rule ensemble system for the actor, ensuring transparency and interpretability for decision-making. Our study establishes a theoretical foundation for integrating rule ensembles into the Advantage Actor-Critic (A2C) framework. Experimental results from seven classic and complex environments demonstrate that our proposed method matches or exceeds the performance of representative RL models, including symbolic methods, while offering self-interpretability and transparency.

## 1 INTRODUCTION

Actor-critic reinforcement learning (RL) methods, such as Advantage Actor-Critic (A2C) and Proximal Policy Optimization (PPO), have delivered exceptional performance across diverse RL tasks (Mnih et al., 2016; Schulman et al., 2017), largely due to the use of neural networks to effectively manage complex, high-dimensional state-action spaces. Yet, as actor-critic RL models expand to broader real-world domains, such as healthcare, finance, and law, interpretability becomes essential for ensuring trust, transparency, and regulatory compliance in decision-making. By revealing how actions are chosen or why specific decisions are made, interpretability supports debugging, model improvement, and the ability to explain automated decisions in line with legal and ethical standards.

Despite the effectiveness of neural network-based actor-critic models, their opaque nature and lack of transparency can limit their use in sensitive or critical applications where comprehending the rationale behind each decision is crucial (Benítez et al., 1997; Castelvecchi, 2016). Symbolic models, e.g., expert systems, decision trees, and rule-based systems, rely on clear, well-defined rules and logic to make decisions, allowing for easier tracing of how conclusions are reached (Ernst et al., 2005; Gupta et al., 2017; Tao et al., 2018; Bastani et al., 2018; Illanes et al., 2020). These methods use symbols (which can represent objects, properties, or relationships) and logic to process and reason about data, making the decision-making process transparent and understandable.

However, substituting neural network-based models with symbolic models in reinforcement learning tasks can be challenging. (i) While symbolic methods are highly valued for their interpretability and transparency, current implementations in reinforcement learning typically rely on predefined knowledge or pre-trained models (Garcez et al., 2018; Lyu et al., 2019). (ii) These symbolic methods can face challenges in managing the complexity and ambiguity that non-symbolic approaches, such as neural networks, are often better equipped to handle (Illanes et al., 2020; Landajuela et al., 2021).

---

[*]Equal contribution.
[†]Corresponding authors: Yue Yang and Hao Wang.

To tackle the above challenges, we propose a new "**N**eural plus **S**ymbolic" framework for **A**ctor-**C**ritic models, namely **NSAC**, as shown in Figure 1. NSAC integrates the strengths of both neural networks and symbolic rule-based systems. Specifically, the neural critic leverages powerful function approximation capabilities to estimate the value of states and state-action pairs effectively. Meanwhile,

the actor component is designed using a rule-based approach, providing intrinsic interpretability by following explicit, predefined rules that govern its decision-making process. This framework allows for an analysis of how particular features or rules influence decisions, offering a level of explainability absent in black-box models. Our theoretical analysis proves that NSAC converges to a local optimum. Through empirical experiments, our proposed NSAC demonstrates performance levels that match those of widely-used reinforcement learning algorithms, such as DQN (Mnih et al., 2013), PPO (Schulman et al., 2017), Rainbow (Hessel et al., 2018), SDSAC (Kong

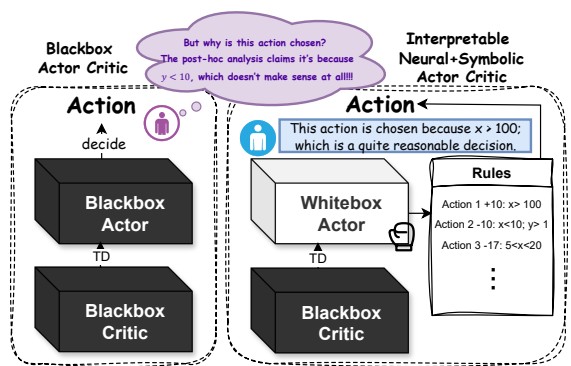

Figure 1: Comparative illustration of Neural+Symbolic Actor-Critic (NSAC) and classic Actor-Critic methods.

et al., 2021), SACBBF (Zhang et al., 2024), and A2C (Mnih et al., 2016). By integrating the adaptability and generalization capabilities of neural networks with the transparency of rule-based decision-making, our model provides a balanced solution that retains high performance while enabling interpretable and trustworthy decision-making processes. It is worth noting that, our work belongs to the broader neural-symbolic family, but differs from compile-logic approaches. NSAC uniquely couples a data-driven neural critic with a rule-based symbolic actor, enabling both statistical learning and explicit reasoning within an actor–critic loop. This work makes three main contributions:

- We propose a novel actor–critic architecture that combines a neural critic for accurate value estimation with a symbolic, rule-based actor that delivers intrinsic interpretability.
- We provide a formal proof of convergence and demonstrate that our rule-based policy satisfies the established interpretability criteria of simulatability, modularity, and low complexity (Murdoch et al., 2019).
- We empirically demonstrate that NSAC matches state-of-the-art RL methods on classic control and energy-system tasks, while uniquely preserving transparency and trustworthiness.

## 2 RELATED WORK

Interpretability has become increasingly important in RL and motivated symbolic approaches that provide transparent reasoning while retaining RL's adaptive capabilities. Existing approaches can be categorized based on their interpretability characteristics, knowledge requirements, and learning paradigms, each with distinct limitations. Tree-based policies represent a widely studied approach to interpretable RL. Native tree methods (Ding et al., 2020; Ernst et al., 2005; Gupta et al., 2017; Roth et al., 2019; Tao et al., 2018) learn decision trees directly from environmental interactions, often achieving strong performance on small to medium-scale problems but facing scalability challenges where large trees become difficult to interpret. Recent gradient boosting approaches (Fuhrer et al., 2024) and programmatic tree policies (Kohler et al., 2024) extend this line by offering editable representations and improved handling of structured features, though interpretability-performance trade-offs remain. Distillation-based tree methods (Bastani et al., 2018; Coppens et al., 2019; Gupta et al., 2015; Liu et al., 2018) extract tree policies from pre-trained neural networks, but face a size versus fidelity trade-off. Their differentiable variants (Silva et al., 2020) and specialized architectures (Ding et al., 2020; Liu et al., 2018) attempt to address these limitations but may produce mathematical notations and complex tree structures that remain difficult for humans to interpret.

Symbolic RL with predefined knowledge leverages domain expertise to guide policy learning. These methods (Garcez et al., 2018; Illanes et al., 2020; Lyu et al., 2019) can maintain interpretability through symbolic rules, modules, or plans, and may achieve strong performance when domain priors

are well-specified. However, they require substantial predefined knowledge, making them sensitive to domain assumptions and limiting their applicability to new domains where such expertise may not be available. To remove this dependency, symbolic policies learn from neural teachers or direct search to discover symbolic representations, such as neural-guided symbolic policy discovery (Landajuela et al., 2021), genetic programming for symbolic expressions (Hein et al., 2018), and program synthesis approaches (Wu et al., 2020; Qiu & Zhu, 2022). While these methods avoid predefined knowledge, they often yield complex mathematical expressions involving trigonometric functions, logarithms, or program syntax that are challenging for humans to simulate and interpret (Murdoch et al., 2019). Recent approaches (Delfosse et al., 2023b; Luo et al., 2024; Shindo et al., 2024) improve the interpretability through nerve guide, end-to-end or mixed framework, although they still produce complex models, or external tools are needed to generate rules. Post-hoc approaches additionally may suffer from approximation errors inherited from neural teachers. The importance of interpretable RL is underscored by recent findings on fundamental challenges in agent behavior, including goal misgeneralization (Di Langosco et al., 2022) and systematic failures on task variations (Delfosse et al., 2025), which have motivated approaches such as concept bottlenecks (Delfosse et al., 2024) for detecting misalignments and object-centric environments (Delfosse et al., 2023a) for visual domains, highlighting the need for transparent and verifiable RL systems.

As discussed, most previous methods either require a certain amount of predefined knowledge, which influences the search space or establishes the programmatic rules, or they are post-hoc solutions that learn from a pre-trained model rather than directly from the environment, potentially distorting the learning objectives, or they generate complex representations or rely on external tools for rule generation rather than learning directly from environmental interactions (See Appendix H). In contrast, our method effectively integrates a neural network-based critic with a symbolic, rule-based, interpretable actor by learning directly through environmental interactions without requiring predefined knowledge or external rule generation tools. Specifically, we address the tree scalability problem by using additive rule ensembles rather than hierarchical structures, where each rule contributes equally to the decision-making process. We employ Orthogonal Gradient Boosting (OGB) (Yang et al., 2024) to discover rule conditions automatically, requiring only basic feature comparisons rather than domain-specific knowledge. Our rule-based actor is trained directly via policy gradients on environmental rewards, ensuring the symbolic representation accurately reflects true decision-making process rather than approximating a black-box teacher. Our approach achieves interpretability through symbolic rules while learning directly from the environment, addressing both the knowledge requirements and post-hoc limitations of existing methods. Post-hoc explainers are only approximations: they often flag correlates, not causes. SHAP (Lundberg & Lee, 2017), for example, marks a feature as "important" while the model flips under tiny perturbations. Worse, black-box models can even be trained to emit persuasive but misleading rationales (Rudin, 2019). Because these explanations are extrinsic, they guarantee neither consistency, fidelity, nor out-of-distribution robustness.

## 3 PRELIMINARIES

### 3.1 ADVANTAGE ACTOR-CRITIC REINFORCEMENT LEARNING

The A2C algorithm represents a synergy of two foundational approaches in reinforcement learning: the actor-critic method and the advantage function estimation. In the actor-critic framework, the critic in the A2C framework estimates the value function $V(s)$, which represents the expected return from a state $s$. The critic is updated by minimizing its loss function, typically the mean squared error between the estimated value and the computed target value. The target value is derived from the rewards obtained and the discounted values of subsequent states: $L_V(\phi) = \mathbb{E}\left[(R_t + \gamma V(s_{t+1}) - V_\phi(s_t))^2\right]$, where $R_t$ is the reward received after taking action $a_t$ in state $s_t$ at $t$, $\gamma$ is the discount factor, $V_\phi(s_{t+1})$ is the value estimate for the next state at $t+1$, and $\phi$ are the parameters of the critic network. The actor updates the policy by adjusting parameters $\theta$ in the direction suggested by the critic's evaluations. The update is guided by the gradient of the policy's performance, estimated using the advantage function: $\nabla_\theta L(\theta) = \mathbb{E}\left[\nabla_\theta \log \pi_\theta(a_t|s_t)\mathcal{A}_t\right]$, where $\mathcal{A}_t = R_t + \gamma V(s_{t+1}) - V(s_t)$ is the advantage at time $t$, indicating how much better (or worse) the action $a_t$ is than the policy's average at state $s_t$. Through these updates, the actor learns to choose actions yielding higher returns than predicted by the critic, while the critic learns to make more accurate predictions about expected returns, forming a feedback loop that enhances both policy and value estimation in the A2C framework.

## 3.2 ADDITIVE RULE ENSEMBLES

**Additive rule ensembles** are a class of probabilistic models (Friedman & Popescu, 2008) that estimate the expected value of a target variable $Y \in \mathbb{R}$ given an input vector $\mathbf{X} \in \mathbb{R}^d$, which is expressed as $\mathbb{E}[Y \mid \mathbf{X} = \boldsymbol{x}] = \mu(f(\boldsymbol{x}))$, where, $\boldsymbol{x}$ is an instance of $\mathbf{X}$, $\mu : \mathbb{R} \to \mathbb{R}$ is an inverse link function mapping $f(\boldsymbol{x})$ to the target variable $Y$, and $f : \mathbb{R}^d \to \mathbb{R}$ is a linear combination of $k$ Boolean query functions:

$$f(\boldsymbol{x}) = \sum_{i=1}^{k} w_i q_i(\boldsymbol{x}) = \sum_{i=1}^{k} w_i \prod_{j=1}^{c_i} p_{i,j}(\boldsymbol{x}). \tag{1}$$

In Equation 1, $w_i$ $(0 \leq i \leq k)$ is the weight of the $i$-th Boolean query function $q_i :$ $\mathbb{R}^d \to \{0,1\}$, and $q_i$ is a conjunction or product of $c_i$ Boolean propositions, where $p_{i,j} \in \left\{ \mathbb{I}\left( sx^{(j)} \leq sx_l^{(j)} \right) : j \in [d], l \in [n], s \in \{\pm 1\} \right\}$ is a threshold function ($[x] = \{1, \ldots, x\}$ is the index set). Each term $w_i q_i(\boldsymbol{x})$ in Equation 1 can be represented as an 'IF... THEN ...' rule, where the query $q_i$ defines the condition of the rule, and the weight $w_i$ is the rule's consequent.

Boosting is an iterative approach for learning additive models (Schapire, 1990; Friedman, 2001; Dembczyński et al., 2010). Given a training set of size $n$, we have $\{(\boldsymbol{x}_1, y_1), \ldots, (\boldsymbol{x}_n, y_n)\}$, where $\boldsymbol{x}_i$ is the input and $y_i$ is the corresponding target, through approximately minimizing the *empirical risk*

$$\hat{L}_\lambda(f) = \frac{1}{n} \sum_{i=1}^{n} l(f(\boldsymbol{x}_i), y_i) + \frac{\lambda \|\mathbf{w}\|^2}{n}, \tag{2}$$

boosting iteratively adds terms into the model and yields a sequence of models $f^{(1)}, \ldots, f^{(k)}$. The last term of Equation 2 is the regularization term, $\mathbf{w} = (w_1, \ldots, w_k)^T \in \mathbb{R}^k$ is the weight vector, and $\lambda \geq 0$ is the regularization parameter. Note that $l$ is a loss function derived as shifted negative log likelihood, measuring the cost of outputting $f(\boldsymbol{x}_i)$ while the true value is $y_i$.

## 4 METHODOLOGY

### 4.1 FRAMEWORK DESIGN

**Actor implemented with Rule-Based Ensembles:** As shown in Figure 2, we replace the neural network-based action function approximation with an actor composed of additive rule ensembles. Let $\mathcal{F}(s) = \{f_a(s) : \forall a \in A\}$ represent the collection of ensemble models for action $a$ in the action space $A$ at state $s \in S$. The size for $A$ is $m$. The ensemble $f_a(s)$ for an action $a$ predicts the expected advantages directly, simplifying its expression as $\mathcal{A}(s,a) = f_a(s)$. This prediction is based on a linear combination of Boolean query functions, weighted by coefficients specific to each action $a \in A$: $f_a(s) = \sum_{j=1}^{k} w_{a,j} q_{a,j}(s)$, where $w_{a,j}$ are the coefficients for $q_{a,j}$. For each action $a$, there are $k$ Boolean query functions $q_{a,j}, \ (j = 1 \ldots, k)$ specific to the action $a$. The actor's policy, represented as $\pi$, is defined by the probability distribution over action values for a given state, which is derived from $\mathcal{F}$ using the softmax function: $\pi(a|s) = \frac{\exp(f_a(s))}{\sum_{f_a \in \mathcal{F}} \exp(f_a(s))}$. The softmax function ensures that $\pi(a|s)$ is a proper probability distribution over the action space.

**Critic implemented with Neural Network:** In our framework, the critic functions similarly to traditional actor-critic systems, leveraging the computational capabilities of neural networks. It serves as a function approximator, implemented using a neural network that estimates the value of being in a specific state $s$. This neural network, paramount in computing state-value functions, is parameterized by weights $\phi$, and the value function it approximates is denoted by $V_\phi(s)$. This design choice ensures that our critic effectively harnesses the processing power of neural networks to deliver precise value estimations for each state. The pseudocode for the algorithm is shown in Appendix B and the theoretical analysis in Appendix C .

### 4.2 POLICY UPDATE

**Critic Policy Update:** The critic in our framework estimates the value function $V(s)$ using a neural network, representing the expected return from a state $s$. The critic is updated by minimizing the

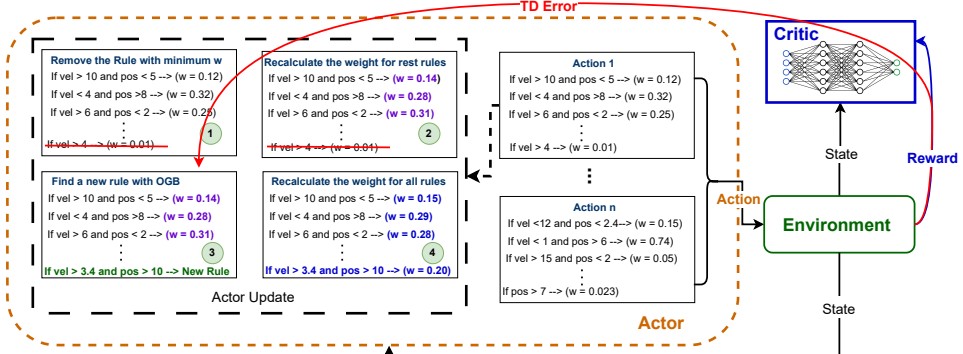

Figure 2: **NSAC learns an interpretable actor by combining a neural critic with a rule-based policy updated via orthogonal gradient boosting.** The diagram illustrates the full NSAC training loop, including critic updates, rule mining, rule replacement, and fully-corrective weight optimization steps for the symbolic actor. Blocks and arrows show how trajectories, advantages, and gradients are propagated through the system to iteratively refine the rule ensemble; additional algorithmic details are provided in Section 4 and Appendix B.

loss function, typically the mean squared error between the estimated value and the target value. The target value is derived from the rewards obtained and the discounted values of subsequent states: $L_V(\phi) = \mathbb{E}\left[(R_t + \gamma V(s_{t+1}) - V_\phi(s_t))^2\right]$, where $R_t$ is the reward received after taking action $a_t$ in state $s_t$, $\gamma$ is the discount factor, $V_\phi(s_{t+1})$ is the value estimate for the next state, and $\phi$ are the parameters of the critic network.

**Actor Update:** The learning process involves iterative updates to the coefficient vector $\mathbf{w}_a$ and the query functions $q_a$ based on observed rewards and state transitions, aiming to refine the ensembles for optimal decision-making. This approach leverages the interpretability of rule-based systems in the actor's policy formulation, enhancing transparency in complex or high-dimensional state spaces. The advantage function, used by the actor for updates, is computed using the critic's value function and the immediate reward: $\mathcal{A}_t(s_t, a_t) = R_t + \gamma V(s_{t+1}) - V(s_t)$. This expression calculates how much better the current action $a_t$ is compared to the average action at state $s_t$ at time $t$, considering the immediate reward and discounted future values. The actor updates the policy by adjusting the parameters $(\mathbf{w}, \mathbf{q})$ in the direction suggested by the critic's evaluations. The update is guided by the gradient of the policy's performance, which is estimated using the advantage function. The loss function derived from the policy gradient is

$$\nabla L_\lambda(\mathbf{w}, \mathbf{q}) = -\mathbb{E}\left[\nabla_{\mathbf{w}, \mathbf{q}} \log \pi(a_t | s_t, \mathbf{w}, \mathbf{q})\mathcal{A}_t\right] + \lambda \nabla_{\mathbf{w}, \mathbf{q}} \|\mathbf{w}\|_2^2. \tag{3}$$

To make the loss function locally convex, we add a regularization term $\lambda\|\mathbf{w}\|_2^2$ to the original loss, where $\lambda$ is the regularization parameter (see Appendix D). The actor is updated by modifying each rule ensemble model $f_a$ in accordance with the calculated policy gradient objective for each action. To obtain the gradient for each action, we use softmax for action selection, taking the natural logarithm of the policy yields:

$$\forall a, \quad \log \pi(a \mid s_t; \mathbf{w}, \mathbf{q}) = \mathbf{w}_a \mathbf{q}_a - \log \sum_{j=1}^m \exp(\mathbf{w}_{a_j} \mathbf{q}_{a_j}). \tag{4}$$

Substituting Equation 4 into Equation 3, we have:

$$\nabla L_\lambda(\mathbf{w}, \mathbf{q}) = -\mathbb{E}\left[\nabla_{\mathbf{w}, \mathbf{q}} \mathcal{A}_t \left(\mathbf{w}_{a_t} \mathbf{q}_{a_t} - \log \sum_{j=1}^m \exp(\mathbf{w}_{a_j} \mathbf{q}_{a_j})\right)\right] + \lambda \nabla_{\mathbf{w}, \mathbf{q}} \|\mathbf{w}\|_2^2. \tag{5}$$

Treating $\mathbf{w}$ and $\mathbf{q}$ as a combined vector $\boldsymbol{\theta} = (\mathbf{w}; \mathbf{q})$, the gradient of $\nabla_{\boldsymbol{\theta}} L(\mathbf{w}, \mathbf{q})$, i.e., $\nabla_{\boldsymbol{\theta}}^2 L(\mathbf{w}, \mathbf{q})$ can be derived as

$$\nabla_{\boldsymbol{\theta}}^2 L_\lambda(\mathbf{w}, \mathbf{q}) = \frac{\partial \nabla l}{\partial \mathbf{w}_a \mathbf{q}_a} = -\mathbb{E} \left[ \nabla_{w_a, q_a} \frac{\mathcal{A}_t \partial}{\partial \mathbf{w}_a \mathbf{q}_a} \left( \mathbf{w}_{a_t} \mathbf{q}_{a_t} - \log \sum_{j=1}^m \exp(\mathbf{w}_{a_j} \mathbf{q}_{a_j}) \right) \right] =$$

$$-\mathbb{E} \left[ \nabla_{w_a, q_a} \mathcal{A}_t \left( \mathbb{I}_{a=a_t} - \frac{\exp(\mathbf{w}_a \mathbf{q}_a)}{\sum_{j=1}^m \exp(\mathbf{w}_{a_j} \mathbf{q}_{a_j})} \right) \right] = -\mathbb{E} \left[ \nabla_{w_a, q_a} \mathcal{A}_t \left( \mathbb{I}_{a=a_t} - \pi(a \mid s_t; \mathbf{w}, \mathbf{q}) \right) \right].$$

Taking into account the two distinct situations $a = a_t$ and $a \neq a_t$, the policy gradient for each rule ensemble is as follows:

1. When the ensemble model $f_a$ corresponds to the action executed in the current step, i.e., $a = a_t$,
$$\nabla_{w_a, q_a}^2 L_\lambda(w_a, q_a) = \mathbb{E} \left[ -\nabla_{w_a, q_a} \mathcal{A}(s_t, a_t)(1 - \pi(a_t|s)) \right]; \tag{6}$$

2. When the ensemble model $f_{a_i}$ does not correspond to the action selected by the actor at that time step, i.e., $a \neq a_t$,
$$\nabla_{w_a, q_a}^2 L_\lambda(w_a, q_a) = \mathbb{E} \left[ \nabla_{w_a, q_a} \mathcal{A}(s_t, a_t) \pi(a_t|s) \right]. \tag{7}$$

This distinction ensures ensemble updates properly align with the actions taken, enabling targeted adjustments that enhance the actor's policy based on observed outcomes and predicted advantages.

**Computational Complexity:** This algorithm has a time complexity of $O(d^2 nk)$ per rule, where $d$ is the observation dimension. It performs at most $d$ iterations to construct a single rule (adding one dimension per iteration), and in each iteration, it evaluates all $d$ dimensions as candidates. The bottleneck lies in computing the orthogonal gradient boosting objective $\text{obj}_{\text{ogb}}(q) = |g_\perp^T q| / (||q_\perp|| + \epsilon)$ for each candidate, which requires $O(nk)$ operations due to the orthogonal projection computation: $g_\perp = g - BB^T g$. This represents a $k$-fold increase in computational cost over standard gradient boosting (which has complexity $O(d^2 n)$). The orthogonalization overhead is essential for the algorithm's ability to select more general rules and achieve better risk-complexity trade-offs in practice. Fully-corrective weight calculation has a complexity of $O(k^2 n)$ from solving a convex optimization problem that re-optimizes all weights after adding a new rule. The $k^2$ scaling comes from computing the Hessian on the $k \times k$ Gram matrix $Q^T Q$. While more expensive than stage-wise approaches, this cost is manageable for small ensemble sizes typically in interpretable rule learning.

## 5 EXPERIMENTS AND RESULTS

In this section, we aim to answer the following questions: (RQ1) How does our algorithm perform relative to both black-box and interpretable baselines? (RQ2) Does our framework support interpretable decision-making in a quantifiable way? (RQ3) What kinds of qualitative insights about the policy can be obtained from inspecting the learned rules?

We evaluated the performance of NSAC in five diverse RL environments: MountainCar-v0, Acrobot-v1, CartPole-v1, Blackjack-v1, and Postman (Andrew & Richard S, 2018; Dietterich, 2000). We also evaluated our NSAC algorithm on a challenging, real-world HVAC control benchmark using Sinergym (see Appendix J for details). This environment models a multi-zone office building with coupled thermal dynamics, where each zone's temperature, humidity, and carbon dioxide concentration evolve according to nonlinear heat-transfer and air-exchange equations. The high dimensionality of the state–action space, the strong inter-zone couplings, and the requirement to satisfy strict comfort bands make it a great testbed for our algorithm.

### 5.1 PERFORMANCE EVALUATION (RQ1)

We conducted a comprehensive performance comparison against established benchmark algorithms, including Q-Tabular (Andrew & Richard S, 2018), DQN (Mnih et al., 2013), PPO (Schulman et al., 2017), and A2C (Mnih et al., 2016), all implemented using Stable-Baselines3 (Raffin et al., 2021). We additionally included Rainbow (Hessel et al., 2018), SDSAC (Kong et al., 2021), and

---

**Algorithm 1** Neural+Symbolic Actor-Critic Algorithm

---

1: Initialize actor with each action rule ensemble function $f_a^{(0)} = 0$; Initialize critic network $V(s, \phi)$ with parameters $\phi$; Initialize environment and observe initial state $s$
2: **repeat**
3:   **for** each step of episode **do**
4:     Choose action $a \sim \pi(a|s, \theta)$
5:     Observe reward $r$ and new state $s'$
6:     Calculate $\delta = r + \gamma V(s', \phi) - V(s, \phi)$
7:     Update critic by minimizing $L(\phi) = \delta^2$
8:     **for** all $a_i$ in action space **do**
9:       **if** $a_i = a$ **then**
10:         Calculate policy gradient $\boldsymbol{g}_t$: $-\nabla_{\mathbf{w}, \mathbf{q}}(1 - \log \pi(a|s, \mathbf{w}^T \mathbf{q}))\delta$ for $f_a$
11:       **else**
12:         Calculate policy gradient $\boldsymbol{g}_t$: $\nabla_{\mathbf{w}, \mathbf{q}} \log \pi(a|s, \mathbf{w}^T \mathbf{q})\delta$ for $f_a$
13:       **end if**
14:       use Rule Replacement Steps and policy gradient to find a new query for $f_a$ (Appendix B)
15:       use OGB to update the weights for $f_a$ (See Appendix B)
16:     **end for**
17:     $s \leftarrow s'$
18:   **end for**
19:   **if** end of episode **then**
20:     Reset environment and observe initial state $s$
21:   **end if**
22: **until** convergence or maximum episodes reached.

---

SACBBF (Zhang et al., 2024) as deep RL baselines. To cover symbolic approaches, we also evaluated three interpretable methods: SYMPOL (Marton et al., 2024), $\pi_{affine}$-D (Qiu & Zhu, 2022), and D-SDT (Silva et al., 2020). For a fair comparison, we use the Gym environments in their original form without any modifications. For each algorithm, we apply the recommended optimal parameters provided by the RLzoo (Raffin & contributors, 2020) with Stable Baselines 3 and report performance averaged over 10 random seeds. As shown in Table 1, NSAC demonstrates strong and reliable performance across a wide spectrum of tasks, outperforming or matching both classic deep RL methods and symbolic approaches. Compared with classical methods such as Q-learning, DQN, A2C, PPO, SAC variants, and Rainbow, NSAC achieves consistently higher returns with lower variance, particularly in settings that demand stability and scalability. Unlike value-based methods that can be brittle in continuous or high-dimensional domains, NSAC maintains robustness while still being competitive in simpler benchmarks. Against symbolic tree-based methods such as SYMPOL, $\pi_{affine}$-D, and D-SDT, NSAC shows even clearer advantages: while symbolic policies can excel on select tasks due to strong inductive biases, they often suffer from collapse or sharp degradation when scaled, whereas NSAC delivers stable performance across all environments. This makes it a practical and effective alternative that closes the gap between the strengths of classic RL and symbolic reasoning while avoiding their respective weaknesses.

**Ablation Study**   In our ablation study, we explored how variations in the rule count and the implementation of a warm start influenced our method. We conducted tests in the CartPole-v1 environment under various rule configurations, specifically using 5, 10, 12, 20, 30, 40, and 50 rules per action. Each configuration was tested both with and without a warm start. *1. Number of Rules:* As depicted in Figure 3, the model configured with 12 rules per action and a warm start yields the highest reward. This outcome suggests that having 12 rules per action provides the model with sufficient flexibility to effectively capture the essential dynamics of the environment without being overly simplistic. Employing fewer than 10 rules, such as 5, may not offer adequate coverage to manage the environment's complexity, resulting in inferior performance. Conversely, a larger number of rules may lead the model to overfit the training data, picking up on noise or irrelevant patterns and diminishing its generalizability to new contexts. The findings from this ablation study highlight a critical balance between the

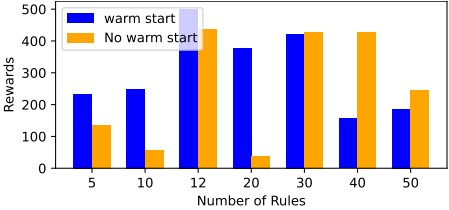

Figure 3: Ablation study on different numbers of rules and warm start.

Table 1: **Our NSAC policy attains rewards that match or exceed deep RL baselines while clearly outperforming symbolic baselines across most environments.** The table reports mean episodic return ($\pm$ standard deviation) for NSAC, classic deep RL methods (e.g., A2C, PPO, Rainbow), and symbolic approaches (SYMPOL, $\pi_{affine}^{D}$, D-SDT) on the Gym and HVAC tasks, showing that NSAC achieves comparable performance to strong neural baselines and markedly higher returns than tree-based policies, especially in more challenging environments, such as HVAC. Best results are highlighted in bold, with all numbers averaged over multiple random seeds; further training and evaluation details are given in Appendix I.

| Environment | Q-table | DQN | A2C | PPO | SDSAC | SACBBF | Rainbow | SYMPOL | $\pi_{affine}$-D | D-SDT | NSAC |
|---|---|---|---|---|---|---|---|---|---|---|---|
| **MCar-v0** | -147.68 $\pm$ 9.52 | *-135.07* $\pm$ 16.42 | -157.51 $\pm$ 37.12 | -150.40 $\pm$ 4.05 | -141.76 $\pm$ 13.91 | -139.23 $\pm$ 17.62 | -137.76 $\pm$ 39.47 | -200 $\pm$ 0 | -200 $\pm$ 0 | -200 $\pm$ 0 | **-132.25** $\pm$ 16.08 |
| **Acrobot-v1** | -201.01 $\pm$ 13.32 | -112.68 $\pm$ 8.93 | -98.931 $\pm$ 29.52 | -82.629 $\pm$ 2.79 | -85.124 $\pm$ 7.82 | -88.708 $\pm$ 7.12 | -89.753 $\pm$ 8.98 | **-80.02** $\pm$ 4.28 | -425.47 $\pm$ 56.81 | -212.31 $\pm$ 4.19 | -87.714 $\pm$ 6.95 |
| **CartPole-v1** | 219.51 $\pm$ 69.2 | 161.00 $\pm$ 55.2 | 453.51 $\pm$ 92.5 | 498.73 $\pm$ 2.4 | 487.61 $\pm$ 2.71 | 481.07 $\pm$ 10.76 | 498.53 $\pm$ 2.12 | **500** $\pm$ 0 | 109 $\pm$ 76.82 | 498.53 $\pm$ 2.12 | **499.14** $\pm$ 1.91 |
| **Blackjack-v1** | **-0.06** $\pm$ 0.02 | **-0.06** $\pm$ 0.02 | -0.07 $\pm$ 0.01 | **-0.06** $\pm$ 0.01 | -0.07 $\pm$ 0.02 | **-0.06** $\pm$ 0.01 | **-0.06** $\pm$ 0.02 | **-0.06** $\pm$ 0.01 | -0.08 $\pm$ 0.01 | -0.08 $\pm$ 0.02 | **-0.06** $\pm$ 0.01 |
| **Postman** | **35.24** $\pm$ 2.06 | 31.91 $\pm$ 5.71 | 24.23 $\pm$ 3.09 | 34.35 $\pm$ 3.82 | 34.91 $\pm$ 2.86 | 31.12 $\pm$ 3.77 | 34.67 $\pm$ 4.68 | 25.34 $\pm$ 3.42 | 15.23 $\pm$ 4.91 | 18.91 $\pm$ 6.28 | 27.14 $\pm$ 3.38 |
| **HVAC-1Zone** | NA | -1445367 $\pm$ 20183 | *-1334562* $\pm$ 148658 | -1865276 $\pm$ 135683 | -1387692 $\pm$ 145619 | -1423573 $\pm$ 34825 | -1465732 $\pm$ 785732 | -1478783 $\pm$ 65293 | -1895286 $\pm$ 42873 | -1862537 $\pm$ 29763 | **-1251321** $\pm$ 85241 |
| **HVAC-5Zone** | NA | -1876253 $\pm$ 198263 | -1984843 $\pm$ 287237 | -1676288 $\pm$ 462837 | -1582742 $\pm$ 227221 | *-1547279* $\pm$ 241625 | -1602142 $\pm$ 342612 | -1586352 $\pm$ 65792 | -1969635 $\pm$ 45241 | -1876374 $\pm$ 33468 | **-1463601** $\pm$ 121678 |

*Note. Red indicates the best performance across all benchmarks. Underscore highlights the best performance among symbolic-based approaches.*

number of rules and overall model performance. They show that merely increasing the number of rules does not invariably improve performance, as an excess can lead to complexity and overfitting, whereas too few can restrict the model's performance. *2. Warm Start:* When the number of rules is small, the model's representational capability is limited, so the initial conditions (e.g., the warm start) become more critical. A warm start can guide the model toward better solutions early on, helping it learn an effective policy faster and avoid getting stuck in poor local optima. In contrast, with a larger rule set, the model has greater capacity to explore various configurations. In such cases, relying on an external initialization can sometimes constrain exploration, preventing the model from fully leveraging its increased flexibility.

## 5.2 INTERPRETABILITY EVALUATION

**Quantitative Evaluation (RQ2):** Common metrics for interpretability include fidelity, comprehensibility, and cognitive load. In our approach, measuring fidelity is unnecessary because it is primarily relevant in post-hoc explanation settings—where one model attempts to approximate and explain the behavior of another. In contrast, both the tree-based approaches (other baselines) and our model are inherently self-explainable, meaning it achieves 100% fidelity by design, as the decisions made by the model directly reflect its underlying logic. For comprehensibility and cognitive load, our rule-based system offers a structured and measurable format. The rule ensembles consist only of boolean propositions formed through simple comparisons (e.g., feature > threshold) and logical conjunctions. In general, comprehensibility is negatively correlated with model complexity, while cognitive load increases with complexity. We define model complexity in terms of the total number of rules, conditions, and logical operations. Models with more parameters and operations tend to be harder to understand and place a greater cognitive burden. Since the performance of $\pi_{affine}$-D, and D-SDT is not directly comparable, we instead evaluate them in terms of interpretability. Using model size as a proxy for complexity, we observe that across all environments our models require, on average, 61.23 symbolic units (number of parameters) to achieve the reported performance. In contrast, SYMPOL uses 95.6 symbolic units (approximately one per node). According to Murdoch et al. (2019), interpretability involves simulatability, modularity, and low complexity. A model is simulatable if its predictions can be easily followed and computed by humans; modular if its components are individually understandable; and low-complexity if it uses few parameters or rules. Our rule-based method satisfies all three. Each decision is made via explicit if-then rules (simulatability), each rule is independently interpretable (modularity), and the total number of rules is kept under 30, ensuring the model remains simple and auditable (low complexity). Although symbolic trees like SYMPOL's are locally simulatable (a single path explains a decision), shared internal nodes mean

edits are not truly modular, and trees tend to grow deep and wide on harder tasks, so they quickly lose low complexity and thus do not fully satisfy all three criteria simultaneously.

**Qualitative Interpretability Case Studies - MountainCar (RQ3):** In this section, we use the MountainCar-v0 environments from OpenAI Gym as examples to illustrate the interpretability of our proposed reinforcement learning algorithm. The Mountain Car environment involves a car situated between two hills, tasked with reaching the flag at the top of the right hill. Due to insufficient engine power, the car must build momentum by oscillating back and forth. The system state is characterized by two continuous variables: the car's position ($p$) and its velocity ($v$). The agent performs one of three discrete actions: push left, do nothing, or push right. For each action, a set of rules evaluate the current state by applying specific conditions based on $p$ and $v$, assigning corresponding scores that influence the decision-making process. Consider a typical state where $p = -0.5$ (slightly to the left of the center) and $v = 0.02$ (moving rightward). For the action *Do Nothing*, one rule imposes a penalty of $-13.485$ since the velocity $v$ exceeds a small positive threshold 0.001, discouraging inaction that could result in insufficient momentum to overcome the hill.

For the action *Push Left*, only one rule applies, which assigns a positive score as the velocity $v$ is above a minimal threshold, encouraging a slight push left to maintain momentum without overcompensating. Conversely, the action *Push Right* activates multiple rules: three contribute $+57.59$ because the velocity $v$ surpasses different threshold values for the corresponding rules, strongly encouraging a push right to build momentum; another rule provides $+11.775$ since both the position $p \geq -0.577$ and velocity is positive are satisfied, indicating that a rightward push is beneficial without risking overshooting the valley; There is also one rule contributing an additional $+1.361$ because the position $p \leq -0.449$, $p \geq -0.874$, and $v \leq 0.023$ hold true, subtly encouraging a push right to maintain the car within a stable range. The cumulative score for *Push Right* thus amounts to $+70.725$, clearly favoring this action. This comprehensive rule-based scoring system demonstrates that, for the given state ($p = -0.5$, $v = 0.02$), pushing right is the optimal choice as it significantly builds the necessary

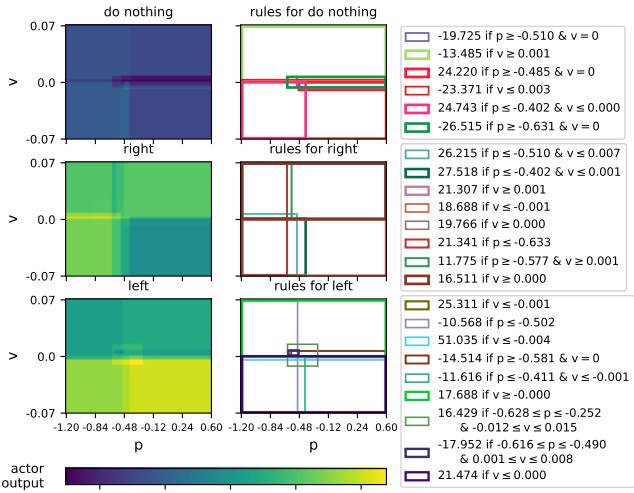

Figure 4: **The learned `MountainCar` rules recover the classic momentum-building strategy in a compact and interpretable form.** The figure visualizes the actor outputs over the $(p, v)$ state space for each action, where higher heatmap values indicate that an action is more strongly preferred in that region. Overlaid squares mark the domains where individual symbolic rules apply, revealing how simple local conditions on position and velocity piece together into the global rocking behavior required to reach the goal; additional visualizations and discussion are provided in Appendix M.

momentum to ascend the hill, outweighing the moderate incentives to push left and the discouragement from doing nothing. We have also visualized the rules of MountainCar in Figure 4 with all rules presented in Appendix M. One can easily identify the corresponding areas associated with a state and locate the applicable rules. To better understand the learned policy, we analyze its behavior in the MountainCar environment in the neighborhood of the valley bottom. The state is two-dimensional, $s = (p, v)$, where $p$ denotes the horizontal position of the car and $v$ its velocity. The car starts in the valley and must build up sufficient kinetic energy by rocking back and forth in order to reach the goal on the right hill. Because the engine is too weak to drive straight up, the policy must carefully coordinate position and velocity to accumulate energy.

In our rule-based policy, one of the most salient decision boundaries appears for positions *to the right of* the valley minimum, i.e. for $p > -0.48$, which is close to the bottom of the track. In this region the height of the track changes only slightly, so the gravitational potential is nearly constant and the immediate direction of motion is determined almost entirely by the sign of the velocity $v$.

Consequently, the line $v = 0$ for $p > -0.48$ becomes a natural decision boundary in the $(p, v)$-plane. States with $p > -0.48$ and $v > 0$ correspond to the car already moving to the right; the optimal action is to keep accelerating right to maximize its speed for the upcoming climb. In contrast, states with $p > -0.48$ and $v < 0$ indicate that the car has started to roll back to the left after an unsuccessful attempt; the best response is to accelerate left and commit to a full swing back, so that the next rightward run can reach a higher point on the hill. Exactly at $v = 0$ the car is at a turning point, momentarily stationary before choosing whether to move left or right, which is why this line marks the decision boundary used by our policy.

**Qualitative Interpretability Analysis - Real Time HVAC control (RQ3):** Alongside standard Gym benchmarks, we evaluated our algorithm on a challenging, real-world HVAC control problem in a biological laboratory, where maintaining temperature within a narrow band by setting effective heating (htg) and cooling (clg) set points is vital to preserving the viability of cultured viruses. In such safety-critical settings, interpretability is crucial: when RL governs biocontainment HVAC systems, even minor errors can destroy cultures or compromise containment. As shown in Table 1, our rule-based actor approach delivers performance nearly equal to—or even surpassing—that of traditional baseline methods. More than this, our proposed algorithm also delivers interpretability by explaining why and how each decision is made in human-understandable language, enabling users to assess the correctness of its rules.

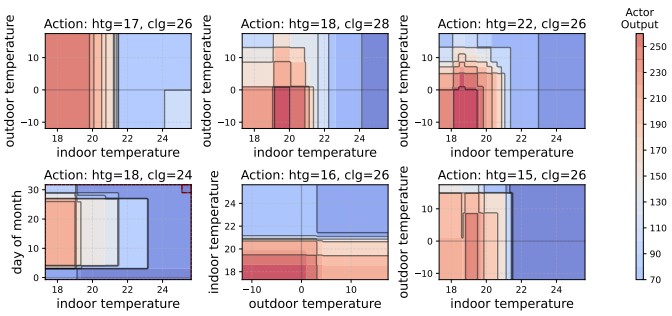

Figure 5: **The HVAC rule ensemble encodes intuitive, human-readable control patterns that balance comfort and energy use.** This figure shows heatmaps of the actor output over key HVAC state variables, where higher values correspond to a stronger preference for each heating–cooling action. The resulting patterns align with domain intuition (e.g., increasing heating when indoor temperatures are low and outdoor conditions are cold), illustrating how NSAC produces a policy that is both effective and easy to interpret; further qualitative examples and time-series analyses are included in Appendix K.

Consider the very first rule as an example: *if outdoor temperature $\leq$ $-4.5$; htg setpoint $\leq$ 20.0; air temperature $\leq$ 21.1503, and HVAC electricity demand rate $\geq$ 1452.0250$(kW)$, +37.6076* What it means, in plain English: When all four of these conditions hold: 1. Outdoor temperature is extremely low; 2. The previous heating setpoint remains at a moderate level; 3. The indoor air temperature is still below the desired comfort threshold; 4. The HVAC system is already drawing high power, then this rule "fires," adding +37.6 to the score for choosing (htg = 15°C, clg = 26°C), i.e., heat turns on below 15 °C and cooling above 26 °C. A large positive weight like +37.6 signifies that, in this scenario, raising the heating setpoint to 15°C is especially beneficial. Based on the visualized rules of HVAC control in Figure 5, when it is freezing outside, your target heating level is still under typical comfort, the room hasn't yet warmed up, and the system is working hard—so raising the heat setpoint to 15°C markedly improves comfort, which this rule captures (see details in Appendix K).

## 6 CONCLUSION

In conclusion, this paper introduces a neural+symbolic approach within the actor-critic framework. This method leverages both the robust function approximation capabilities of the neural-networks critic and the interpretability provided by rule-based actor models. Users can inspect how the model makes each decision and form a view about its underlying reasoning. The limitations of this work are discussed in Appendix A. In the future, we plan to assess the scalability of our method in more complex environments across different domains, such as robotics, finance, healthcare, or autonomous driving, and adapt with other actor-critic frameworks, such as Soft Actor-Critic.

## ETHICAL STATEMENT

This work does not involve human subjects, personal data, or identifiable information. All experiments are conducted entirely in simulated environments, including stylized energy-system tasks designed to probe safety and performance trade-offs. These simulations do not interface with or control any real-world infrastructure. Our proposed NSAC framework aims to improve the interpretability and trustworthiness of reinforcement learning systems by combining symbolic rules with statistical learning. This direction supports the broader goal of promoting transparency in AI decision-making. While the symbolic actor promotes explainability, we acknowledge the importance of careful rule design to avoid unintended consequences in safety-critical domains. However, our current implementation is strictly research-oriented and intended for controlled, offline evaluation.

## REPRODUCIBILITY STATEMENT

To support reproducibility, we include comprehensive details of our training procedures, hyperparameter settings, and evaluation metrics within the main text and appendix. Upon publication, we will release the full codebase, including: Random seeds and environment specifications, Training scripts to replicate all experiments, Plotting utilities to reproduce all figures. All results can be reproduced using a single command-line interface without manual intervention. Pretrained models and logs will also be provided to facilitate benchmarking and verification.

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

# A    LIMITATION AND FUTURE WORK

A fundamental limitation of our approach lies in the natural trade-off between interpretability and performance. While our symbolic, rule-based actor offers clear advantages in terms of transparency and simulatability, this structure inherently constrains policy expressiveness compared to fully neural counterparts—particularly in complex, high-dimensional environments. Although NSAC achieves competitive performance across benchmark tasks, there may be scenarios where its symbolic policy cannot match the nuanced optimization achievable by deep learning-based policies.

NSAC uses rule ensembles over input features, so the candidate rule space grows with the number of attributes and thresholds. In practice, we keep policies compact and simulatable by capping the rule budget and rule depth. This makes NSAC best suited to low–to–medium dimensional feature spaces or to settings with a good compact feature representation; truly high-dimensional inputs would require prior feature extraction or a hybrid neural–symbolic architecture. Also, boosting-based rule updates are currently performed per action, so runtime scales roughly linearly with the number of discrete actions. Our experiments use small to moderate action spaces, and we control cost by limiting rules per action and exploiting trivial parallelism across actions. Very large action spaces would likely require extensions such as shared rule ensembles or multi-class boosting. NSAC is not well suited to highly partially observable or sparse reward environments, such as PONG Delfosse et al. (2024), that demand rich temporal memory or very high-capacity function approximators.

Moreover, interpretability is not a binary property but exists on a spectrum. Different symbolic approaches offer varying degrees of clarity, modularity, and simulatability, often depending on the complexity of the rule structure, the nature of the environment, and the domain-specific knowledge embedded within the rules. As such, the perceived interpretability of a symbolic method may vary across users and use cases. Practitioners must therefore carefully consider the type and level of interpretability required for their application—balancing traceability with scalability—when choosing or designing a symbolic component. Future work may explore adaptive symbolic structures or hybrid neuro-symbolic policy representations to better navigate this trade-off across diverse tasks.

## B  DETAILED RULE UPDATE

*Step 1*: In time-step $t$, for $f_a$, we first **select a new query** $q_t$ from a candidate set $\mathcal{Q}$ by solving

$$q_t = \arg \max_{q \in \mathcal{Q}} \text{obj}\big(q; f_a^{(t-1)}\big), \tag{8}$$

where we adopt OGB (Yang et al., 2024) for the objective function. Let $\boldsymbol{q} = (q(\boldsymbol{x_1}), \ldots, q(\boldsymbol{x_n}))^T \in \{0,1\}^n$ is the query vector of the query $q$, and $\boldsymbol{g}^{(t-1)} = (\partial L_\lambda(f_a^{(t-1)})/\partial f_a^{(t-1)}(\boldsymbol{x_1}), \ldots, \partial L_\lambda(f_a^{(t-1)})/\partial f_a^{(t-1)}(\boldsymbol{x_n}))^T$ is the gradient vector of the empirical risk with respect to the predicted values of $f_a^{(t-1)}$. $\boldsymbol{g}_\perp^{(t-1)}$ and $\boldsymbol{q}_\perp$ are the projection of $\boldsymbol{g}$ and $\boldsymbol{q}$ onto the orthogonal complement of the range of $\boldsymbol{Q}_{t-1} = (\boldsymbol{q_1}, \ldots, \boldsymbol{q_{t-1}})^T$, where $\boldsymbol{q_i} = (q_i(\boldsymbol{x_1}), \ldots, q_i(\boldsymbol{x_n}))^T \in \{0,1\}^n$ is the query vector of the query $q_i$ for $i = 1, \ldots, t-1$. We express the objective function as follows: $\text{obj}(q; f_a^{(t-1)}) = \text{obj}_{\text{ogb}}(q) = |\boldsymbol{q}_\perp^T \boldsymbol{g}_\perp^{(t-1)}|/\|\boldsymbol{q}_\perp\|$. The candidate query set $\mathcal{Q}$ comprises all possible conjunctions of propositions, where each proposition corresponds to a condition that splits the attribute space based on attribute values of given data points (e.g., threshold-based splits). The objective function is maximized using greedy or beam search.

*Step 2:* After adding the updated new query, we re-calculate the optimal weight vector by solving

$$\mathbf{w}_t = \arg \min_{\mathbf{w} \in \mathbb{R}^t} \hat{L}_\lambda\big(\boldsymbol{Q}_t, \mathbf{w}\big), \tag{9}$$

where $\mathbf{Q}_t = \mathbf{Q}_{t-1} \cup \boldsymbol{q}_t$ is the $n \times t$ matrix with the outputs of the selected queries.

*Step 3:* After there are $k$ rules in the system, instead of continuing to add a new rule for each time, we adopt the **post-processing replacement** step of boosting algorithm (Shalev-Shwartz et al., 2010). In each iteration, we remove one rule with minimal absolute weight, leaving $k - 1$ rules in the model: $k^\star = \arg \min_{1 \le r \le k} |w_r|$, and remove the query function $q_{k^\star}$ together with its coefficient $w_{k^\star}$, thereby reducing the ensemble to $f^{(k^-)}$ with $k - 1$ terms. Then, we need to recompute the weight vector $\mathbf{w}_{k^-} = (w_r)_{r \in [k], r \neq k^\star}^T$ ([k]={1,…,k}) of $f^{(k^-)}$ along with the updated query functions $\{q_r\}_{r \in [k], r \neq k^\star}$: $\mathbf{w}_{k^-} = \arg \min_{\mathbf{w} \in \mathbb{R}^{k-1}} \hat{L}_\lambda\big(\mathbf{Q}_{k^-}, \mathbf{w}\big)$, where $\mathbf{Q}_{k^-} = \{q_r\}_{r \in [k], r \neq k^\star}$ is the $n \times (k-1)$ matrix with the outputs of the selected queries.

*Step 4:* After this, repeat *Step 1* to add a new rule to the system. In the end, the weights of all the $k$ rules are recalculated (same as 8 and 9) in *Step 2*. If the expected loss does not decrease after the post-processing procedure, the model before post-processing is restored. The performance of the post-processing with fully-corrective boosting is guaranteed by the following theorem.

---

**Algorithm 2** Orthogonal Gradient Boosting (OGB)

---

1: **Input:** data $(\boldsymbol{x}_i, y_i)$, number of rules $k$ and $\boldsymbol{g}_t$
2: $f^{(0)} = 0$
3: **for** $t = 1, \ldots, k$ **do**
4:     $q_t = \arg \max_q |\boldsymbol{g}_{\perp t}^T \boldsymbol{q}|/\|\boldsymbol{q}_\perp\|$
5:     $\mathbf{w}^{(t)} = \arg \min_{(w_1, \ldots, w_t) \in \mathbb{R}^t} \hat{L}_\lambda(\sum_{j=1}^t w_j q_j)$
6:     $f^{(t)}(\cdot) = w_1^{(t)} q_1(\cdot) + \cdots + w_t^{(t)} q_t(\cdot)$
7: **end for**
8: **Output:** $f^{(k)}$

---

---

**Algorithm 3** Rule Replacement Steps

---

**Input:** data $(\boldsymbol{x}_i, y_i)$, $i = 1, \ldots, n$, original rule ensemble with $k$ rules $f_k^{(0)}$, maximum iteration number $T$ and $\boldsymbol{g}$
**for** $t = 1, \ldots, T$ **do**
  Find the index of the smallest weight absolute value $r = \arg\min_{j \in 1, \ldots, k} |w_j^{(t-1)}|$
  $\tilde{\mathbf{w}}^{(t-1)} = \arg\min_{\mathbf{w} \in \mathbb{R}^{k-1}} \hat{L}_\lambda(\sum_{i \in [k]-\{r\}} \tilde{w}_i^{(t-1)} q_i)$
  $\hat{f}_{k-1}^{(t-1)}(\cdot) = \sum_{i \in [k]-\{r\}} w_i^{(t-1)} q_i(\cdot)$
  Find the query $q_k^{(t)} = \arg\max_q \|q^T \boldsymbol{g}_\perp\|/|q_\perp|$
  $\mathbf{w}^{(t)} = \arg\min_{\mathbf{w} \in \mathbb{R}^k} \hat{L}_\lambda(\sum_{i \in [k]-\{r\}} w_i q_i + w_r q_r^{(t)})$
  Let $\delta = \hat{L}_\lambda(f^{(t)}) - \hat{L}_\lambda(\sum_{i \in [k]-\{r\}} w_i^{(t)} q_i + w_r^{(t)} q_r^{(t)})$
  **if** $\delta > 0$ **then**
    $f_k^{(t)} = \sum_{i \in [k]-\{r\}} w_i^{(t)} q_i + w_r^{(t)} q_r^{(t)}$
  **else**
    break
  **end if**
**end for**
**Output:** $f_k^{(t)}$

---

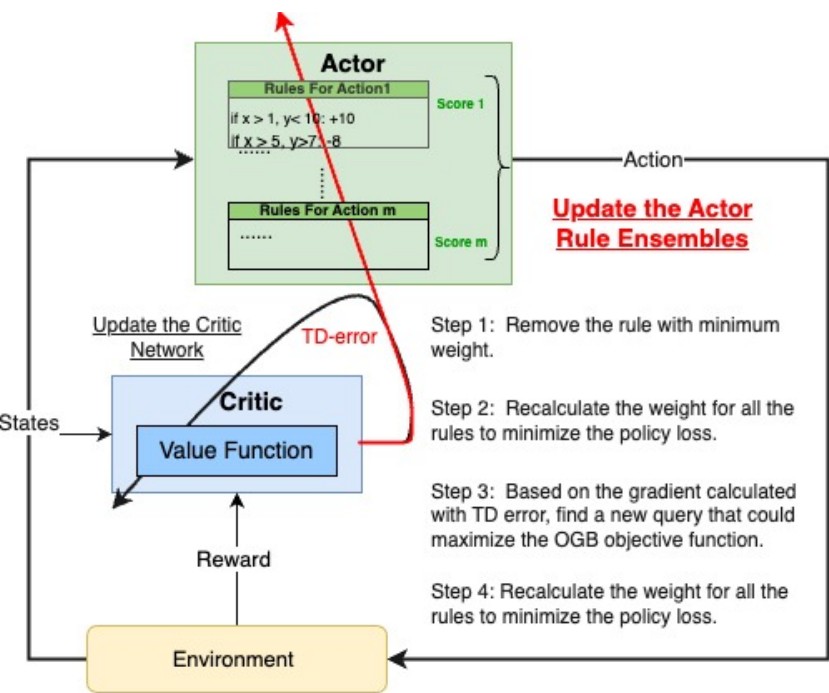

Figure 6: Our proposed NSAC algorithm.

## C  THEORETICAL ANALYSIS

We present a comprehensive theoretical convergence analysis of our NSAC algorithm. Under standard assumptions, our algorithm converges to a local optimum. We assume the following conditions:

**Assumption C.1** (Step Sizes)**.** Assume $\alpha_{(\mathbf{w},\mathbf{q})}$ be the learning rate for actor and let $\alpha_\phi$ be the learning rate for the critic. The sequences $\{\alpha_\phi(t)\}$ and $\{\alpha_{(\mathbf{w},\mathbf{q})}(t)\}$ are positive and satisfy

$$\sum_{t=0}^\infty \alpha_\phi(t) = \infty, \quad \sum_{t=0}^\infty \alpha_{(\mathbf{w},\mathbf{q})}(t) = \infty, \sum_{t=0}^\infty \left(\alpha_\phi(t)^2 + \alpha_{(\mathbf{w},\mathbf{q})}(t)^2\right) < \infty, \quad \lim_{t \to \infty} \frac{\alpha_{(\mathbf{w},\mathbf{q})}(t)}{\alpha_\phi(t)} = 0,$$

so that the critic update occurs on a faster timescale.

**Assumption C.2** (Markovian Sampling & Ergodicity)**.** We assume mild mixing or ergodicity conditions ensuring that each state-action pair is visited infinitely often in the limit, and that standard stochastic approximation methods apply.

**Assumption C.3.** We assume a Lipschitz-smooth approximator for $V_\phi(s)$. We assume the "compatibility" holds or that any residual bias in the critic's estimate does not prevent recovering the true gradient direction in expectation.

**Theorem C.4.** *(Theorem 2.9 in (Shalev-Shwartz et al., 2010)) Let $L(f)$ be the risk function defined by a $\beta$-smooth loss function, where the expectation is with respect to an arbitrary distribution over $X \times Y$. Let $\lambda > 0$ be a scalar, $f$ is an additive rule ensemble with $k$ rules, $\bar{f}$ is a reference rule ensemble with $k_0$ rules, if $k + 1 \geq k_0(1 + 16/\lambda^2)$, and assume that $L$ is $(k + 1 + k_0, \lambda)$-sparsely-strongly convex. Additionally, let $\tau$ be an integer such that*

$$\tau \geq \frac{\lambda(k + 1 - k_0)}{2\beta} \log\left(\frac{L(0) - L(\bar{f})}{\epsilon}\right). \tag{10}$$

*Then if the fully corrective boosting is run for $k$ iterations and its last predictor is provided as input for the post-processing replacement procedure, which is then run for $\tau$ iterations, then when the procedure terminates at time $t$, we have $L(f^{(t)}) - L(\bar{f}) \leq \epsilon$.*

**Theorem C.5.** *Let $\phi$ be the fixed parameters of a critic, inducing a fixed value function $V_\phi(s)$ for each state $s \in \mathcal{S}$. Let $f$ be a parameterized function (e.g., the actor's approximation to the advantage) that is updated iteratively according to the update rule, the sequence $\{f_k\}$ converges to $\bar{f}$.*

*Proof.* We first calculate the risk for rule ensembles for $\bar{f}$ and $f_k$ based on the loss functions: $L(\bar{f}) = -\log \pi_{\bar{f}}(a_t|s_t)\bar{f}$ and $L(f_k) = -\log \pi_{f_k}(a_t|s_t)f_k$, respectively. When $R(\cdot)$ is error bounded, there exists a lower bound with a constant $\gamma > 0$ satisfying $\left|L(f_1) - L(f_2)\right| \geq \gamma \|f_1 - f_2\|$, for any $f_1$ and $f_2$ in the parameter space. Based on Theorem C.4, when Equation 10 is satisfied, we have $\|f_k - \bar{f}\| \leq \frac{1}{\gamma}\left|L(f_k) - L(\bar{f})\right| \leq \frac{\varepsilon}{\gamma}$. Hence, bounding the difference in risk $|L(f_k) - L(\bar{f})|$ by $\varepsilon$ forces the parameter distance $\|f_k - \bar{f}\|$ to lie within $\frac{\varepsilon}{\gamma}$. □

**Theorem C.6** (Convergence of NSAC)**.** *Let $\{(w_t, q_t), \phi_t\}$ be the sequence of actor and critic parameters updated via Equations 6 and 7 under the Assumptions C.1–C.3. Then $\{(w_t, q_t), \phi_t\}$ converges* almost surely *to a set of stationary points of the associated ordinary differential equation (ODE) system: $\dot{\phi} = F(\phi; (\mathbf{w}, \mathbf{q})), (\dot{w}, \dot{q}) = G((\mathbf{w}, \mathbf{q}); \phi)$ where $F$ represents the temporal-difference (critic) dynamics, and $G$ represents the policy-gradient (actor) dynamics. In particular, $(w, q)$ converges to a* local optimum *of $J(w, q)$, where $J$ is the expected cumulative reward.*

*Proof.* We employ the concepts of *Two-timescale* theorems (Borkar & Meyn, 2000; Wu et al., 2020) and Theorem C.5 for the proof. Please refer to Appendix E for the detailed proof of convergence. □

# D ACTOR'S LOSS WITH L2 REGULARIZATION

**Theorem D.1** (Local Strong Convexity of Actor's Loss with L2 Regularization). *Consider the actor's loss function in an A2C framework defined as*

$$L(\theta) = -\mathbb{E}_{s\sim\rho^\pi, a\sim\pi_\theta}\left[A^\pi(s,a)\log\pi_\theta(a|s)\right] + \frac{\lambda}{2}\|\theta\|^2,$$

*where $\lambda > 0$ is the regularization coefficient for L2 regularization.*

*We assume:*

1. *The policy $\pi_\theta(a|s)$ is twice continuously differentiable with respect to $\theta$;*

2. *At a local minimum $\theta^*$, the Hessian of the unregularized loss $L_{unreg}(\theta) = -\mathbb{E}[A^\pi(s,a)\log\pi_\theta(a|s)]$ satisfies $\nabla^2 L_{unreg}(\theta^*) \succeq 0$ (positive semi-definite);*

3. *The regularization coefficient $\lambda$ is chosen such that $\lambda > 0$.*

*Then, there exists a neighborhood $\mathcal{U}$ around $\theta^*$ where the regularized loss $L(\theta)$ is strongly convex. Specifically, within $\mathcal{U}$,*

$$\nabla^2 L(\theta) \succeq \lambda I,$$

*where $I$ is the identity matrix in $\mathbb{R}^{d\times d}$.*

*Proof.* We aim to show that the regularized loss function $L(\theta)$ is strongly convex in a neighborhood around $\theta^*$.

The regularized loss can be expressed as the sum of the unregularized loss and the L2 regularization term:

$$L(\theta) = L_{\text{unreg}}(\theta) + L_{\text{reg}}(\theta),$$

where

$$L_{\text{reg}}(\theta) = \frac{\lambda}{2}\|\theta\|^2.$$

Compute the gradient and Hessian of $L(\theta)$:

$$\nabla L(\theta) = \nabla L_{\text{unreg}}(\theta) + \lambda\theta,$$

$$\nabla^2 L(\theta) = \nabla^2 L_{\text{unreg}}(\theta) + \lambda I.$$

At the local minimum $\theta^*$, by assumption:

$$\nabla^2 L_{\text{unreg}}(\theta^*) \succeq 0.$$

Therefore,

$$\nabla^2 L(\theta^*) = \nabla^2 L_{\text{unreg}}(\theta^*) + \lambda I \succeq \lambda I.$$

This shows that the Hessian of the regularized loss at $\theta^*$ is positive definite, as $\lambda > 0$. Since $\pi_\theta(a|s)$ is twice continuously differentiable, $\nabla^2 L_{\text{unreg}}(\theta)$ is continuous in $\theta$. Hence, there exists a neighborhood $\mathcal{U}$ around $\theta^*$ where:

$$\nabla^2 L_{\text{unreg}}(\theta) \succeq 0, \quad \forall\theta\in\mathcal{U}.$$

Within $\mathcal{U}$,

$$\nabla^2 L(\theta) = \nabla^2 L_{\text{unreg}}(\theta) + \lambda I \succeq \lambda I.$$

This inequality holds because $\nabla^2 L_{\text{unreg}}(\theta) \succeq 0$ and $\lambda I$ is positive definite. Since $\nabla^2 L(\theta) \succeq \lambda I$ for all $\theta \in \mathcal{U}$, the loss function $L(\theta)$ is **strongly convex** in the neighborhood $\mathcal{U}$ around $\theta^*$ with strong convexity parameter $\lambda$.

$\square$

**Theorem D.2** (Local Error Bound under Local Strong Convexity). *Let $L : \mathbb{R}^d \to \mathbb{R}$ be continuously differentiable and $\mu$-strongly convex in a neighborhood $\mathcal{N}$ of some point $\hat{f} \in \mathbb{R}^d$. In other words, there exists $\mu > 0$ such that for all $f \in \mathcal{N}$,*

$$L(f) - L(\hat{f}) \ \geq \ \frac{\mu}{2}\|f - \hat{f}\|^2.$$

*Then, for every $f$ in that neighborhood $\mathcal{N}$,*

$$\|f - \hat{f}\| \ \leq \ \sqrt{\frac{2}{\mu}\left(L(f) - L(\hat{f})\right)}.$$

*Proof.* Since $R$ is $\mu$-strongly convex in the neighborhood $\mathcal{N}$ around $\hat{f}$, we have the inequality

$$L(f) - L(\hat{f}) \geq \frac{\mu}{2} \| f - \hat{f} \|^2, \quad \forall f \in \mathcal{N}.$$

Rearrange this to obtain

$$\| f - \hat{f} \|^2 \leq \frac{2}{\mu} \left( L(f) - L(\hat{f}) \right).$$

Taking the square root of both sides yields

$$\| f - \hat{f} \| \leq \sqrt{\frac{2}{\mu} \left( L(f) - L(\hat{f}) \right)}.$$

Thus, in the local region $\mathcal{N}$, a small gap in the objective value $L(f) - L(\hat{f})$ forces $f$ to be close to $\hat{f}$. This is precisely the local *error-bound* property. $\square$

# E    PROOF FOR CONVERGENCE OF OUR PROPOSED ALGORITHM

**Theorem E.1** (Convergence of our Proposed Algorithm). *Let $\{(w_t, q_t), \phi_t\}$ be the sequence of actor and critic parameters updated via Equations 6 and 7 under the Assumptions C.1–C.3. Then $\{(w_t, q_t), \phi_t\}$ converges almost surely to a set of stationary points of the associated ODE system:*

$$\begin{cases} \dot{\phi} = F(\phi; (\mathbf{w}, \mathbf{q})), \\ (\dot{w}, \dot{q}) = G((\mathbf{w}, \mathbf{q}); \phi), \end{cases}$$

*where $F$ represents the temporal-difference (critic) dynamics and $G$ represents the policy gradient (actor) dynamics. In particular, $(w, q)$ converges to a local optimum of $J(w, q)$.*

*Proof.* By Assumption C.1, $\alpha_{(\mathbf{w}, \mathbf{q})}(t)/\alpha_\phi(t) \to 0$. Hence, on the "fast" timescale, we may treat $\theta_t$ as if it were *quasi-static*. Then the critic update is a standard temporal-difference learning procedure (or mean-squared error minimization) *for a fixed policy $\pi_{(\mathbf{w}, \mathbf{q})}$*.

From classical TD-learning results (or generalized linear function approximation theory), we know that $\phi_t$ converges to the set

$$\{ \phi^*(\mathbf{w}, \mathbf{q}) : F(\phi^*(\mathbf{w}, \mathbf{q})); (\mathbf{w}, \mathbf{q}) = 0 \},$$

provided that $\alpha_\phi(t)$ diminishes appropriately and under the usual conditions.

Formally, one shows that for each fixed $(\mathbf{w}, \mathbf{q})$, the ODE

$$\dot{\phi} = F(\phi; (\mathbf{w}, \mathbf{q}))$$

has a globally asymptotically stable equilibrium at $\phi^*(\mathbf{w}, \mathbf{q})$. By the Two-timescale lemma (Borkar & Meyn, 2000; Wu et al., 2020), the actual sequence $\phi_t$ tracks this stable equilibrium as $t \to \infty$.

On the slower timescale, we consider the actor update:

$$(\mathbf{w}, \mathbf{q})_{t+1} = (\mathbf{w}, \mathbf{q})_t + \alpha_{(\mathbf{w}, \mathbf{q})}(t) \nabla_{(\mathbf{w}, \mathbf{q})} \log \pi_{(\mathbf{w}, \mathbf{q})_t}(a_t \mid s_t) \widehat{A}_t.$$

As $\phi_t$ converges quickly to $\phi^*(\mathbf{w}, \mathbf{q})$, the advantage estimate $\widehat{A}_t$ converges to $A^{\pi(\mathbf{w}, \mathbf{q})}(s_t, a_t)$. Hence, up to diminishing approximation errors, the gradient update for our rule based model is driven from the true policy gradient from the policy gradient theorem. Based on Theorem C.5, ignoring the small the errors, we have

$$G((\mathbf{w}, \mathbf{q}); \phi^*(\mathbf{w}, \mathbf{q})) = \nabla_{wq} J(\mathbf{w}, \mathbf{q}).$$

Putting the fast and slow processes together, we return to the coupled ODE; By the previous steps:

$$F(\phi; (\mathbf{w}, \mathbf{q})) = 0 \text{ implies } \phi = \phi^*, \tag{11}$$

$$G((\mathbf{w}, \mathbf{q}); \phi^*) \approx \nabla_{wq} J(\mathbf{w}, \mathbf{q}). \tag{12}$$

Hence, an equilibrium $(\phi^*, (\mathbf{w}, \mathbf{q}))$ of the ODE occurs precisely, when

$$F(\phi^*; (\mathbf{w}, \mathbf{q})) = 0 \quad \text{and} \quad G((\mathbf{w}, \mathbf{q}); \phi^*) = 0,$$

which in turn implies

$$\nabla J(\mathbf{w}, \mathbf{q}) = 0, \quad \text{if the advantage estimation is unbiased.}$$

Therefore, the equilibrium corresponds to a stationary point of $J(w, q)$. Under mild conditions (e.g., local convexity/concavity arguments or strict monotonicity), one concludes that the limit points are *local maxima* of $J(w, q)$.

We now invoke the standard *Two-timescale* theorems (Borkar & Meyn, 2000; Wu et al., 2020). Then the combined updates converge to an *internally consistent* equilibrium of the ODE. By the structure of the ODE, the equilibrium is a stationary point and *local* optimum of $J(w, q)$. □

# F    HYBRID WARM START FOR ACCELERATING CONVERGENCE

In NSAC, we also adopt a novel hybrid training strategy that incorporates both neural network-based actor and rule-based actor. This hybrid strategy seeks to merge the quick learning attributes of neural networks with the clear and straightforward nature of rule-based systems.

**Step 1:** We employ a traditional neural network-based actor-critic model to train the system. In this phase, both the actor and the critic are implemented as neural networks, leveraging their ability to approximate complex functions and capture high-dimensional patterns in the data. The model is trained for a certain number of episodes but not until complete convergence. This pre-training phase is designed to establish a preliminary value function estimation that captures essential features of the optimal strategies in a computationally efficient manner.

**Step 2:** The neural network-based actor is replaced with the rule ensemble actor. This rule-based system is designed to provide greater transparency and interpretability in decision-making. With the rule ensemble actor in place, training continues, utilizing the previously trained neural network critic. The critic assists the new actor in refining its policy through ongoing feedback and value estimation.

## F.1    ABLATION STUDY

In our ablation study, we explored how variations in the rule count and the implementation of a warm start influenced our method. We conducted tests in the CartPole-v1 environment under various rule configurations, specifically using 5, 10, 12, 20, 30, 40, and 50 rules per action. Each configuration was tested both with and without a warm start.

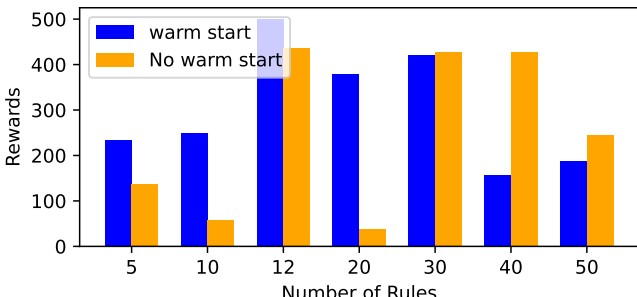

Figure 7: Ablation study on different numbers of rules and warm start.

**1. Number of Rules:**  As depicted in Figure 7, the model configured with 12 rules per action and a warm start yields the highest reward. This outcome suggests that having 12 rules per action provides the model with sufficient flexibility to effectively capture the essential dynamics of the environment without being overly simplistic. Employing fewer than 10 rules, such as 5, may not offer adequate coverage to manage the environment's complexity, resulting in inferior performance. Conversely, a larger number of rules may lead the model to overfit the training data, picking up on noise or irrelevant patterns and diminishing its generalizability to new contexts. The findings from this ablation study highlight a critical balance between the number of rules and overall model performance. They demonstrate that merely increasing the number of rules does not invariably improve performance, as an excess can lead to complexity and overfitting, whereas too few can restrict the model's performance.

**2. Warm Start:** When the number of rules is small, the model's representational capability is limited, so the initial conditions (e.g., the warm start) become more critical. A warm start can guide the model toward better solutions early on, helping it learn an effective policy faster and avoid getting stuck in poor local optima. In contrast, with a larger rule set, the model has greater capacity to explore various configurations. In such cases, relying on an external initialization can sometimes constrain exploration, preventing the model from fully leveraging its increased flexibility. As a result, the model without a warm start may discover a better configuration on its own when there are many rules to learn from.

## G  Continuous Environments

Continuous environments differ fundamentally from the perspective of the theoretical framework we propose. In this work, theoretical analysis is restricted to discrete action spaces, as our method enables us to establish formal convergence guarantees.

In order to guarantee the convergence of our proposed method, we restrict our analysis to discrete action spaces. The convergence proof relies on several technical assumptions—such as bounded gradients, Lipschitz continuity, and stable critic updates—that are typically satisfied in discrete action spaces, but are difficult to guarantee in continuous settings, particularly for actor–critic methods (Borkar & Meyn, 2000; Wu et al., 2020). Discrete policies (e.g., softmax) yield smooth and bounded log-policy gradients, which facilitate stable actor updates and tractable Bellman backups using finite sums. These same conditions are also required for the convergence of the rule-based function class used in OGB. As a gradient-boosting-based algorithm, OGB requires the loss function to be Lipschitz continuous, which means that the gradients are bounded. According to Theorem 2.4, 2.8 and 2.9 in Shalev-Shwartz et al. (2010), Lipschitz continuity is a necessary condition for the learned models to converge to a theoretically optimal model within a certain number of iterations for (fully corrective) boosting methods and the rule replacement steps.

In contrast, continuous action spaces present challenges, such as unbounded gradients (e.g., the policy gradient can become unbounded as the standard deviation in Gaussian policies), integration-based Bellman targets, and higher variance in gradient estimates. These factors violate the core assumptions underpinning both the two-timescale stochastic approximation framework and the rule ensemble model used in our convergence analysis.

# H    RELATED WORK

| Ref | Work (short) | Category | Interp. at scale | P/Pt/E | Performance |
|---|---|---|---|---|---|
| Ernst et al. (2005); Gupta et al. (2017) | Tree-based batch RL | Tree policy (native) | Not interp.(large trees) | N/N/Y | Strong on S/M; degrades on L/complex |
| Tao et al. (2018) | Tree RL for DTRs | Tree policy (native) | Not interp.(multi-stage) | N/N/Y | Hard to scale beyond structured domains |
| Bastani et al. (2018) | Policy extraction | Tree policy (extracted) | Often not interp. (size balloons) | N/Y/N | Tracks teacher; small typical drop |
| Roth et al. (2019) | Conservative Q-Improvement | Tree policy (native) | Not interp.(large trees) | N/N/Y | Strong on S/M; degrades on L/complex |
| Silva et al. (2020) | Differentiable decision trees | Tree policy (diff.) | Not interp.(math notation) | N/N/Y | Solid on modest tasks; mixed scaling |
| Ding et al. (2020) | Cascading Decision Trees | Tree policy (native) | Not interpretable at scale | N/N/Y | Good early; plateaus with complexity |
| Gupta et al. (2015) | Policy Tree(distill.) | Tree policy (distilled) | Often not interpretable at scale | N/Y/N | Tracks teacher; bigger drop for small trees |
| Coppens et al. (2019) | Soft DT distillation | Tree policy (distilled) | Often not interpretable at scale | N/Y/N | Approximates teacher; mild–moderate drop |
| Liu et al. (2018) | Linear Model U-Trees | Tree policy (distilled) | Not interp.(math notation) | N/Y/N (onl) | Good local fidelity; may trail overall |
| Garcez et al. (2018) | SRL with commonsense | Symbolic RL (predefined) | Interpretable (symbolic rules) | Y/N/Y | Strong when priors fit; brittle if misspecified |
| Lyu et al. (2019) | SDRL (symbolic planning) | Symbolic RL (predefined) | Interpretable (modules) | Y/N/Y | Often strong & sample-efficient; planner-dependent |
| Illanes et al. (2020) | Symbolic plans as instr. | Symbolic RL (predefined) | Interpretable (plans) | Y/N/Y | Good when plans align; limited otherwise |
| Landajuela et al. (2021) | Discovering symb. policies | Symbolic policy (from NN) | Not interp.(math notation) | N/Y/N | Near-teacher on seen tasks; drops under shift |
| Hein et al. (2018) | GP-based symbolic rules | Symbolic policy (direct) | Not interp.(math notation) | N/N/Y | Competitive on simpler control; search limits scale |
| Verma et al. (2018) | Programmatic RL (PIRL) | Program synthesis policy | Not interp.(math notation) | N/N/Y | Lags in high-dimensional |
| Ours | NSAC | Symbolic RL | Interpretable(symbolic rules) | N/N/Y | Often strong & sample-efficient |

Table 2: Compact comparison of policy representations and learning setups.

We summarise representative interpretable RL approaches across tree-based, symbolic, and programmatic policy classes. *Interp. at scale* highlights whether interpretability persists as problem size/complexity grows. *P/Pt/E* abbreviates *Predefined knowledge? / Post-hoc from pretrained? / Learn directly from environment?* (Y/N). *Performance* sketches typical empirical behavior. Overall, native tree policies often face scalability–interpretability trade-offs, post-hoc distilled trees track teachers but lose fidelity when heavily pruned, symbolic methods offer strong transparency and sample efficiency when priors/plans match the domain, and programmatic policies remain challenging in high-dimensional settings.

# I  BENCHMARK SETTING

We recognize that certain algorithms might perform better with changes to the environment (e.g., adjusting the number of parallel environments or modifying rewards). Still, finding a single modification that consistently benefits all tested algorithms remains challenging. Therefore, we rely on the most basic version of each environment to maintain consistency across all methods. Additionally, it is important to note that the complexity of our model is designed to balance comprehensibility with the cognitive effort needed to understand all of the rules involved. Therefore, in our paper, we deliberately selected 12 rules per action model, which helps ensure the interpretability of our proposed method.

All experiments in this paper are conducted on a computer with processor "3.1GHz 6-Core Intel Core i5" and a memory of "72GB 2133 MHz DDR4."

## J  ENVIRONMENT DESCRIPTIONS

- **CartPole-v1 (Gym).** The state $s \in \mathbb{R}^4$ is continuous and consists of cart position, cart velocity, pole angle, and pole angular velocity. The action space is discrete with two actions: push left / push right. The reward is $+1$ at each timestep until termination (when the pole angle exceeds the threshold or the cart goes out of bounds).

- **MountainCar-v0 (Gym).** The state $s = (\text{position}, \text{velocity}) \in \mathbb{R}^2$ is continuous. The action space is discrete with three actions: push left / do nothing / push right. The reward is $-1$ at each timestep until the car reaches the goal position at the top of the hill.

- **Acrobot-v1 (Gym).** The state $s \in \mathbb{R}^4$ is continuous and encodes the two joint angles and their angular velocities. The action space is discrete with three torque actions applied at the actuated joint. The reward is $-1$ at each timestep until the end-effector reaches the target height, at which point the episode terminates.

- **Blackjack-v1 (Gym).** The state is a discrete tuple (player sum, dealer showing, usable ace flag). The action space is discrete with two actions: hit / stick. The reward is $+1$ for a win, $-1$ for a loss, and $0$ for a draw.

- **Postman (grid-world).** The state is discrete and encodes the agent's grid location together with the locations and delivery status of parcels (picked up / delivered). The action space is discrete: move north / south / east / west and pick-up / drop-off when applicable. The reward is $-1$ per timestep and a positive bonus when a parcel is successfully delivered, encouraging short, efficient delivery routes.

- **HVAC (building control).** The state is high-dimensional and continuous, including (for each thermal zone) indoor air temperature, humidity, and $CO_2$ concentration, together with outdoor temperature, time-of-day, and occupancy-related features. Actions are discrete combinations of heating and cooling setpoints (e.g., heating and cooling setpoints chosen from a finite grid of admissible values). The reward at each control step penalizes both energy consumption and comfort violations: large negative penalties are applied when zone temperatures leave the comfort band, and a smaller penalty is proportional to electricity consumption, so the agent is encouraged to maintain comfort with minimal energy use.

# K  RULES VISUALIZATION FOR HVAC CONTROL WITH SINERGYM

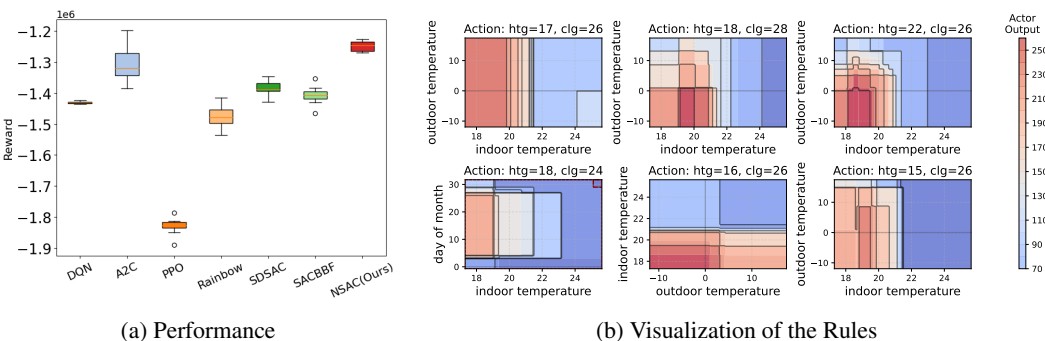

(a) Performance                    (b) Visualization of the Rules

Figure 8: Comparing the performance of NSAC with baseline methods and visualizing the rules.

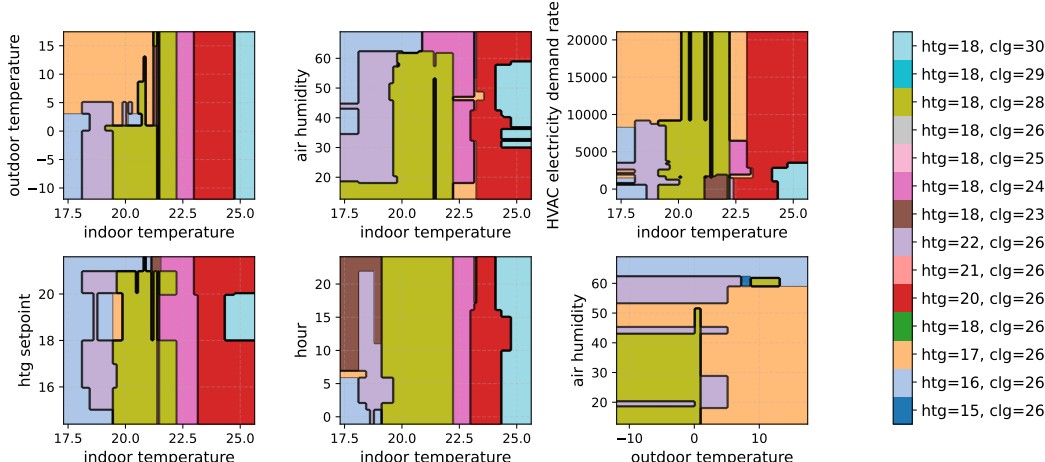

Figure 9: Visualization of action selection for NSAC.

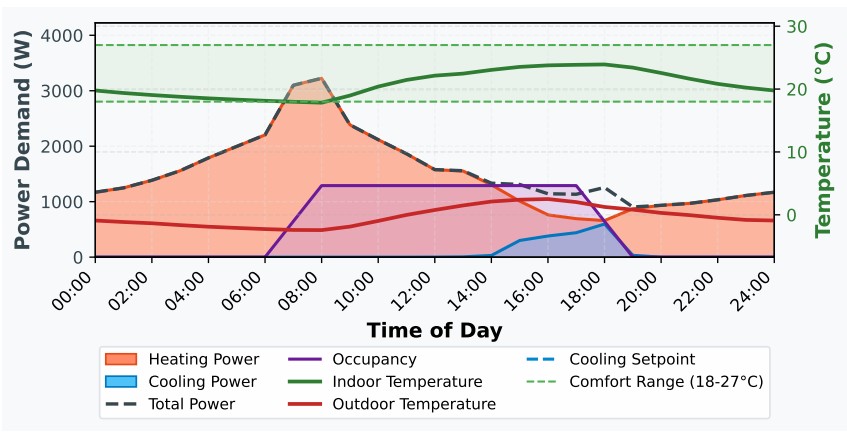

Figure 10: Average performance of our proposed NSAC algorithm.

Typical rules for HVAC control with sinergym:

Action htg=16, clg=26:

$+50.5500$ if $-11.7075 \leq$ outdoor_temperature $\leq 15.0$ & htg_setpoint $\leq 20.0$ & air_temperature $\leq 20.8431$

$+42.3222$ if hour $\leq 6.0$ & $-9.5 \leq$ outdoor_temperature $\leq 4.3200$ & air_temperature $\leq 21.1095$

$+33.7433$ if hour $\leq 6.0$ & outdoor_temperature $\leq 3.0$ & HVAC_electricity_demand_rate $\geq 3524.0383$

$+30.0530$ if air_temperature $\leq 19.4323$ & HVAC_electricity_demand_rate $\geq 3492.0372$

$+23.8401$ if hour $\leq 6.0$ & outdoor_temperature $\leq -6.0$ & HVAC_electricity_demand_rate $\geq 1836.0156$

$+19.5165$ if outdoor_temperature $\leq -5.0$ & air_temperature $\leq 20.8367$ & air_humidity $\geq 34.4557$

$+19.3194$ if outdoor_temperature $\leq 4.3200$ & clg_setpoint $\leq 26.0$ & air_temperature $\leq 20.7468$
      & air_humidity $\geq 44.9061$ & HVAC_electricity_demand_rate $\leq 8328.7136$

$+17.1997$ if outdoor_temperature $\leq -6.0$ & air_temperature $\leq 20.6569$ & air_humidity $\geq 34.4701$

$+12.2539$ if outdoor_temperature $\leq 1.1000$ & air_temperature $\leq 19.8760$ & air_humidity $\geq 48.4554$ &
      HVAC_electricity_demand_rate $\geq 2194.6457$

$+10.1033$ if day_of_month $\leq 16.0$ & outdoor_temperature $\leq -9.5$ & air_temperature $\leq 21.4822$

$+9.9095$ if outdoor_temperature $\leq -1.2000$ & clg_setpoint $\leq 28.0$ & air_temperature $\leq 18.6297$ &
      $20.1259 \leq$ air_humidity $\leq 57.8947$

$+6.5348$ if hour $\leq 5.0$ & outdoor_temperature $\leq -6.2000$ & air_humidity $\geq 46.6687$ &
      HVAC_electricity_demand_rate $\leq 6047.4648$

Action htg=18, clg=26:

$+52.4710$ if outdoor_temperature $\leq 2.0$ & htg_setpoint $\geq 18.0$ & air_temperature $\geq 24.6918$ &
      $36.4788 \leq$ air_humidity $\leq 59.9498$ & $655.1863 \leq$ HVAC_electricity_demand_rate $\leq 3286.0521$

$+31.8345$ if $-7.3000 \leq$ outdoor_temperature $\leq -2.4800$ & clg_setpoint $\geq 28.0$ & air_temperature $\geq 22.7824$ &
      air_humidity $\leq 53.0045$ & HVAC_electricity_demand_rate $\leq 2804.0908$

$+25.6196$ if day_of_month $\leq 23.0$ & hour $\geq 21.0$ & htg_setpoint $\geq 18.0$ &
      air_temperature $\leq 21.2843$ & $22.2264 \leq$ air_humidity $\leq 57.4836$ &
      HVAC_electricity_demand_rate $\leq 4727.3044$

$+14.6693$ if hour $\leq 16.0$ & clg_setpoint $\geq 28.0$ & $24.3635 \leq$ air_temperature $\leq 25.3421$ &
      air_humidity $\leq 43.6038$ & $1974.6005 \leq$ HVAC_electricity_demand_rate $\leq 2847.7105$

$+12.8951$ if day_of_month $\geq 17.0$ & $1.0 \leq$ hour $\leq 5.0$ & htg_setpoint $\leq 16.0$ &
      air_temperature $\leq 19.6280$ & HVAC_electricity_demand_rate $\geq 728.1273$

$+6.6281$ if $14.0 \leq$ hour $\leq 15.0$ & outdoor_temperature $\leq -2.4800$ & $24.1909 \leq$ air_temperature $\leq 25.1225$
      & air_humidity $\leq 44.0097$ & HVAC_electricity_demand_rate $\leq 2856.4850$

$+3.6962$ if outdoor_temperature $\leq -2.4800$ & air_temperature $\geq 25.9993$

$+2.2511$ if hour $\geq 18.0$ & $-5.0 \leq$ outdoor_temperature $\leq -2.4800$ & air_temperature $\geq 22.3801$
      & air_humidity $\leq 52.5242$ & $1164.7935 \leq$ HVAC_electricity_demand_rate $\leq 1443.5973$

$+1.7191$ if $-7.3000 \leq$ outdoor_temperature $\leq -3.0$ & clg_setpoint $\geq 30.0$ &
      air_humidity $\leq 52.1981$ & $1411.8365 \leq$ HVAC_electricity_demand_rate $\leq 2783.3056$

$+1.6791$ if $5.0 \leq$ day_of_month $\leq 28.0$ & $-3.4000 \leq$ outdoor_temperature $\leq 6.6000$ &
      $22.4682 \leq$ air_temperature $\leq 23.2356$ & HVAC_electricity_demand_rate $\leq 1559.4128$

Action htg=18, clg=25:

+76.0682 if month $\leq$ 2.0 & day_of_month $\leq$ 29.0 & $-2.4800 \leq$ outdoor_temperature $\leq$ 15.0 &
    htg_setpoint $\leq$ 21.0 & air_temperature $\leq$ 20.4542

+35.7797 if month $\leq$ 2.0 & day_of_month $\leq$ 28.0 & outdoor_temperature $\leq$ 15.0 &
    air_temperature $\leq$ 20.4547 & HVAC_electricity_demand_rate $\leq$ 888.4346

+19.5429 if hour $\leq$ 17.0 & outdoor_temperature $\leq$ 4.3200 & air_temperature $\geq$ 21.3982
    & air_humidity $\geq$ 22.1544 & HVAC_electricity_demand_rate $\geq$ 3353.0663

+11.0059 if outdoor_temperature $\leq -7.3000$ & air_temperature $\geq$ 23.0022
    & HVAC_electricity_demand_rate $\geq$ 3343.0970

+10.4447 if month $\leq$ 1.0 & outdoor_temperature $\leq$ 0.0 & air_temperature $\geq$ 22.3607
    & air_humidity $\geq$ 35.8469 & HVAC_electricity_demand_rate $\geq$ 3237.7531

 +9.2884 if day_of_month $\leq$ 28.0 & $-2.4800 \leq$ outdoor_temperature $\leq$ 15.0 & clg_setpoint $\leq$ 25.0
    & air_temperature $\leq$ 20.4644

 +8.4585 if hour $\leq$ 16.0 & air_temperature $\geq$ 23.5872 & HVAC_electricity_demand_rate $\geq$ 3378.0424

 +7.6878 if 2.0 $\leq$ day_of_month $\leq$ 8.0 & hour $\leq$ 15.0 & outdoor_temperature $\leq -6.7000$ &
    air_temperature $\geq$ 22.9565 & HVAC_electricity_demand_rate $\leq$ 5120.2045

 +6.0944 if hour $\leq$ 16.0 & outdoor_temperature $\leq -9.5$ & air_temperature $\geq$ 22.3765

 +6.0912 if $-2.4800 \leq$ outdoor_temperature $\leq$ 1.0 & clg_setpoint $\leq$ 29.0 & air_temperature $\leq$ 18.5048
    & 28.1472 $\leq$ air_humidity $\leq$ 57.5569

 +4.1747 if air_temperature $\geq$ 25.1225 & HVAC_electricity_demand_rate $\geq$ 4135.6304

 +2.7032 if 2.0 $\leq$ day_of_month $\leq$ 29.0 & hour $\geq$ 1.0 & $-11.7075 \leq$ outdoor_temperature $\leq -5.6000$
    & 18.0977 $\leq$ air_temperature $\leq$ 24.9366 & 4026.8635 $\leq$ HVAC_electricity_demand_rate
    $\leq$ 7054.0210

Action htg=18, clg=28:

+47.6207 if outdoor_temperature $\leq$ 1.1000 & air_temperature $\leq$ 21.3847

+42.5021 if month $\leq$ 2.0 & outdoor_temperature $\leq$ 13.0150 & clg_setpoint $\leq$ 29.0 &
    air_temperature $\leq$ 20.8932 & HVAC_electricity_demand_rate $\leq$ 9236.5036

+37.5818 if month $\leq$ 2.0 & htg_setpoint $\leq$ 21.0 & 19.0381 $\leq$ air_temperature $\leq$ 22.2181
    & air_humidity $\leq$ 61.5915

+37.5560 if month $\leq$ 2.0 & htg_setpoint $\leq$ 21.0 & 19.0971 $\leq$ air_temperature $\leq$ 24.1215
    & air_humidity $\leq$ 62.4223

+26.1161 if outdoor_temperature $\leq -3.4000$ & 17.0 $\leq$ htg_setpoint $\leq$ 18.0
    & air_temperature $\leq$ 21.6418 & HVAC_electricity_demand_rate $\geq$ 1616.5198

+25.3651 if month $\leq$ 2.0 & hour $\leq$ 6.0 & outdoor_temperature $\leq$ 8.8000 & air_temperature $\leq$ 21.0920 &
    air_humidity $\leq$ 51.7392

+21.4283 if hour $\leq$ 6.0 & outdoor_temperature $\leq -2.0$ & htg_setpoint $\leq$ 18.0 & air_temperature $\leq$ 20.0191 &
    HVAC_electricity_demand_rate $\geq$ 1266.1656

+19.3831 if air_temperature $\leq$ 19.0718

+18.9771 if clg_setpoint $\geq$ 28.0 & air_temperature $\leq$ 22.6279
    & 3313.9887 $\leq$ HVAC_electricity_demand_rate $\leq$ 9094.2658

+15.9370 if outdoor_temperature $\leq$ 1.0 & htg_setpoint $\leq$ 20.0 & air_temperature $\leq$ 21.1680
    & air_humidity $\geq$ 46.0692 & HVAC_electricity_demand_rate $\geq$ 3766.4288

+11.4827 if outdoor_temperature $\leq$ 1.1000 & air_temperature $\leq$ 18.9835
    & 2059.7858 $\leq$ HVAC_electricity_demand_rate $\leq$ 8526.6964

+11.1224 if 3.0 $\leq$ day_of_month $\leq$ 28.0 & hour $\geq$ 22.0 & outdoor_temperature $\leq -1.0$ & htg_setpoint $\leq$ 18.0
    & 20.5696 $\leq$ air_humidity $\leq$ 57.0262 & HVAC_electricity_demand_rate $\geq$ 2007.6986

# L   RULES VISUALIZATION FOR BLACKJACK-V1

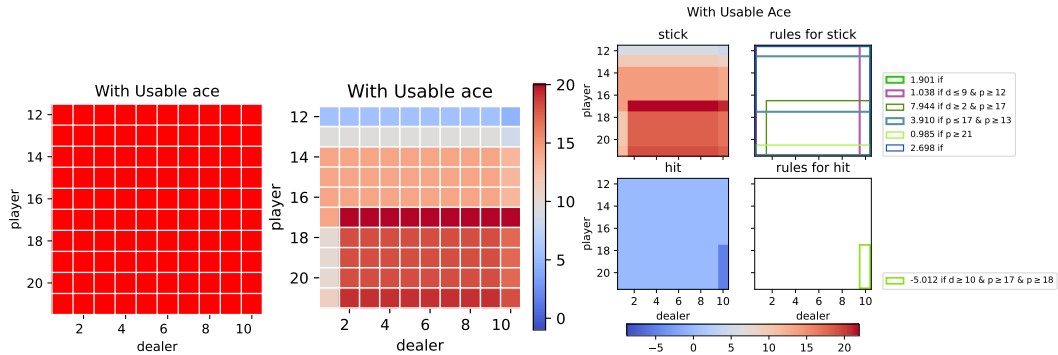

Figure 11: Visualization for action, value, and rules for Blackjack with Ace.

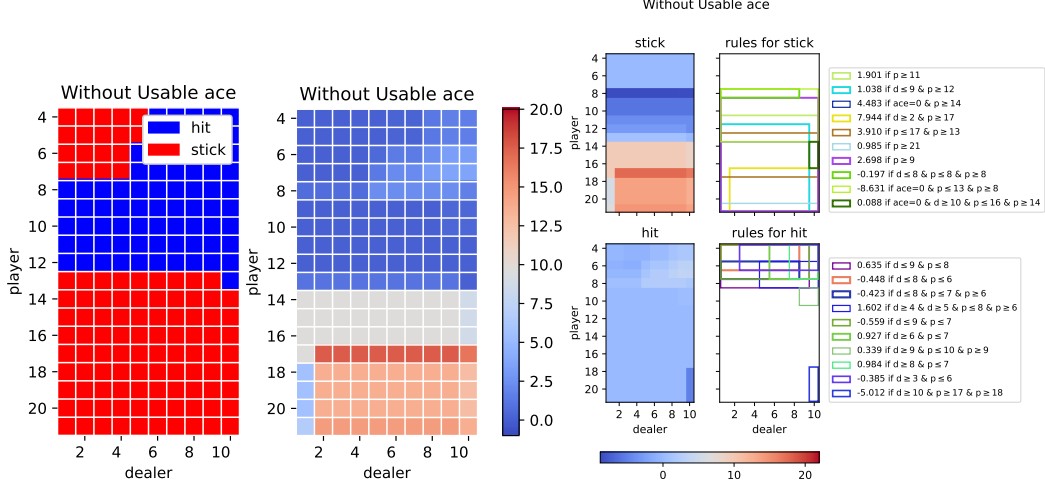

Figure 12: Visualization for action, value, and rules for Blackjack with no Ace.

Action 0 (Stick):

$$
\begin{aligned}
&+1.9012 &&\text{if } p \geq 11 \\
&+1.0376 &&\text{if } d \leq 9 \text{ and } p \geq 12 \\
&+4.4833 &&\text{if ace} \geq 1 \text{ and } p \geq 14 \\
&+7.9438 &&\text{if } d \geq 2 \text{ and } p \geq 17 \\
&+3.9105 &&\text{if } p \leq 17 \text{ and } p \geq 13 \\
&+0.9851 &&\text{if } p \geq 21 \\
&+2.6981 &&\text{if } p \geq 9 \\
&-0.1967 &&\text{if } d \leq 8 \text{ and } d \leq 9 \text{ and } p \leq 8 \text{ and } p \geq 8 \\
&-8.6313 &&\text{if ace} \leq 0 \text{ and } p \leq 13 \text{ and } p \geq 8 \\
&+0.0882 &&\text{if ace} \leq 0 \text{ and } d \geq 10 \text{ and } p \leq 16 \text{ and } p \geq 14
\end{aligned}
$$

Action 1 (Hit):

$$
\begin{aligned}
&+0.6347 && \text{if } d \leq 9 \text{ and } p \leq 8 \\
&-0.4481 && \text{if } d \leq 8 \text{ and } p \leq 6 \\
&-0.4231 && \text{if } d \leq 8 \text{ and } p \leq 7 \text{ and } p \geq 6 \\
&+1.6016 && \text{if } d \geq 4 \text{ and } d \geq 5 \text{ and } p \leq 8 \text{ and } p \geq 6 \\
&-0.5585 && \text{if } d \leq 9 \text{ and } p \leq 7 \\
&+0.9266 && \text{if } d \geq 6 \text{ and } p \leq 7 \\
&+0.3394 && \text{if } d \geq 9 \text{ and } p \leq 10 \text{ and } p \geq 9 \\
&+0.9844 && \text{if } d \geq 8 \text{ and } p \leq 7 \\
&-0.3849 && \text{if } d \geq 3 \text{ and } p \leq 6 \\
&-5.0124 && \text{if } d \geq 10 \text{ and } p \geq 17 \text{ and } p \geq 18
\end{aligned}
$$

## M   MOUNTAINCAR RULES

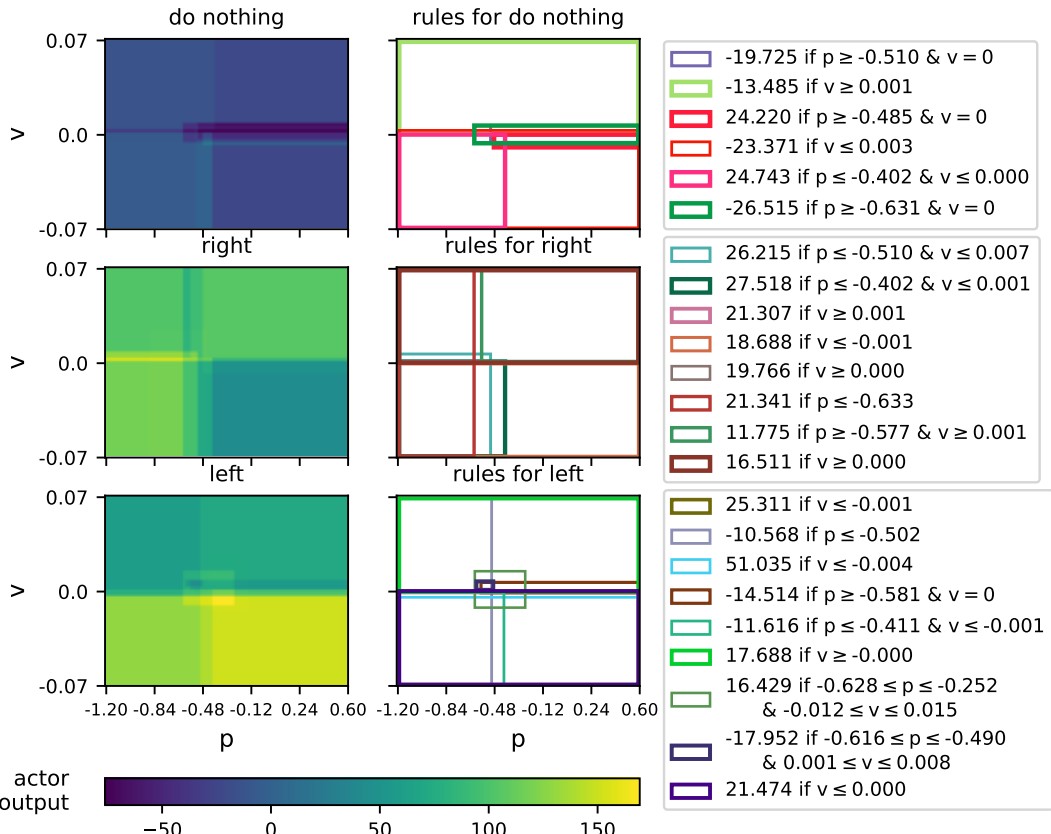

Figure 13: **Visual representation of the rules for MountainCar.** Higher actor output values in the heatmap indicate that the corresponding action is more strongly preferred at that state. The squares indicate the regions of the state space where the corresponding rules apply.

### M.1   CASE STUDY: DECISION BOUNDARY AROUND $p > -0.48$ IN MOUNTAINCAR

To better understand the learned policy, we analyze its behavior in the MountainCar environment in the neighborhood of the valley bottom. The state is two-dimensional, $s = (p, v)$, where $p$ denotes the horizontal position of the car and $v$ its velocity. The car starts in the valley and must build up sufficient kinetic energy by rocking back and forth in order to reach the goal on the right hill. Because the engine is too weak to drive straight up, the policy must carefully coordinate position and velocity to accumulate energy.

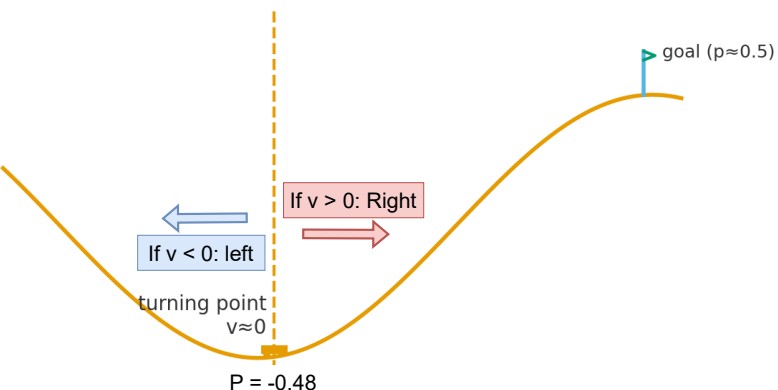

Figure 14: **MountainCar Environment Near the Valley Decision Boundary.** This figure shows the MountainCar track with the goal flag on the right hill and the car rendered near the bottom of the valley at $P = -0.48$. The vertical dashed line marks the region $P > -0.48$, where the policy's decision depends on the sign of the velocity: for $v > 0$ the agent accelerates to the right to build speed toward the goal, whereas for $v < 0$ it accelerates left to commit to a full swing back and gain momentum for the next climb.

In our rule-based policy, one of the most salient decision boundaries appears for positions *to the right of* the valley minimum, i.e. for $p > -0.48$, which is close to the bottom of the track (see Figure 14). In this region the height of the track changes only slightly, so the gravitational potential is nearly constant and the immediate direction of motion is determined almost entirely by the sign of the velocity $v$.

Consequently, the line $v = 0$ for $p > -0.48$ becomes a natural decision boundary in the $(p, v)$-plane. States with $p > -0.48$ and $v > 0$ correspond to the car already moving to the right; the optimal action is to keep accelerating right to maximize its speed for the upcoming climb. In contrast, states with $p > -0.48$ and $v < 0$ indicate that the car has started to roll back to the left after an unsuccessful attempt; the best response is to accelerate left and commit to a full swing back, so that the next rightward run can reach a higher point on the hill. Exactly at $v = 0$ the car is at a turning point, momentarily stationary before choosing whether to move left or right, which is why this line marks the decision boundary used by our policy.

Figure 14 visualizes this situation directly in the environment: the car is rendered near $p \approx -0.48$ at the bottom of the valley, and the vertical dashed line indicates the region $p > -0.48$ where the policy switches between "go left" and "go right" depending on the sign of the velocity.

## M.2 THE RULES

Action Left:

$$
\begin{aligned}
&+25.3111 &&\text{if } v \leq -0.0006471572560258208 \\
&-10.5678 &&\text{if } p \leq -0.5023447513580322 \\
&+53.8697 &&\text{if True} \\
&+51.0351 &&\text{if } v \leq -0.004265955928713083 \\
&-14.5140 &&\text{if } p \geq -0.5813989758491516 \text{ and } v \leq 0.006803702469915152 \\
& &&\text{and } v \geq -0.0008947865571826689 \\
&-11.6162 &&\text{if } p \leq -0.41075775027275085 \text{ and } v \leq -0.000661600707098839 \\
&+17.6882 &&\text{if } v \geq -1.5593287753289978 \times 10^{-5} \\
&+16.4293 &&\text{if } p \leq -0.25240878462791444 \text{ and } p \geq -0.6275408387184143 \\
& &&\text{and } v \leq 0.01510303020477297 \text{ and } v \geq -0.01191 \\
&-17.9516 &&\text{if } p \leq -0.4901819765567779 \text{ and } p \geq -0.615831172466278 \\
& &&\text{and } v \leq 0.00759 \text{ and } v \geq -0.004150 \text{ and } v \geq 0.00126822 \\
&+21.4738 &&\text{if } v \leq 0.00040989847620949166 \\
&-2.3676 &&\text{if } p \geq -0.5437554597854615 \text{ and } v \leq 0.0016861518379300846 \\
& &&\text{and } v \geq -0.0034104006830602885 \\
&+1.2316 &&\text{if } p \leq -0.1402401 \text{ and } p \geq -0.4494079709053038 \\
& &&\text{and } v \leq 0.023414026945829402 \text{ and } v \geq -0.01632060520350933
\end{aligned}
$$

No Action:

$$
\begin{aligned}
&-19.7248 &&\text{if } p \geq -0.5101138830184937 \text{ and } v \leq 0.006803702469915152 \\
& &&\text{and } v \geq -0.005573090445250273 \\
&-13.4848 &&\text{if } v \geq 0.0007145821116864693 \\
&+24.2199 &&\text{if } p \geq -0.4848527312278747 \text{ and } v \leq 0.00018940809495689726 \\
& &&\text{and } v \geq -0.009227151423692702 \\
&-23.3706 &&\text{if } v \leq 0.003292182506993414 \\
&-6.6878 &&\text{if } v \leq -9.685811237431741 \times 10^{-5} \\
&+24.7429 &&\text{if } p \leq -0.4022643387317657 \\
& &&\text{and } v \leq 0.00020952874911017827 \\
&-26.5154 &&\text{if } p \geq -0.6311534523963929 \text{ and } v \leq 0.006872396916151048 \\
& &&\text{and } v \geq -0.006245836243033409 \\
&-8.7515 &&\text{if } p \geq -0.39260063171386717 \text{ and } v \geq 0.0019062188919633629 \\
&+0.7214 &&\text{if } v \leq -0.006146392878144977 \\
&-0.9839 &&\text{if True} \\
&+6.8736 &&\text{if } p \geq -0.5437554597854615 \text{ and } v \leq 0.0038646903820335875 \\
&+1.0000 &&\text{if } v \geq -0.0010096890269778618
\end{aligned}
$$

Action Right:

| | |
|---|---|
| $+26.2150$ | if $p \leq -0.5101138830184937$ and $v \leq 0.006803702469915152$ |
| $+6.3636$ | if $p \geq -0.4848527312278747$ and $v \geq -0.009227151423692702$ |
| $+6.1251$ | if $p \geq -0.4855665504932403$ and $v \geq 0.003292182506993414$ |
| $+24.0837$ | if True |
| $+27.5176$ | if $p \leq -0.4022643387317657$ and $v \leq 0.0012711382005363714$ |
| $+21.3072$ | if $v \geq 0.0007119119516573855$ |
| $+18.6883$ | if $v \leq -0.0005520289996638883$ |
| $+19.7662$ | if $v \geq 0.0004626010428182804$ |
| $+21.3412$ | if $p \leq -0.6329279899597166$ |
| $+11.7750$ | if $p \geq -0.5767600297927856$ and $v \geq 0.0012682238826528204$ |
| $+16.5113$ | if $v \geq 0.00040989847620949166$ |
| $+1.3610$ | if $p \leq -0.4494079709053038$ and $p \geq -0.8744302272796631$ and $v \leq 0.023414026945829402$ |

# N   EXPLANATION OF CART-POLE RULES

Figure 15: Visual representation of the rules for CartPole involving position and velocity.

The CartPole environment consists of a pole mounted on a cart that moves along a track. The objective is to balance the pole upright by controlling the cart's movement left or right. The state space includes the cart's position($p$), velocity($v$), pole angle($\theta$), and angular velocity($\dot{\theta}$), while the action space includes applying force to move the cart in either direction. Figure 15 is the visualization of our actor's rule regarding the position and velocity. Please refer to Appendix O for all the resulted rules.

Here we will use a typical state in the Cart-Pole game, where the optimal action is to move the cart to the right, characterized by the parameters $p = -0.05, v = 0.1, \theta = 0.05$ and $\dot{\theta} = 0.3$, as an example to illustrate the decision-making process for our proposed method. In this scenario, pushing right reduces the pole's angular velocity, helping stabilize the pole by balancing the torque induced by the tilt. In our model, three rules become active when considering a leftward action; each rule incurs a negative score because the pole's angle is positive, decisively discouraging a left move. In contrast, the overall score for moving right is 2.3, notably higher than that for moving left. Although the car's slight leftward position imposes a small penalty, the pole's rightward angular velocity plays a critical role: one rule alone contributes +19.8 to the total, effectively countering any negative effects from the other rules. Consequently, given the pole's rightward tilt and velocity, the agent's optimal choice is to push right, thereby maintaining stability and preventing undesirable outcomes.

## O CARTPOLE RULES

Left:

$$
\begin{aligned}
-6.83310 \quad & \text{if pa} \geq -0.06048 \\
-6.82240 \quad & \text{if pa} \geq -0.05751 \\
+14.59180 \quad & \text{if pa} \leq 0.04701 \text{ and pav} \leq 0.26781 \\
+15.72630 \quad & \text{if pav} \leq -0.52576 \\
+3.47030 \quad & \text{if cp} \leq 0.36470 \text{ and cp} \geq 0.03103 \text{ and cv} \leq 0.20766 \text{ and cv} \geq -0.00962 \\
& \text{and pa} \geq -0.01513 \text{ and pav} \leq 0.28779 \text{ and pav} \geq -0.05232 \\
+0.88280 \quad & \text{if cp} \geq 0.24231 \text{ and cv} \geq -0.40319 \text{ and pa} \leq -0.03636 \text{ and pa} \geq -0.05565 \\
-7.36080 \quad & \text{if cp} \geq 0.36775 \text{ and cv} \geq -0.00425 \text{ and pa} \geq -0.03416 \text{ and pav} \leq 0.12249 \\
+11.59770 \quad & \text{if cp} \geq 0.25174 \text{ and pa} \leq 0.00381 \text{ and pa} \geq -0.02022 \text{ and pav} \leq 0.11855 \\
& \text{and pav} \geq -0.37647 \\
-7.77910 \quad & \text{if pa} \geq -0.02969 \\
-0.86900 \quad & \text{if cp} \leq 0.16311 \text{ and cp} \leq 0.30448 \text{ and cp} \geq -0.11618 \text{ and cv} \leq -0.01313 \\
& \text{and cv} \leq 0.17905 \text{ and pa} \leq 0.03930 \text{ and pa} \geq -0.02479 \text{ and pav} \leq 0.14835 \\
-3.02560 \quad & \text{if cp} \geq -0.42154 \text{ and cv} \leq 0.32329 \text{ and cv} \geq -0.01879 \text{ and pa} \geq -0.02018 \\
& \text{and pa} \geq 0.01293 \text{ and pav} \leq 0.13896 \text{ and pav} \geq -0.01362 \\
+7.09090 \quad & \text{if cp} \leq 0.21659 \text{ and cv} \leq 0.21317 \text{ and cv} \geq -0.00898 \text{ and pa} \leq -0.00355 \\
& \text{and pa} \geq -0.01990 \text{ and pav} \leq -0.29080
\end{aligned}
$$

Right:

$$
\begin{aligned}
-3.54700 \quad & \text{if cp} \leq 0.04146 \\
+19.18060 \quad & \text{if cp} \geq -0.07721 \text{ and pav} \geq 0.20059 \\
-3.77380 \quad & \text{if cp} \geq -0.12061 \\
+2.03760 \quad & \text{if cp} \geq 0.05832 \text{ and cv} \leq 0.06116 \text{ and pa} \geq -0.06911 \text{ and pav} \leq -0.05532 \\
-9.54210 \quad & \text{if cp} \leq 0.04726 \\
-3.07880 \quad & \text{if cp} \geq -0.01648 \text{ and cv} \leq 0.16053 \text{ and cv} \geq -0.19434 \text{ and pa} \leq -0.03636 \text{ and pa} \geq -0.05565 \\
+10.28650 \quad & \text{if cp} \geq 0.36775 \text{ and cv} \geq -0.00425 \text{ and cv} \geq 0.20400 \text{ and pa} \leq 0.00317 \\
& \text{and pa} \geq -0.03416 \text{ and pav} \leq 0.12249 \text{ and pav} \geq -0.37842 \\
+2.04960 \quad & \text{if cv} \geq -0.34930 \text{ and pa} \geq 0.00381 \text{ and pav} \leq 0.11855 \\
-5.77560 \quad & \text{if cv} \geq -0.36800 \text{ and pa} \leq -0.00339 \\
+5.11750 \quad & \text{if cp} \geq 0.40156 \text{ and cv} \leq 0.39100 \text{ and pa} \leq -0.00914 \text{ and pav} \leq -0.33129 \\
+0.00510 \quad & \text{if cp} \geq 0.02529 \text{ and cv} \geq -0.01879 \text{ and pa} \leq 0.03998 \text{ and pa} \geq 0.01293 \\
& \text{and pav} \geq 0.13896 \\
-4.80880 \quad & \text{if cp} \leq 0.21659 \text{ and cp} \geq 0.00944 \text{ and cv} \leq -0.28634 \text{ and pa} \leq -0.00355 \\
& \text{and pav} \leq 0.30114 \text{ and pav} \geq -0.01324
\end{aligned}
$$

## P    ACROBOT RULES

0:

$+8.7846$    if $w_1 \geq -0.09082$ and $w_2 \leq 0.22991$

$+9.3353$    if $\cos(\theta_2) \geq 0.89409$

$+10.4909$    if $\sin(\theta_2) \leq 0.44980$

$+8.2458$    if $\sin(\theta_2) \leq 0.37771$ and $w_1 \geq -0.12610$ and $w_2 \leq 0.15699$

$+11.6541$    if True

$+6.8474$    if $\sin(\theta_1) \leq 0.24220$ and $w_2 \leq 0.04877$

$+1.2850$    if $\cos(\theta_2) \geq 0.99115$ and $\sin(\theta_1) \leq -0.01332$

$+1.7239$    if $\sin(\theta_1) \leq -0.01472$ and $w_2 \leq 0.02869$

$+3.5813$    if $\cos(\theta 1) \geq 0.88816$ and $\sin(\theta_1) \geq -0.28723$ and $w_1 \leq 0.77240$

$+1.3335$    if $\cos(\theta_1) \geq 0.48252$ and $w_1 \leq 1.78991$ and $w_1 \geq -1.70488$

$+1.0000$    if $\cos(\theta_2) \leq 0.95137$ and $\sin(\theta_1) \geq 0.01719$

1:

$-3.0189$    if $\cos(\theta_2) \leq 0.99319$

$-3.3464$    if $\cos(\theta_1) \leq 0.99048$ and $w_2 \leq 0.12677$

$-8.6917$    if $\cos(\theta_1) \geq 0.81787$ and $w_1 \leq 0.92265$

$-5.4697$    if True

$-9.3263$    if $w_1 \leq 0.21755$ and $w_2 \geq -0.10354$

$-5.5746$    if $\sin(\theta_2) \leq 0.37771$ and $w_1 \leq 1.28679$ and $w_2 \geq 0.15699$

$+5.4709$    if $\sin(\theta_1) \geq -0.26647$ and $\sin(\theta_2) \leq 0.44500$ and $w_1 \geq -1.07104$ and $w_2 \leq 0.04877$

$-8.2010$    if $\sin(\theta_1) \leq 0.04400$

$+1.5784$    if $\cos(\theta_1) \geq 0.99341$

$-9.7897$    if $\sin(\theta_1) \leq 0.15513$ and $\sin(\theta_1) \geq -0.70608$ and $\sin(\theta_2) \geq -0.17949$
and $w_1 \leq 1.78991$ and $w_1 \geq -1.70488$

$-1.2848$    if $\cos(\theta_1) \geq 0.99674$

$+1.0000$    if $\cos(\theta_2) \leq 0.99350$ and $\sin(\theta_1) \leq -0.21224$ and $w_1 \geq 0.11238$
and $w_2 \leq -0.22888$

2:

$+3.5980$    if $\cos(\theta_1) \geq 0.99689$

$+2.5385$    if $\cos(\theta_1) \geq 0.99048$

$+16.7525$    if True

$-3.7130$    if $\cos(\theta_1) \leq 0.99999$ and $\sin(\theta_2) \leq -0.01789$

$-1.9920$    if $\sin(\theta_2) \leq -0.00190$

$+12.0937$    if $w_1 \leq -0.02180$ and $w_2 \geq 0.04994$

$+9.6534$    if $w_2 \geq 0.04994$

$+1.1957$    if $\cos(\theta_2) \geq 0.24062$ and $w_1 \geq -0.32756$ and $w_2 \leq 1.64803$

$-3.6820$    if $w_1 \geq -0.12540$

$+15.8530$    if $\cos(\theta_1) \geq 0.88816$ and $w_2 \geq -0.41702$

$+10.0707$    if $\cos(\theta_1) \leq 0.99674$ and $\cos(\theta_2) \geq 0.92457$

$+1.6685$    if $\cos(\theta_2) \leq 0.99350$ and $\sin(\theta_1) \leq -0.21224$ and $w_1 \geq 0.11238$
and $w_2 \leq -0.22888$

## Q   INTERPRETABILITY OF TREE-BASED AND RULE-BASED POLICIES

Following the widely cited definition of interpretability by Murdoch et al. (2019), an interpretable model should satisfy *simulatability*, *modularity*, and *low complexity*. Simulatability means that a human can easily compute the model's output from its explicit expression, given an input and the model parameters. Modularity means that individual components of the model can be understood in isolation. Complexity can be quantified by, for example, the total number of terms or parameters in the model.

SYMPOL learns an axis-aligned decision tree policy directly with policy gradients in a standard on-policy RL setting, yielding an interpretable tree-based policy class. Conceptually, our method plays a similar role but uses a rule-based policy class instead of trees. While trees and rules are closely related—each root-to-leaf path can be seen as a rule—they differ in how they are used and in the kind of interpretability they afford.

From the perspective of interpreting individual decisions, both trees and rule sets provide simulatability. For a given state, a tree explains its choice via a single path of the form:

if (condition A at node 1)$\wedge$(condition B at node 2)$\wedge\cdots\wedge$(condition M at node M) $\Rightarrow$ take action LEFT.

A rule-based model explains the same decision by indicating which rules fired and how they were combined, for example: *"Rule 1 and Rule 7 are activated and jointly give the highest weight; therefore the policy selects LEFT."*

In terms of complexity, rule-based models tend to be slightly easier to keep compact: we can directly control the number and length of rules. Tree-based policies, by contrast, often grow deep or wide as task complexity increases, which can quickly erode practical interpretability even if the representation remains transparent in principle.

For modularity, the difference is more pronounced. In a tree, one can treat each root-to-leaf path as a "module" (a rule), but internal nodes are shared across many paths, so modifying a single split typically affects a large portion of the state space. In a rule set, each rule is a self-contained piece of logic. This makes the rule-based policy highly modular: individual rules can be inspected, added, or removed without having to mentally re-parse or globally restructure the entire model.

This modularity also makes certain safety-style queries much easier to answer with rules than with trees. For example, consider the property: *"Is there any situation where the policy chooses RIGHT while $|\theta| > 0.25$ and $x > 1.2$? If so, list those situations."* In a rule-based policy, we can simply scan for rules whose antecedent includes $|\theta| > 0.25$ and $x > 1.2$ and whose consequent is action $=$ RIGHT. If such a rule exists (say, Rule V1), we can immediately say: *"Yes, a violation occurs whenever Rule V1 fires."* If no such rule exists, we can cleanly state: *"The rule set guarantees the safety property: in no rule does $|\theta| > 0.25 \wedge x > 1.2$ appear with action $=$ RIGHT."* For an equivalent tree policy, answering the same question requires an exhaustive traversal and aggregation of all relevant paths, which is considerably less direct in practice.

## R   PONG ENV

### R.1   RULES

**Do nothing**

| Aspect | SYMPOL (tree-based policy) | Rule-based policy (ours) |
|---|---|---|
| **Policy class** | Axis-aligned decision tree policy; each internal node is a threshold on one feature, and leaves output action logits. | Explicit rule set / rule list; each rule is a conjunction of conditions $\Rightarrow$ action (optionally with weights or priorities). |
| **Simulatability (single decision)** | High: for one state, follow a single root-to-leaf path $\Rightarrow$ clear "if $A \wedge B \wedge \ldots$ then action." | High: for one state, see which rule(s) fired and how they combine (e.g., "Rule 1 + Rule 7 vote for LEFT"). |
| **Global structure** | Clear hierarchy of decisions: top splits show the most salient features. | Global picture is less rigid; rules form a flat or lightly structured set, and regimes are inferred by grouping similar rules. |
| **Modularity** | Each path can be read as a rule, but internal nodes are shared: changing a split affects many paths at once. | Highly modular: each rule is a self-contained unit. One can inspect, add, or remove rules without restructuring the whole model. |
| **Complexity control** | Control via depth limits, pruning, and tree size; trees can still grow large and become hard to read. | Control via the number of rules and rule length; it is easier to set an explicit rule budget and keep individual rules short. |

Table 3: Comparison between the SYMPOL tree-based policy and our rule-based policy from an interpretability perspective (simulatability, modularity, and complexity).

$+\,0.9642$ if $v_{x,b} \geq 0.1$

$+\,0.0053$ if $v_{x,b} \leq -0.1 \wedge v_{y,b} \geq -0.1$

$+\,0.9613$ if $x_b \leq 0.6187499761581421$

$-\,4.2170$ if $y_b \leq 0.6523809432983398 \wedge \mathbf{y}_e \geq 0.5476190447807312$

$-\,3.1880$ if $v_{y,b} \geq 0.1 \wedge y_b \leq 0.6523809432983398$

$+\,4.1575$ if $v_{y,b} \leq -0.2$

$+\,0.9695$ if $bf \geq 0.4571428596973419$

$-\,3.1775$ if $v_{y,b} \geq 0.1 \wedge \mathbf{y}_e \leq 0.5476190447807312$

$-\,0.0000$ if $x_b \geq 0.800000011920929$

$-\,3.1775$ if $v_{y,b} \geq 0.1 \wedge x_b \geq 0.48750001192092896 \wedge \mathbf{y}_e \leq 0.5476190447807312$

$-\,4.2170$ if $x_b \leq 0.6187499761581421 \wedge \mathbf{y}_e \leq 0.6428571343421936 \wedge \mathbf{y}_e \geq 0.5476190447807312$

$+\,0.9693$ if $x_b \leq 0.7875000238418579$

$+\,0.0005$ if $x_b \geq 0.7875000238418579$

$+\,3.3783$ if $x_b \geq 0.48750001192092896$

$+\,0.9695$ if $\mathbf{y}_e \leq 0.6428571343421936$

$+\,0.9695$ if $\mathbf{y}_e \leq 0.6523809432983398$

$-\,0.1303$ if $bf \leq 0.8999999761581421 \wedge v_{x,b} \geq 0.1$

$-\,0.0000$ if $\mathbf{y}_e \geq 0.7809523940086365$

$-\,0.0000$ if $bf \leq 0.30000001192092896 \wedge \mathbf{y}_e \geq 0.9095237851142883 \wedge y_p \leq 0.30000001192092896$

$-\,4.4359$ if $v_p \geq 0.30000001192092896$

## Up

$-1.0691$    if true

$+9.5734$    if $v_{x,b} \leq -0.1$

$+9.7684$    if $v_{x,b} \leq -0.1 \wedge v_{y,b} \geq -0.1 \wedge y_b \leq 0.738095223903656$

$-3.6835$    if $v_{x,b} \geq -0.1 \wedge v_{y,b} \geq -0.1$

$+1.8689$    if $x_b \leq 0.800000011920929$

$+6.6212$    if $x_b \geq 0.800000011920929 \wedge \mathbf{y}_e \geq 0.5476190447807312$

$+9.8937$    if $v_{x,b} \leq -0.1 \wedge y_b \leq 0.7809523940086365$

$+3.2889$    if $v_p \leq 0.0 \wedge x_b \geq 0.800000011920929 \wedge \mathbf{y}_e \geq 0.5476190447807312$

$+2.1446$    if $x_b \leq 0.793749988079071$

$-2.7148$    if $v_{x,b} \geq 0.1 \wedge v_{y,b} \leq 0.1 \wedge x_b \leq 0.6187499761581421 \wedge \mathbf{y}_e \leq 0.776190459728241$

$-10.3563$    if $v_{x,b} \geq 0.1 \wedge y_b \geq 0.5476190447807312 \wedge \mathbf{y}_e \geq 0.5476190447807312$

$+2.2118$    if $x_b \geq 0.6187499761581421 \wedge \mathbf{y}_e \leq 0.776190459728241 \wedge \mathbf{y}_e \geq 0.776190459728241$

$+1.1660$    if $v_{y,b} \leq -0.2 \wedge y_b \leq 0.7809523940086365$

$+9.0361$    if $x_b \leq 0.800000011920929 \wedge x_b \geq 0.48750001192092896 \wedge y_b \geq 0.5476190447807312$

$-10.8639$    if $v_{x,b} \geq 0.1 \wedge v_{y,b} \geq 0.1 \wedge y_b \leq 0.7809523940086365$

$+6.7201$    if $x_b \leq 0.800000011920929 \wedge y_b \geq 0.7809523940086365 \wedge \mathbf{y}_e \geq 0.776190459728241$

$+1.9128$    if $bf \geq 0.4571428596973419 \wedge x_b \geq 0.48750001192092896$

$+12.6463$    if $\mathbf{y}_e \geq 0.9095237851142883$

$+0.8748$    if $bf \geq 0.12857143580913544 \wedge v_p \leq -0.6000000238418579 \wedge y_b \geq 0.7809523940086365$

$+1.1194$    if $v_p \leq 0.30000001192092896 \wedge v_{x,b} \leq -0.1 \wedge y_b \leq 0.6523809432983398$

## Down

$-0.1153$    if true

$+0.2371$    if $v_{x,b} \geq 0.1$

$-1.1869$    if $v_{y,b} \leq 0.1 \wedge \mathbf{y}_e \geq 0.6428571343421936$

$+6.6905$    if $x_b \leq 0.800000011920929$

$-1.6155$    if $v_p \leq 0.20000000298023224 \wedge v_{y,b} \leq 0.1 \wedge v_{y,b} \geq 0.1$

$-7.3399$    if $x_b \geq 0.6187499761581421 \wedge y_b \geq 0.6523809432983398 \wedge \mathbf{y}_e \geq 0.6428571343421936$

$-1.0679$    if $v_p \leq 0.0 \wedge x_b \leq 0.6187499761581421 \wedge \mathbf{y}_e \geq 0.5476190447807312$

$-0.1710$    if $v_{y,b} \leq -0.2$

$-1.6300$    if $y_b \geq 0.5476190447807312 \wedge \mathbf{y}_e \leq 0.5476190447807312$

$-0.0564$    if $v_{y,b} \geq 0.1 \wedge x_b \leq 0.6187499761581421 \wedge \mathbf{y}_e \geq 0.6428571343421936$

$-0.2991$    if $\mathbf{y}_e \geq 0.9095237851142883$

$-6.6643$    if $v_p \geq -0.30000001192092896 \wedge \mathbf{y}_e \geq 0.6428571343421936$

$+6.4798$    if $v_{x,b} \geq 0.1 \wedge x_b \leq 0.800000011920929 \wedge x_b \geq 0.48750001192092896$

$-0.6501$    if $v_p \leq 0.10000000149011612 \wedge x_b \geq 0.6187499761581421 \wedge y_b \geq 0.7809523940086365$

$-0.7654$    if $x_b \geq 0.6187499761581421 \wedge \mathbf{y}_e \geq 0.776190459728241$

$-0.0399$    if $v_{y,b} \leq -0.2 \wedge \mathbf{y}_e \geq 0.776190459728241$

$-0.0802$    if $v_{y,b} \leq -0.2 \wedge \mathbf{y}_e \geq 0.6428571343421936$

$-1.2114$    if $v_{y,b} \leq 0.1 \wedge x_b \geq 0.48750001192092896 \wedge \mathbf{y}_e \geq 0.6428571343421936$

$-0.0482$    if $x_b \geq 0.9750000238418579 \wedge y_p \leq 0.12857143580913544$

$+1.0000$    if $x_b \leq 0.6187499761581421 \wedge y_b \leq 0.7809523940086365$

**Feature importance analysis.** To diagnose why this particular rule ensemble performs poorly, we compute a simple global importance score for each feature by summing the absolute values of all rule coefficients in which it appears. This tells us which variables the actor relies on most when selecting actions. For the rules in previous section, we obtain:

$$\text{ball } x\text{-position } (x_b)\colon \ 89.18, \quad \textbf{opponent } y\text{-position } (\mathbf{y}_e)\colon \ 88.85,$$

$$\text{ball } y\text{-position } (y_b)\colon \ 77.05, \quad \text{ball } x\text{-velocity } (v_{x,b})\colon \ 65.8, \quad \text{ball } y\text{-velocity } (v_{y,b})\colon \ 47.9,$$

$$\text{paddle velocity } (v_p)\colon \ 19.7, \quad \text{block fraction } (bf)\colon \ 3.9, \quad \text{paddle } y\text{-position } (y_p)\colon \ 0.05.$$

These numbers show that the policy is dominated by the geometry of the ball and opponent, with the ball's horizontal position $x_b$ and the opponent's vertical position $\mathbf{y}_e$ emerging as the most influential features. In contrast, important control variables such as the paddle velocity $v_p$ and especially $y_p$ contribute much less overall. Although focusing on ball–opponent geometry is not unreasonable, closer inspection shows that many of the largest coefficients involving $\mathbf{y}_e$ appear in rules that trigger clearly suboptimal actions. In other words, the ensemble has learned *strong but misaligned* dependencies on $\mathbf{y}_e$, effectively overfitting to spurious thresholds on the opponent's position. This explains why the resulting policy is both interpretable and systematically wrong: it selects actions for transparent, but ultimately bad, reasons.

# S  ADDITIONAL EXPERIMENTS ON OCATARI

## S.1  PERFORMANCE

We also evaluate NSAC on the OCAtari-Breakout environment, a low-dimensional variant of the classic Atari *Breakout* game exposed through the OCAtari Delfosse et al. (2023a). The agent observes summary features such as the ball position (`ball_y`, `rel_x`), ball velocities (`vx`, `vy`), paddle position (`paddle_edge`), and brick-related indicators (`brick_prev`, `brick_cur`), and must choose between three discrete actions: *Fire/Do nothing*, *Right*, and *Left*. Performance in this environment is reported as the total game score, averaged over 5 runs with different random seeds for each method. Among all baselines, PPO achieves the highest overall score on OCAtari-Breakout. Nonetheless, NSAC also attains strong performance, with an average score above 300 over 5 random seeds, making it competitive with neural baselines while clearly outperforming all other symbolic methods. This demonstrates that NSAC can retain much of the raw performance of powerful deep RL algorithms, while still yielding an interpretable rule ensemble.

| Environment | Q-table | DQN | A2C | PPO | SDSAC | SACBBF | Rainbow | SYMPOL | $\pi$affine-D | D-SDT | NSAC |
|---|---|---|---|---|---|---|---|---|---|---|---|
| Breakout | 41.23± 14.20 | 127.23± 25.78 | 344.23± 31.78 | **389.21** ± 14.33 | 342.00± 39.62 | 267.00± 27.12 | 235.76± 31.37 | 227.00± 35.76 | 23.86 ± 14.76 | 34.26 ± 7.86 | 312.76± 31.60 |

Table 4: Performance comparison on OCAtari-Breakout.

## S.2  REPRESENTATIVE RULES AND THEIR INTERPRETATION.

To illustrate the kind of behaviors captured by the learned rule ensemble, we highlight two representative rules, the detailed rules are listed in the following section.

**Rule 1 (Action 1: Fire / Do nothing).** One of the highest-weight rules for the "do nothing" action is

$$+6.1152 \quad \text{if } 0.5000 \le \texttt{ball\_y\_cur} \le 0.7714, \ \texttt{rel\_x\_cur} \le -0.5375,$$
$$\texttt{vy\_prev} \le -0.0190, \ \texttt{vy\_cur} \ge -0.0381.$$

Here, `ball_y_cur` in $[0.5, 0.77]$ indicates that the ball is still relatively high on the screen, while `rel_x_cur` $\le -0.5375$ means it is clearly to the left of the paddle. The vertical velocities `vy_prev` $\le -0.0190$ and `vy_cur` $\ge -0.0381$ show that the ball is moving downwards, but not extremely fast. In this regime, the rule strongly favors *not* moving the paddle: it is too early and too uncertain to chase the ball aggressively, and keeping the paddle near a default position preserves flexibility for a later, more accurate intercept. This encodes an intuitive "do not overreact too early" timing heuristic.

**Rule 2 (Action 3: Left).** A strongly negative rule for the "move left" action is

$$-5.7469 \quad \text{if } 0.4143 \le \texttt{ball\_y\_cur} \le 0.8381, \ \texttt{rel\_x\_prev} \le -0.0063,$$
$$-0.0500 \le \texttt{vx\_prev} \le -0.0375, \ \texttt{vy\_prev} \le -0.0190, \ \texttt{vy\_cur} \le 0.0286.$$

In this situation, the ball is in the middle-to-upper region of the screen, already to the left of the paddle (`rel_x_prev` $\le -0.0063$), moving leftwards (`vx_prev` $< 0$) and downwards (`vy_prev` $< 0$). The large negative coefficient indicates that moving the paddle further left in this configuration is strongly discouraged. Intuitively, shifting left would risk moving away from the eventual intercept point or overshooting it; instead, the policy prefers to stay more central or delay horizontal motion until the ball is lower and the crossing point is clearer. This rule thus captures a natural "do not move away from an incoming ball" behavior.

## S.3 RULES

**Action 1: Fire / Do nothing**

$-2.6037$  if `paddle_edge_prev` $\geq 0.1375$

$-4.0445$  if `vy_prev` $\geq 0.0190$ and `vy_cur` $\geq -0.0190$

$+2.6453$  if `ball_y_prev` $\leq 0.7714$ and `ball_y_cur` $\leq 0.7714$ and `rel_x_cur` $\leq -0.2000$
    and $-0.0500 \leq$ `vx_prev` $\leq 0.0500$ and $-0.0286 \leq$ `vy_cur` $\leq 0.0381$

$+5.9613$  if `ball_above_prev` $\geq 1.0000$ and `ball_y_cur` $\leq 0.8381$ and `rel_x_prev` $\leq -0.0063$
    and $-0.0500 \leq$ `vx_cur` $\leq 0.0500$ and `vy_prev` $\geq -0.0190$ and `vy_cur` $\leq 0.0048$

$-1.9160$  if `ball_y_prev` $\leq 0.7714$ and `rel_x_cur` $\geq -0.4125$ and `vy_prev` $\leq 0.0381$ and `vy_cur` $\leq -0.0190$

$-2.3335$  if `rel_x_cur` $\geq -0.1000$ and `vy_prev` $\leq 0.0381$

$+3.6887$  if `ball_y_prev` $\leq 0.7714$ and `ball_y_cur` $\leq 0.7048$ and `rel_x_prev` $\leq -0.1000$
    and `vx_prev` $\geq -0.0375$ and `vy_prev` $\geq -0.0381$ and `vy_cur` $\leq 0.0190$

$+2.4687$  if `ball_y_prev` $\geq 0.6381$ and `vy_prev` $\leq -0.0286$

$+3.8568$  if `ball_y_prev` $\leq 0.7714$ and `rel_x_cur` $\leq -0.0063$ and `vx_prev` $\geq -0.0500$
    and `vx_cur` $\geq -0.0500$ and `vy_prev` $\leq 0.0048$ and `vy_cur` $\geq -0.0190$

$-2.7193$  if `ball_y_cur` $\leq 0.7048$ and `brick_cur` $\geq 0.0900$ and `vy_prev` $\leq 0.0381$ and `vy_cur` $\leq 0.0048$

$+1.9062$  if `ball_y_prev` $\leq 0.5000$ and `ball_y_cur` $\leq 0.5000$ and `brick_prev` $\geq 0.0900$
    and `brick_cur` $\geq 0.0900$ and `rel_x_prev` $\leq -0.0063$ and `vx_prev` $\geq 0.0500$ and `vx_cur` $\geq 0.0500$

$-0.8540$  if `ball_y_prev` $\leq 0.8381$ and `rel_x_prev` $\leq -0.1000$ and
    $0.0375 \leq$ `vx_prev` $\leq 0.0500$ and `vy_prev` $\leq 0.0190$

$-2.8242$  if `ball_y_cur` $\leq 0.4143$ and `vy_prev` $\leq 0.0048$

$+2.4663$  if `ball_y_prev` $\leq 0.8381$ and `vx_prev` $= -0.0500$ and `vy_prev` $\leq -0.0190$
    and $-0.0381 \leq$ `vy_cur` $\leq 0.0381$

$-1.5553$  if `ball_y_prev` $\leq 0.6381$ and `vx_prev` $\leq -0.0375$

$-1.8001$  if `ball_y_cur` $\geq 0.6381$ and `vx_cur` $\geq -0.0375$ and `vy_cur` $\geq -0.0286$

$+6.1152$  if $0.5000 \leq$ `ball_y_cur` $\leq 0.7714$ and `rel_x_cur` $\leq -0.5375$
    and `vy_prev` $\leq -0.0190$ and `vy_cur` $\geq -0.0381$

$-3.4766$  if `ball_y_prev` $\geq 0.4143$ and $0.5000 \leq$ `ball_y_cur` $\leq 0.7714$
    and $0.0000 \leq$ `vx_prev` $\leq 0.0500$ and `vx_cur` $\leq 0.0500$ and `vy_prev` $\leq 0.0381$ and `vy_cur` $\geq -0.0381$

$-1.4946$  if `rel_x_prev` $\leq -0.6688$ and `rel_x_cur` $\leq -0.6688$

$-0.7953$  if $0.4143 \leq$ `ball_y_cur` $\leq 0.8381$ and `brick_cur` $\leq 0.1100$ and `rel_x_prev` $\leq -0.1000$
    and `rel_x_cur` $\leq -0.1000$ and `vx_prev` $\geq -0.0375$ and `vy_prev` $\leq 0.0381$ and `vy_cur` $\geq -0.0381$

$-1.2368$  if $0.4143 \leq$ `ball_y_cur` $\leq 0.7714$ and `paddle_edge_prev` $\geq 0.1375$
    and `rel_x_prev` $\geq -0.6688$ and `vx_prev` $\leq 0.0500$ and `vx_cur` $\leq 0.0500$

$+2.5430$  if `ball_y_prev` $\geq 0.5000$ and $0.5000 \leq$ `ball_y_cur` $\leq 0.8381$ and `rel_x_prev` $\leq -0.2000$
    and `vx_prev` $\leq 0.0000$ and `vx_cur` $\geq -0.0500$ and $-0.0381 \leq$ `vy_prev` $\leq 0.0048$

$+0.7993$  if `ball_above_prev` $\geq 1.0000$ and `ball_above_cur` $\geq 1.0000$ and `rel_x_prev` $\geq -0.5375$
    and $0.0000 \leq$ `vx_prev` $\leq 0.0500$ and `vx_cur` $\geq -0.0375$ and `vy_prev` $\leq 0.0286$

$+1.7013$  if `ball_above_prev` $\geq 1.0000$ and `ball_y_cur` $\geq 0.6381$ and `rel_x_cur` $\geq -0.2000$ and
    `vy_prev` $\leq 0.0048$

$-1.1456$  if `rel_x_prev` $\geq -0.3063$ and $-0.3063 \leq$ `rel_x_cur` $\leq -0.0063$
    and `vy_prev` $\leq 0.0048$ and `vy_cur` $\geq -0.0381$

**Action 2: Right**

$-3.7358$    if `ball_y_cur` $\leq 0.8381$

$+3.3079$    if `rel_x_cur` $\geq -0.5375$ and `vy_prev` $\geq 0.0190$ and `vy_cur` $\geq 0.0048$

$-9.9162$    if `ball_above_cur` $\geq 1.0000$ and `vx_prev` $\leq 0.0500$ and `vy_cur` $\leq 0.0381$

$+4.6580$    if `ball_above_prev` $\geq 1.0000$ and `ball_y_cur` $\geq 0.7048$
       and `vy_prev` $\geq 0.0190$ and `vy_cur` $\geq -0.0190$

$+2.7250$    if `paddle_edge_prev` $\geq 0.1375$ and `vx_prev` $\geq 0.0000$ and `vx_cur` $\geq -0.0375$
       and `vy_prev` $\geq -0.0190$ and `vy_cur` $\geq -0.0381$

$-1.5765$    if `ball_y_cur` $\leq 0.7714$ and `vy_prev` $\geq 0.0190$

$+4.5538$    if `ball_y_prev` $\geq 0.5714$ and `rel_x_prev` $\geq -0.6688$ and `vx_prev` $\leq 0.0500$
       and `vy_prev` $\geq 0.0286$ and `vy_cur` $\geq -0.0381$

$+3.0658$    if `rel_x_cur` $\geq -0.3063$ and $-0.0375 \leq$ `vx_prev` $\leq 0.0500$
       and `vy_prev` $\geq -0.0190$ and `vy_cur` $\geq 0.0048$

$+2.8135$    if `ball_above_prev` $\geq 1.0000$ and `rel_x_cur` $\leq -0.0063$
       and `vx_prev` $\geq -0.0500$ and `vy_prev` $\geq 0.0381$ and `vy_cur` $\geq 0.0190$

$+2.8650$    if `ball_above_prev` $\geq 1.0000$ and `ball_y_prev` $\geq 0.7714$
       and `rel_x_cur` $\geq -0.6688$ and `vy_prev` $\geq 0.0190$ and `vy_cur` $\geq -0.0381$

$-4.0301$    if `ball_y_prev` $\leq 0.8381$ and `rel_x_cur` $\leq -0.4125$

$-6.5868$    if `ball_above_cur` $\geq 1.0000$ and `vy_prev` $\leq 0.0381$ and `vy_cur` $\leq -0.0190$

$-2.9138$    if `ball_y_prev` $\leq 0.7048$ and `rel_x_prev` $\leq -0.0063$ and `rel_x_cur` $\leq -0.0063$
       and `vx_cur` $\leq 0.0375$ and `vy_prev` $\geq 0.0048$

$-2.0466$    if `ball_above_cur` $\geq 1.0000$ and `ball_y_prev` $\leq 0.8381$ and `rel_x_cur` $\leq -0.2000$
       and `vx_cur` $\leq 0.0000$ and `vy_prev` $\geq 0.0048$

$+3.3836$    if `ball_y_cur` $\geq 0.7714$ and `rel_x_prev` $\leq -0.1000$
       and `vx_cur` $\geq -0.0375$ and `vy_cur` $\geq 0.0048$

$-1.8377$    if `ball_above_cur` $\geq 1.0000$
       and $-0.0375 \leq$ `vx_prev` $\leq 0.0500$ and $-0.0375 \leq$ `vx_cur` $\leq 0.0000$ and `vy_prev` $\geq 0.0048$

$+3.7873$    if `ball_y_cur` $\geq 0.8381$ and $0.0190 \leq$ `vy_cur` $\leq 0.0381$

$-2.0489$    if `ball_above_cur` $\geq 1.0000$ and `rel_x_prev` $\leq -0.5375$ and `vx_cur` $\leq 0.0375$

$-2.7407$    if `ball_y_prev` $\leq 0.7714$ and `ball_y_cur` $\geq 0.4143$ and `rel_x_prev` $\geq -0.6688$
       and `vx_prev` $\leq 0.0500$ and `vy_prev` $\geq 0.0190$

$-2.5147$    if `ball_y_cur` $\leq 0.5714$ and `brick_prev` $\geq 0.1000$ and `rel_x_prev` $\geq -0.1000$

$-1.7701$    if `ball_above_cur` $\geq 1.0000$ and `brick_cur` $\geq 0.0900$
       and `vy_prev` $\leq 0.0286$ and `vy_cur` $\geq 0.0190$

$+1.5424$    if `ball_y_prev` $\geq 0.5000$ and `ball_y_cur` $\geq 0.6381$ and `brick_prev` $\geq 0.0900$
       and `rel_x_prev` $\geq -0.2000$ and `vx_cur` $\leq 0.0375$ and `vy_prev` $\geq 0.0190$ and `vy_cur` $\geq -0.0381$

$+1.4835$    if `ball_y_cur` $\geq 0.4143$ and `brick_prev` $\geq 0.0900$ and `rel_x_prev` $\geq -0.6688$
       and `rel_x_cur` $\leq -0.1000$ and `vx_cur` $\geq 0.0000$ and `vy_prev` $\geq 0.0048$ and `vy_cur` $\geq -0.0286$

$+1.8123$    if `ball_y_prev` $\geq 0.7048$ and `brick_cur` $\leq 0.1300$ and `vx_cur` $\leq 0.0000$
       and `vy_prev` $\geq 0.0381$ and `vy_cur` $\geq -0.0381$

$+1.4038$    if `rel_x_prev` $\geq -0.3063$ and `vx_cur` $\geq 0.0000$
       and `vy_prev` $\geq 0.0048$ and `vy_cur` $\geq -0.0381$

**Action 3: Left**

$+3.2914$ if `ball_y_cur` $\leq 0.8381$ and `paddle_edge_prev` $\geq 0.1375$

$-5.9441$ if `ball_y_cur` $\leq 0.7714$ and `rel_x_prev` $\leq -0.0063$ and `rel_x_cur` $\leq -0.1000$
and $-0.0500 \leq$ `vx_prev` $\leq 0.0500$ and `vy_prev` $\leq 0.0048$ and $-0.0286 \leq$ `vy_cur` $\leq 0.0381$

$-3.0919$ if `ball_y_cur` $\geq 0.7714$ and `rel_x_cur` $\geq -0.6688$ and `vy_cur` $\geq 0.0048$

$+3.1300$ if `ball_above_cur` $\geq 1.0000$ and `paddle_edge_prev` $\geq 0.1375$ and `vx_prev` $\leq 0.0500$
and `vy_prev` $\leq 0.0381$ and `vy_cur` $\leq 0.0381$

$-2.7873$ if `ball_above_prev` $\geq 1.0000$ and `ball_y_cur` $\geq 0.7048$ and `rel_x_prev` $\geq -0.3063$

$-2.4206$ if `ball_above_prev` $\geq 1.0000$ and `rel_x_prev` $\leq -0.3063$ and `rel_x_cur` $\leq -0.3063$
and `vx_cur` $\leq 0.0500$ and `vy_prev` $\geq -0.0381$ and `vy_cur` $\leq 0.0048$

$-1.4360$ if `rel_x_prev` $\geq -0.6688$ and `vx_prev` $\geq -0.0375$
and $-0.0375 \leq$ `vx_cur` $\leq 0.0500$ and `vy_prev` $\geq 0.0286$

$-5.7469$ if $0.4143 \leq$ `ball_y_cur` $\leq 0.8381$ and `rel_x_prev` $\leq -0.0063$
and $-0.0500 \leq$ `vx_prev` $\leq -0.0375$ and `vy_prev` $\leq -0.0190$ and `vy_cur` $\leq 0.0286$

$-3.8718$ if `vx_prev` $\geq 0.0000$ and `vx_cur` $\geq 0.0000$ and `vy_cur` $\geq 0.0048$

$-3.4132$ if `ball_y_cur` $\geq 0.5714$ and `vx_cur` $\geq -0.0375$ and `vy_prev` $\leq -0.0381$

$-1.9780$ if `vx_prev` $\leq 0.0000$ and $-0.0500 \leq$ `vx_cur` $\leq 0.0000$ and `vy_prev` $\geq -0.0286$

$+3.7481$ if `ball_above_cur` $\geq 1.0000$ and `ball_y_prev` $\geq 0.4143$ and `ball_y_cur` $\geq 0.5000$
and `vx_prev` $\leq 0.0500$ and `vy_prev` $\geq -0.0190$ and $-0.0381 \leq$ `vy_cur` $\leq 0.0286$

$+1.8542$ if `ball_y_cur` $\leq 0.7714$ and `paddle_edge_prev` $\geq 0.1375$ and `rel_x_cur` $\geq -0.5375$
and `vx_prev` $\leq 0.0500$ and `vx_cur` $\leq 0.0500$ and `vy_prev` $\leq 0.0381$

$+0.6150$ if `ball_above_cur` $\geq 1.0000$ and `rel_x_prev` $\leq -0.4125$ and `rel_x_cur` $\leq -0.4125$
and `vy_prev` $\geq -0.0381$ and `vy_cur` $\geq -0.0286$

$+2.3428$ if `rel_x_prev` $\geq -0.5375$ and `rel_x_cur` $\geq -0.5375$
and $-0.0375 \leq$ `vx_cur` $\leq 0.0375$ and `vy_cur` $\leq -0.0190$

$-1.2518$ if `ball_above_prev` $\geq 1.0000$ and `rel_x_prev` $\leq -0.1000$
and $-0.5375 \leq$ `rel_x_cur` $\leq -0.1000$ and $-0.0500 \leq$ `vx_cur` $\leq 0.0500$
and `vy_prev` $\geq -0.0286$ and `vy_cur` $\leq 0.0286$

$-2.2239$ if `ball_y_prev` $\leq 0.7714$ and `ball_y_cur` $\leq 0.7048$
and `vx_prev` $\geq -0.0375$ and `vy_prev` $\geq -0.0190$ and `vy_cur` $\leq 0.0190$

$-0.8174$ if `ball_above_prev` $\geq 1.0000$ and `ball_y_cur` $\geq 0.6381$
and `vx_prev` $\leq 0.0375$ and `vy_prev` $\geq -0.0381$

$-0.6264$ if `ball_above_prev` $\geq 1.0000$ and `rel_x_prev` $\geq -0.5375$ and `rel_x_cur` $\geq -0.5375$
and `vx_prev` $\geq 0.0000$ and `vx_cur` $\geq -0.0500$

$+2.5593$ if `ball_y_prev` $\leq 0.7714$ and `rel_x_cur` $\leq -0.3063$
and `vy_prev` $\geq 0.0381$ and `vy_cur` $\geq -0.0190$

$+0.9760$ if `ball_above_cur` $\geq 1.0000$ and `ball_y_cur` $\geq 0.4143$
and $-0.3063 \leq$ `rel_x_prev` $\leq -0.0063$ and `rel_x_cur` $\leq -0.1000$
and `vy_prev` $\leq 0.0381$ and $-0.0381 \leq$ `vy_cur` $\leq 0.0381$

$+1.6089$ if `ball_y_cur` $\leq 0.6381$ and `rel_x_prev` $\geq -0.2000$ and `rel_x_cur` $\geq -0.2000$

$-0.7929$ if `rel_x_cur` $\geq -0.5375$ and `vx_prev` $\geq -0.0500$ and `vy_cur` $\geq -0.0190$

$-0.5734$ if `ball_y_prev` $\geq 0.5000$ and `brick_cur` $\leq 0.0900$ and `vx_prev` $\geq -0.0500$
and $-0.0500 \leq$ `vx_cur` $\leq 0.0375$ and `vy_cur` $\geq -0.0381$

$-1.0272$ if `ball_y_prev` $\geq 0.8381$ and `ball_y_cur` $\geq 0.7714$ and `vy_prev` $\leq 0.0381$

## T    STATEMENT ON LLM USAGE

Large language models (LLMs), such as ChatGPT, were used solely for editorial assistance in this work. Their role was limited to improving grammar, rephrasing sentences, and enhancing clarity and readability of the authors' original text. No LLM was used to generate original scientific content, analysis, or results. The authors take full responsibility for the integrity and validity of the work presented.

