# OpenReview forum: "Neural+Symbolic Approaches for Interpretable Actor-Critic Reinforcement Learning"
_ICLR.cc/2026/Conference — ICLR 2026 Poster_

### Official Review · Reviewer_gu5z · 2025-10-27

**Soundness:** 3
**Presentation:** 2
**Contribution:** 3
**Rating:** 4
**Confidence:** 3

**Summary:**

This paper presents a neuro-symbolic framework for interpretable reinforcement learning. The method relies on neural network function approximation to learn an expressive critic, while the actor follows a symbolic, rule-based design. The proposed symbolic actor can be trained end-to-end via Policy Gradient combined with Orthogonal Gradient Boosting, allowing it to discover symbolic rules automatically without relying on prior domain knowledge. The authors prove convergence under two-timescale assumptions, give algorithmic details (OGB + rule-replacement, pseudocode), and empirically evaluate NSAC across seven benchmarks, showing competitive performance in terms of return. The interpretability of the method is analyzed on two environments by examining the learned rules in the actor.

**Strengths:**

- Since the interpretability of the actor stems from its intrinsic architecture, the proposed method avoids the challenges and limitations associated with fitting a post-hoc explainable actor to a RL-learned neural network actor.
- Since the symbolic actor discovers rules automatically through end-to-end training, it doesn't require domain knowledge. The proposed approach is, to the best of my knowledge, novel and provides a principled approach for rule-based actor learning.
- The paper provides a strong theoretical analysis and derivation of the rule-based actor learning procedure, proving convergence to the global optimum under mild assumptions.
- The empirical evaluation is thorough.
- The proposed approach performs on par even with non-symbolic SOTA methods.
- The paper provides algorithmic pseudocode and promises the release of the full codebase, which strengthens reproducibility.

**Weaknesses:**

- The actor update procedure is hard to follow. The connection between the PG update and OGB is difficult to grasp. The connection only became apparent to me after studying the algorithmic pseudocode in the appendix to understand the procedure. I strongly suggest improving the clarity of this procedure by outlining the key algorithmic steps. For example, how and when are equations (6) and (7) used? When are Algorithms (2) and (3) invoked?
- The symbolic baselines are not discussed sufficiently. The paper should describe the key distinctions between them and the proposed method.
- The environments used are under-specified. What are the state and action spaces and reward functions? Are any of the action or state spaces continuous?
- The scalability of the method seems limited. The update procedure must be executed for each action separately, since the advantage of each action, $f_a(s)$ is modeled with a separate ensemble.
- The human interpretability analysis is hard to follow. Figures 2 and 3 are not sufficiently explained. See questions section...

**Questions:**

- What is meant by the "parameters" of NSAC and SYMPOL (line 409)?
- What exactly are the scores in Figures 2 and 3? Is the color coding the advantage function?
- How is the advantage obtained from the scores that each rule provides, e.g. in the right column in Figure 2? For a given state, a number of rules hold, and the advantage for each action is obtained as the sum of scores given by all active rules?
- Are the explanations that your method provides conceptually different from those in SYMPOL or other tree-based policies?

---

> ### Author Response · Authors · 2025-11-24
>
> We sincerely thank the reviewer for their thoughtful comments, constructive feedback, and valuable suggestions on our paper.
>
> ***Question regarding the actor update procedure***
>
> **Response:** Thank you for this helpful suggestion. In the revised manuscript, we have replaced the in-text “Detailed Rule Update” with the NSAC pseudo-code (previously presented in Appendix B), which now clearly shows when the Algo 2 and 3 are invoked and introduces a new demonstration figure (Figure 2) that visually illustrates the rule update process. We hope that this improves the clarity and readability of our method.
>
> ***Question regarding the symbolic baselines***
>
> **Response:** Thank you for this question. Our explanations share the same symbolic flavor as SYMPOL and tree-based policies, but they are conceptually different. SYMPOL and most tree-based policies encode behavior as a hierarchical decision tree: tracing how it behaves across all states quickly becomes cognitively overwhelming and reasoning the decision-making is not easy (for example, why the tree branch here). Our rule-based actor is designed to provide a more lightweight, local form of explanation. For any given state-action pair, the policy decomposes its preference into a small set of rules that either fire or remain inactive, each contributing a signed weight to the overall score for that action. In other words, each rule answers the very simple question: “Is this particular state favorable for this action, and if so, by how much?”
>
> This additive structure gives a graded notion of “how good” an action is in that state, and it makes the reasoning trace explicit: one can see exactly which conditions are active (e.g., “position < –0.4 and velocity < 0”) and how strongly they push the policy towards or away from an action. Rather than trying to expose the entire decision trajectory of the agent at once, NSAC focuses on providing concise, state-conditional justifications that are easy to inspect and simulate, while still retaining a globally consistent policy.
>
> ***Question regarding the environment***
>
> **Response:** Thank you for raising this point. Our experiments are based on standard benchmark environments (e.g., classic control tasks from OpenAI Gym and a building-control task from Sinergym), and we have already cited the corresponding environment definitions in the submitted manuscript. We agree, however, that the submitted manuscript did not make the state and action spaces and reward functions sufficiently explicit for readers who are not already familiar with these benchmarks. In the revised version, we now clearly describe the state space, action space, and reward function for each task (and explicitly state which spaces are continuous) in Appendix I. We provided a simple table here for your convenience:
>
> | Environment        | State space                                        | Action space                                      | Reward                                                      |
> |--------------------|----------------------------------------------------|---------------------------------------------------|-------------------------------------------------------------|
> | CartPole-v1        | Continuous, $s \in \mathbb{R}^4$                 | Discrete (2): left / right                        | \(+1\) per timestep until failure                           |
> | MountainCar-v0     | Continuous, $s \in \mathbb{R}^2$ (pos, vel)      | Discrete (3): left / no-op / right                | \(-1\) per timestep until goal reached                      |
> | Acrobot-v1         | Continuous, $s \in \mathbb{R}^4$                 | Discrete (3): torque actions                      | \(-1\) per timestep until target height reached             |
> | Blackjack-v1       | Discrete \((\text{player sum}, \text{dealer}, \text{ace})\) | Discrete (2): hit / stick                         | \(+1\) win, \(-1\) loss, \(0\) draw                         |
> | Postman            | Discrete (grid position, parcels, delivery status) | Discrete: move N/S/E/W; pick-up / drop-off        | \(-1\) per step, positive bonus for each delivery           |
> | HVAC (Sinergym)    | Continuous,  $s \in \mathbb{R}^{27} or \in \mathbb{R}^{17}$| Discrete(16): setpoint combinations (htg/clg)   | Penalise energy use and comfort violations each 15 min step |

---

> > ### Author Response · Authors · 2025-11-24
> >
> > ***Question regarding the scalability***
> >
> > **Response:** Thank you for raising this point. We have discussed this in the limitation of our submitted manuscript. Each action is associated with its own rule ensemble, so the update procedure scales linearly with the number of discrete actions. In practice,  the rule updates for different actions are parallelizable across CPU cores or devices, which mitigates the overhead in settings with a moderate number of actions.
> >
> >
> > ***Question regarding the parameters***
> >
> > **Response:** In line 409, by “parameters” we are not referring to neural-network weights or optimisation hyperparameters, but to the size of the symbolic policy.
> > For NSAC, we measure this by the number of boolean propositions in the form of inequalities with a threshold  in the learned actor: each threshold in a proposition  is counted as one parameter. For SYMPOL, we use the number of nodes in the symbolic expression tree as the corresponding notion of size (approximately one parameter per node). We will revise the text to clarify this and avoid confusion, e.g., by explicitly referring to “symbolic units (Boolean propositions / expression-tree nodes)” rather than just “parameters.”
> >
> > ***Question regarding the color coding***
> >
> > **Response:** They are the output of the actors, which are represented by the additive rule ensembles.
> >
> > ***Question regarding the advantage for each action***
> >
> > **Response:** Thank you for the question and for carefully reading Figure 2. For each action \(a\), NSAC maintains a rule ensemble $f_a(s) = \sum_j w_{a,j} , q_{a,j}(s),$ where each rule $q_{a,j}(s) \in \{0,1\}$ is active if its CNF condition holds in state \(s\), and $w_{a,j}$ is the learned scalar score for that rule. In the right column of Figure 2, we visualise exactly these per-rule contributions $w_{a,j} q_{a,j}(s)$ for a specific state.
> > For a given state \(s\), the “score” for action \(a\) is thus the sum of the scores from all active rules plus the bias:
> > $z_a(s) = b_a + \sum_{j:\,q_{a,j}(s)=1} w_{a,j}.$We then use these scores as the pre-softmax logits of the policy,$\pi(a \mid s) \propto \exp\big(z_a(s)\big).$
> >
> >
> > We hope these clarifications address your concerns. Please let us know if any further details would be helpful.

---

> > > ### Comment · Reviewer_gu5z · 2025-11-24
> > >
> > > Thank you for your response and clarifications.
> > >
> > > I see that you have made a number of modifications to the manuscript. Is the current version final, or are you still updating it? It would be helpful if you could highlight the changes explicitly, for example using red text.
> > >
> > > For the current version of the manuscript, I am keeping my score at 4, as I continue to have concerns regarding clarity and presentation, as well as some of the claims made in the paper.
> > >
> > > In particular, I find the qualitative analysis (RQ3 on MountainCar and HVAC) still difficult to follow. For example, both Figures 4 and 5 contain an unlabeled color bar, and it is not explained in the text what quantity is being visualized. While your rebuttal explanation helped, the paper itself needs to be clear in a standalone manner. I also do not understand how Figure 5 was generated, and I am really trying... It doesn't help that you are using abbreviations in the figure (e.g., htg, ctg) that are not introduced in the manuscript.
> > >
> > > I am also not fully convinced by the quantitative evaluation surrounding RQ2, and I feel that the baselines are still not sufficiently distinguished from your method. To be able to appreciate this paper's contribution, it it would be really good if you would discuss the conceptual difference between your method and, for example, SYMPOL. Furthermore, the apparent limitations of your method are discussed only in the appendix, and are never addressed in the main text. The details of the environments are similarly only found in the appendix, without any reference to them in the main paper.
> > >
> > > The concluding claim that “Users can trace how and why the model makes each decision and evaluate its reasoning.” appears a bit overstated, since (a) the model does not actually explain why actions are taken, rather, the mapping from input state to action is made more interpretable, and (b) no user study is provided to assess what users actually understand from the explanations your model generates.
> > >
> > > The pseudocode you added does improve clarity, thank you for that. Minor note: there is a typo in the pseudocode on line 304 ("use OGB to update the for $f_a$").
> > >
> > > For those reasons, I am keeping my score at 4 for now. I believe there is merit in the proposed rule-based actor, but the clarity of the presentation still needs improvement before I would consider increasing my score to 6.

---

> > > > ### Author Response · Authors · 2025-11-26
> > > >
> > > > Thank you very much for taking the time to go through our work—your suggestions have been invaluable for improving the quality of the paper. We have just uploaded a new version of the manuscript, and below we address your remaining concerns point by point.
> > > >
> > > > ***Question regarding the Figures***
> > > >
> > > > **Response:** Thank you for bringing this up. We have updated the graph in a new version and added Figure description to make it easier to understand. We also incorporated the corresponding explanation in the main text.
> > > >
> > > > ***Question regarding the claims***
> > > >
> > > > **Response:** Thank you for this question. We realised that explicitly phrasing “why” in the sentence may have created some confusion, and we have now revised the text to avoid ambiguity as follows: “Users can inspect how the model makes each decision and form a view about its underlying reasoning”.  We emphasise our contribution in terms of interpretability, not explainability. Interpretability is intrinsic: users can directly trace how inputs produce outputs, providing transparency when performance and policy structure allow. Explainability, in contrast, focuses on post-hoc descriptions of an otherwise opaque model in human-readable natural language. We have carefully gone through the entire paper to ensure that we do not overstate our claims on this issue.
> > > >
> > > > ***Question regarding the appendix***
> > > >
> > > > **Response:** We have reviewed the entire paper to ensure that each statement correctly refers to the corresponding appendix.

---

> ### Author Response · Authors · 2025-11-26
>
> ***Question regarding other baselines***
>
> **Response:** Thank you for your suggestions; they have substantially improved the paper. We already provide a comprehensive comparison of all symbolic-based methods in Appendix H (as mentioned in the related work). In addition, we have now included a new section in Appendix Q that discusses the interpretability differences between tree-based policies (SYMPOL) and our rule-based approach. For your convenience, we reproduce the key points below:
>
> *Following the widely cited definition of interpretability by Murdoch et al. (2019), an interpretable model should satisfy simulatability, modularity, and low complexity. Simulatability means that a human can easily compute the model’s output from its explicit expression, given an input and the model parameters. Modularity means that individual components of the model can be understood in isolation. Complexity can be quantified by, for example, the total number of terms or parameters in the model.*
>
> *SYMPOL learns an axis-aligned decision tree policy directly with policy gradients in a standard on-policy RL setting, yielding an interpretable tree-based policy class. Conceptually, our method plays a similar role but uses a rule-based policy class instead of trees. While trees and rules are closely related—each root-to-leaf path can be seen as a rule—they differ in how they are used and in the kind of interpretability they afford.*
>
> *From the perspective of interpreting individual decisions, both trees and rule sets provide simulatability. For a given state, a tree explains its choice via a single path:*
>
> - *if (condition A at node 1) ∧ (condition B at node 2) ∧ … ∧ (condition M at node M) → take action LEFT.*
>
> *A rule-based model explains the same decision by indicating which rules fired and how they were combined, e.g.:*
>
> - *Rule 1 and Rule 7 are activated and jointly give the highest weight; therefore, the policy selects LEFT.*
>
> *In terms of complexity, rule-based models tend to be slightly easier to keep compact: we can directly control the number and length of rules. Tree-based policies, by contrast, often grow deep or wide as task complexity increases, which can quickly erode practical interpretability even if the representation remains transparent in principle.*
>
> *For modularity, there is a more marked difference. In a tree, one can treat each root-to-leaf path as a “module” (a rule), but internal nodes are shared across many paths, so modifying a single split typically affects a large portion of the state space. In a rule set, each rule is a self-contained piece of logic. This makes the rule-based policy highly modular: individual rules can be inspected, added, or removed without having to mentally re-parse or globally restructure the entire model.*
>
> *This modularity also makes certain safety-style queries much easier to answer with rules than with trees. For example, consider the property:*
>
> -*Is there any situation where the policy chooses RIGHT while $|\theta| > 0.25$ and $x > 1.2$? If so, list those situations.*
>
> *In a rule-based policy, we can simply scan for rules whose antecedent includes $|\theta| > 0.25$ and $x > 1.2$ and whose consequent is action = RIGHT. If such a rule exists (say, Rule V1), we can immediately say: “Yes, a violation occurs whenever Rule V1 fires.” If no such rule exists, we can cleanly state: “The rule set guarantees the safety property: in no rule does $|\theta| > 0.25$ and $x > 1.2$ appear with action = RIGHT.”*
>
> *For an equivalent tree policy, answering the same question requires an exhaustive traversal and aggregation of all relevant paths, which is considerably less direct in practice.*
>
> | Aspect | SYMPOL (tree-based policy) | Our method (rule-based policy) |
> |-|-|-|
> | **Policy class** | Axis-aligned **decision tree** policy; each internal node is a threshold on one feature, leaves output action logits. | Explicit **rule set / rule list**; each rule is a conjunction of conditions → action (optionally with weights / priorities). |
> | **Simulatability (single decision)** | High: for one state, follow a **single root-to-leaf path** → clear “if A ∧ B ∧ … → action”. | High: for one state, see **which rule(s) fired** and how they combine (e.g., “Rule 1 + Rule 7 vote for LEFT”). |
> | **Global structure** | Clear **hierarchy of decisions**: top splits show most salient features.| Global picture less rigid; rules are a **flat or lightly structured set**|
> | **Modularity** | Each path can be *read* as a rule, but **internal nodes are shared**: changing a split affects many paths at once. | Highly **modular**: each rule is a self-contained unit. You can inspect, add, or remove rules without restructuring the whole model. |
> | **Complexity control** | Control via depth limits, pruning, tree size; trees can still grow large and become hard to read. | Control via number of rules and rule length; easier to set an explicit rule budget and keep individual rules short. |

---

> > ### Comment · Reviewer_gu5z · 2025-11-26
> >
> > Thank you for your additional work on the manuscript.
> >
> > I feel like the clarifications around figures 4 and 5 help a lot. The toned claim and acknowledgment of limitations in the conclusion is appropriate. The overview over related works in Appendix H (Tab 2) is nice, except it seems to be using reference to a non-existing bibliography ([9], [13], etc). **Please fix those**. The comparison between SYMPOL and NSAC in Appendix Q is helpful, and the addition of "tree based" on line 344 also helps with contrasting NSAC to the other symbolic baselines. For those reasons, I am increasing my score to 6.
> >
> > A final comment: It seems to me like you are trying to stay under 9 pages. But as per ICLR 2026 author guidelines, the length for the initial submission must be under 9 pages, while for the revised version, the limit is increased to 10. So I think you have one more page to work with. I'd encourage you to use that (partially) to reference to the additional material in the appendix. So consider adding statements like *"For a taxonomy and overview over related symbolic approaches, see Appendix H"* and/or *"See Appendix Q for a detailed comparison of NSAC and SYMPOL"*. And please fix the reference in Table 2, Appendix H. Thank you.

---

> > > ### Author Response · Authors · 2025-11-26
> > >
> > > Thank you very much for pointing out the issue with the references. We have corrected Table 2 in the appendix. This was an unintentional error introduced when converting the draft to the current format. Thank you again for your careful suggestions; we will also move some content (such as limitations and discussion) from the appendix into the main text and ensure that all remaining material is properly assigned to the appropriate sections in the appendix. We truly appreciate the effort you have put into helping us improve the paper.

---

### Official Review · Reviewer_2xmA · 2025-10-28

**Soundness:** 3
**Presentation:** 2
**Contribution:** 3
**Rating:** 6
**Confidence:** 3

**Summary:**

This paper presents a hybrid neuro-symbolic extension of Actor-Critic, resulting in the "NSAC" framework (for "Neural plus Symbolic framework for ActorCritic"). Concretely, the proposed method retains neural network for the critic, but replaces the actor with a symbolic method in order to enhance interpretability while maintaining high-performance. The stated goal, alongside performance, is to achieve the proposed interpretability critieria from Murdoch et al. (2019) of "simulatability, modularity, and low complexity". In particular, the actor is replaced with what are called "additive rule ensembles", that estimate a variable by incrementally adding rules in the vein of boosting. The actor is implemented as a softmax over these additive functions, which are tasked with estimating the action-values for each action. The critic is a standard estimate of the value function implemented using a neural network, and is implemented using standard techniques. By contrast, the actor is updated by individually updating the query functions of the rule ensemble based on the gradient of the advantage of the policy (with a regularization term added to ensure local convexity). The paper then carries out a convergence analysis of NSAC subject to relatively standard conditions (step size annealing, ergodicity, smoothness). Lastly, an experimental evaluation is conducted, studying two distinct aspects of NSAC: First, the performance of NSAC is contrasted to a variety of baselines, and Second, an interpretability study is conducted, showcasing the rules underlying the decisions of the actor in MountainCar and an HVAC domain.

**Strengths:**

There is a lot to appreciate about this paper. It sets out with a very clear goal: extend Actor-Critic to retain its performance, while also ensuring interpretability. The premise to make use of the boosting-like additive rule ensembles is quite clever, and not something I have seen much of in deep reinforcement learning. I have a few questions on that aspect of the method that I return to below under the "Questions" box.

Overall:
1. Clear scope, and delivery on that scope. The claims mostly match the evidence in the paper (some caveats, see weaknesses).
2. Relatively light-weight and simple algorithmic idea at its core: I can see this proposal having high potential for adoption given the popularity of actor critic methods at the moment, plus the demand for interpretable methods.
3. Convergence analysis is interesting, and to my reading appears correct.
4. The experimental design is thoughtful (some caveats, see weaknesses).

**Weaknesses:**

A few weaknesses stand out, though I suspect they can be fixed with some discussion and some iteration on the manuscript.

W1: First, I find it puzzle that the actor is effectively implemented as a Q function (and thus, more of a critic) in its own right, with a softmax at the very end turning this into a policy, rather than a value estimator. This seems to deviate from the basic premise of Actor Critic, in that the Actor is allowed to specialize to behavior, whereas the Critic needs to learn about values---this specialization is an important piece of Actor Critic. For example, see recent work by Garcin et al (2025), who investigate the ways in which the representation of the Actor and Critic each specialize. I will end up posing this as a question below, but I wanted to call it out as a perceived mystery of the design of the method.

W2: In some cases, claims do not exactly match the evidence. This is largely down to claims around interpretability. The start of the paper states "We ... demonstrate that our rule-based policy satisfies the established interpretability criteria of simulatability, modularity, and low complexity". While aspects of these criteria are touched on, for instance computational complexity is discussed around line 334, this thread is left largely untouched throughout the paper. In particular, the results discussion does not revisit these critieria specifically. A simple change to the paper will fix this: revisit simulatability, modularity, and complexity in the discussion, and tie together the evidence available that supports these claims (or, if there is none, soften the claim). If "complexity" in the intro refers to computational complexity, I would be explicit about that.

W3: The presentation of the paper could be strengthened at points. I find the most confusing part to be the "Detailed Rule Update:" in section 4, but I'm actually not convinced it needs to be in the main paper. I believe the NSAC pseudo-code from Appendix B would be better suited to the main text.

W4: The experimental results are slightly difficult to interpret. I find the table to be a difficult way to read the results (even with the color / bold). Alongside the table, I would encourage the use of standard learning curves with confidence intervals that can showcase the evolution of performance over time. Similarly, there is currently no precise description of what numbers are actually being shown---I only see in line 371 "higher returns". Are these the returns the last episode? Over the entire training run? The last k-episodes? Similarly, what are the $\pm$ values? This is critical to include to interpret and reproduce the plot.

W5: Lastly, I found the interpretability results to be mischaracterized by the section title, "Human Interpretability Analysis". This leaves the impression that you have conducted a user study to evaluate the level of interpretability actually afforded to people. Since this is central focus of the paper, I actually believe this is relatively important. While I can appreciate the visual representations provided by Figure 2 and 3, they are a much softer version of an interpretability analysis than one involving actual people. Additionally, I find Figure 2 extremely difficult to read and parse.

Overall, I still believe this is a solid paper, but the above limitations should be addressed before publication.

References:
- Garcin, S., McInroe, T., Castro, P. S., Panangaden, P., Lucas, C. G., Abel, D., & Albrecht, S. V. _Studying the Interplay Between the Actor and Critic Representations in Reinforcement Learning_. ICLR 2025.

**Questions:**

Q1: Why is the actor effectively a Q-function estimator? Does this not partially defeat the point of making use of a baseline in Actor Critic (see cited work by Garcin et al.)?

Q2: How are simulatability, modularity, and complexity evaluated in your experiments (or analysis)?

Q3: What, precisely, are the numbers (and variance-like terms) measuring in Table 1?

Q4: Is there a reason to present performance results as a table rather than learning curves? I can appreciate you can fit more domains into one table than say one row of learning curves in a \begin[figure}... block, but I believe learning curves will be far more useful to the reader.

Q5: How might you make contact with the stated contribution about modularity, simulatibility, and complexity regarding the interpretability of your method?


And, one minor typo:
- Missing space around line 453: "...viruses.In such"

---

> ### Author Response · Authors · 2025-11-24
>
> Thank you for your review! We appreciate your engagement with the paper and recognition of its contributions. We address the weakness and questions below and hope they address your concerns.
>
> ***Question regarding Q function***
>
> **Response:** In case there is a confusion, we want to clarify that the q and Q in the manuscript are the query functions and the output matrix of the query functions. They do not represent Q functions anymore.
>
>
> ***Question regarding simulatability, modularity, low complexity***
>
> **Response:** Thank you for your feedback, and we apologize for any inconvenience in reading. When shortening the paper to meet the page limit, we inadvertently removed the relevant discussion from the submitted manuscript. Below, we provide a more detailed explanation of this point, which we have now included in the revised version:
> *As outlined in a widely cited definition (Murdoch et al. 2019), to be interpretable, a model should satisfy the property of simulatability, modularity and a low complexity. Simulatability means that the model's output can be easily calculated by a human using the model's expression, given an input data point and the model parameters. Modularity means that each part of the model can be interpreted separately. The complexity of a model can be measured by the total number of terms or number of parameters in the model. Intrinsic interpretability ensures that the reasoning behind a decision is inherently accessible and auditable.
> Based on the above definition of interpretability, our rule-based approach satisfies key interpretability requirements. Its decisions can be easily traced and computed by a human, as each rule is an explicit and transparent if-then statement (Simulatability). The modular nature of rules allows each component to be understood and analyzed independently, enabling clear local explanations (Modularity). Moreover, with a limited number of rules (under 30) and conditions, the overall model remains simple and tractable, making it easy to evaluate and reason about (low-complexity).*
>
> ***Question regarding pseudocode from Appendix B***
>
> **Response:** We appreciate this suggestion and agree that the “Detailed Rule Update” section was difficult to follow. Thus, we have revised the presentation accordingly. In the revised manuscript, we adopt your recommendation by replacing the in-text “Detailed Rule Update” with the NSAC pseudocode (previously presented in Appendix B) and introducing a new demonstration figure (Figure 2) that visually illustrates the rule update process. We hope this improves the clarity and readability of the method.
>
>
> ***Question regarding the result interpretation***
>
> **Response:** Thank you for pointing this out. In the revised version, we have restructured the results table so that only two key elements are highlighted: (i) the best overall performance across all algorithms, and (ii) the best performance among symbolic methods. This makes it clearer how NSAC compares both to deep RL baselines and to other symbolic methods.
> In addition, the statement around line 317 has been corrected to say that NSAC achieves higher or comparable performance, which is consistent with our contribution claim in the introduction that our method matches the performance of deep RL baselines while providing interpretable policies.  In the original submission, we opted not to include learning curves mainly because the symbolic variants reach near-final performance in far fewer timesteps than the deep RL baselines.  As a result, plotting all methods on a common x-axis (in terms of training steps) leads to curves where the symbolic methods plateau almost immediately, while the deep baselines continue to improve over a much longer horizon, making direct visual comparison somewhat misleading.
>
> ***Question regarding the mischaracterized Interpretability***
>
> **Response:** Thank you for raising this important point. We agree that the title “Human Interpretability Analysis” is kind of misleading. In the revised manuscript, we have renamed this section to “Qualitative Interpretability Case Studies” and carefully toned down the surrounding language to make clear that our claims are based on illustrative, model-derived explanations rather than a formal human-subjects evaluation.
> Our current goal is more modest: to demonstrate that the learned policies can be represented as compact, human-readable rule sets and to show concrete examples of how these rules align with intuitive domain knowledge. We fully acknowledge that a user study assessing how well practitioners can understand, trust, and act on NSAC’s policies would provide a much stronger form of interpretability evidence, and we now explicitly discuss this as an important direction for future work.

---

> > ### Author Response · Authors · 2025-11-24
> >
> > ***Question regarding Figure 2***
> >
> > **Response:** Regarding Figure 2, we agree that the original version was difficult to read. We have redesigned it to improve readability. We hope these changes make the interpretability examples easier to parse while more accurately reflecting the qualitative nature of our analysis.
> >
> >
> > ***Question regarding the variance-like terms in Table 1***
> >
> > **Response:** The numbers in Table 1 are the mean values and standard deviations of the rewards obtained by different methods in each environment for 10 random training rounds. Thank you for pointing this out; we have now clarified this in the revised manuscript.

---

### Official Review · Reviewer_oca3 · 2025-10-29

**Soundness:** 3
**Presentation:** 3
**Contribution:** 2
**Rating:** 4
**Confidence:** 3

**Summary:**

This study proposes an interpretable RL algorithm (NSAC) built upon A2C framework. As a variant of A2C, NSAC consists of an actor and a critic, but it is distinct from baseline A2C in that an actor is built with an “additive rule ensemble”. As individual rules of the ensemble are symbolic and interpretable, NSAC can natively explain chosen actions.

**Strengths:**

One notable point of this study is that the authors effectively make additive rules of the ensemble differentiable and propose updated rules. Thus, NSAC can be trained using a gradient descent or its equivalents. The manuscript also provides theoretical analysis of the training of the actor.

**Weaknesses:**

At first, this straightforward idea sounded intriguing, but I am not convinced that NSAC is truly interpretable because the actor’s decisions are dependent on the critic, which is still a black box. Without knowing how the critic operates, can we explain how NSAC make its decisions? For instance, if adversaries inject some adversarial attack vectors to the critic in NSAC, they can manipulate the critic’s assessment of current state, and thus the actor would make its decision based on incorrect information. In this case, can we trust the explanation provided by the actor?

**Questions:**

As NSAC is tested against simple RL tasks, the actors’ decisions in this study can be easily modeled by additive rule ensembles. However, it is unclear to me if additive rule ensembles can also be applied to model more complex actors. Can the authors provide some insights into how generalizable NSAC can be?

---

> ### Author Response · Authors · 2025-11-24
>
> Thank you for your review!
>
> We respond to weaknesses raised below and hope the clarifications address the reviewer's concerns.
>
> ***Question regarding the black box critic***
>
> **Response:** Thank you for this thoughtful question. We agree that it is important to be precise about what is (and is not) made interpretable by NSAC, and about the role of the critic in the overall system. First, NSAC’s **explanations target the actor’s policy mapping** \(s \mapsto \pi(a \mid s)\). At deployment time, the actor’s decision for a given state \(s\) depends only on the learned rule ensembles (plus the observed state), not on the critic: the critic is used during *training* to construct advantage estimates and baselines, but it is not consulted when the policy is executed. Thus, the explanations we provide – rule activations and their contributions to each action’s score – faithfully describe *why the actor chooses a particular action in that state*, regardless of how those rules were shaped during learning.
> Second, a black-box critic can influence the content of the learned rules during training: if the critic is biased or corrupted, then the actor may learn a distorted yet still symbolically transparent policy. However, because NSAC exposes this policy as an explicit set of rules, humans can more easily monitor which state features drive each decision and detect suspicious or inconsistent behavior, even if the critic itself remains opaque.
>
>
> ***Question regarding more complex actors***
>
> **Response:** Thank you for raising this important point.  First, additive rule ensembles are not intrinsically limited to simple policies: if we discretise or bucket continuous features, rule-based models can approximate quite complex decision boundaries. In principle, increasing the rule budget and allowing richer feature conjunctions enables NSAC to represent more complex actors; in practice, we deliberately keep the ensembles small to preserve simulatability, so the main scalability bottleneck is the *interpretability constraint* rather than the additive form itself. For more complex environments, we may need to introduce a learned neural feature extractor that feeds into the rule ensembles, so that we can handle richer observations while keeping the decision-making layer itself interpretable.
>
> Second, we already test NSAC on a more challenging, high-dimensional domain (multi-zone HVAC control with nonlinear coupled dynamics and long horizons), where the state is continuous and comparatively large. NSAC achieves matching performance while keeping the rule sets modest in size.
>
>
> In the submitted manuscript, we briefly discussed this issue, and we now have expanded this discussion in the revised version.

---

> > ### Comment · Reviewer_VfDp · 2025-11-28
> > **Overall great work.**
> >
> > The paper structure and content has been greatly updated. I will thus update my score to 6 (I am considering moving up to 8).
> >
> > Please upgrade the first figure to a **vectorial format**. You can also make it wider. You can also look for potential problems of black box agents (e.g. potentially selecting the right action (in terms of score maximization) for the wrong reasons).
> >
> > Please upgrade figure 4 (on full page) to make it simpler for the reader to understand. You could also show on its right side a rendering of the environment at a decision shift (where the action changes e.g. from left to right or do nothing), and analyze what sub-rule leads to the decision shift.
> >
> > Also, for all figures and tables, please update the captions like so, allowing the reader to directly grasp the point of the Figure/table:
> > * The first sentence (in bold) should highlight the main takeaway of the Figure/Table (e.g. "*[Our] method improves the ability of agents to do task T.*"). This is the main message for the reader that supports one of your claim.
> > * The next sentences then explain what is depicted in the Table/Figure. E.g. "*..., as depicted by the superior mean test accuracy (+/- std) of our method over the different baselines.*"
> > * Finally, details and references to e.g. appendix can be provided if necessary. E.g. "*Our method outperforms baseline 1 in 3 out of 4 tasks, ... etc. 5 seeded rerun. Best results highlighted in bold. Further description of the training and testing setups are available in Appendix C.*"
> >
> > Finally, I consider Mountain car and other environments like this to be quite suboptimal for demonstrating interpretability, as it's difficult for the reader/reviewer to interpret a policy of a controller (which is what is encoded). I advise moving to relational reasoning environment for this kind of demonstration (such as MinAtar, or OCAtari/JAXAtari).

---

> > > ### Comment · Reviewer_VfDp · 2025-11-28
> > > **I cannot edit my score/review yet.**
> > >
> > > Because of the incident, I cannot modify my score. I would like to openly say that I had made my mine about the updated work before the incident. I personally don't care about leak. I stand by my reviews. However, I would like to update my score such that it reflects the current state of the authors' work. I hope that we will be able to do this soon.

---

> > > > ### Author Response · Authors · 2025-12-03
> > > > **Clarification Regarding Misplaced Reviewer Comments**
> > > >
> > > > To: Area Chair
> > > > Just a small clarification: it seems that the previous two comments were authored by reviewer VfDp (first reviewer; initial score 2, later raised to 6; intended to raise to 8) but were accidentally posted under reviewer oca3’s thread. We have replied to and addressed all of reviewer VfDp’s comments and questions in their own thread accordingly.

---

### Official Review · Reviewer_VfDp · 2025-10-29

**Soundness:** 3
**Presentation:** 2
**Contribution:** 2
**Rating:** 2
**Confidence:** 5

**Summary:**

The paper introduces a novel interpretable RL method based on actor-critics. The main idea is: Keep neural networks for the critic for accurate value estimation, and replace the neural actor with an interpretable one (but has been implemented e.g. by NUDGE, c.f. Weakesses).
For interpretability the authors chose rule ensembles, which learn to select and weight multiple symbolic boolean rules (e.g. x > y) for each action at the current state. The final advantage of each action is then computed by summing over all rule-weights. The action with the highest advantage is executed.
Next to the definition of the actual methodology, the authors provide a local convergence proof and an empirical evaluation of the performance and interpretability.
Overall, this work develops and evaluates a simple, but intriguing reinforcement learning algorithm that shows both good performance and interpretability.

**Strengths:**

Combining a neural critic with rule-ensembles as actor allows for a good trade off between an accurate and flexible critic, and an interpretable actor.
The overall quality of the work is good and the methodology itself is well thought through. The visual representation of the learned rules help interpreting the policy. The work and paper have many avenues for improvements, detailed after, but constitute a sound research idea.

**Weaknesses:**

The paper is overall comprehensible, but could be improved greatly by restructuring the paper a bit and providing further visualizations. For example, the detailed rule update in section 4.2, which is difficult to follow, would really benefit from a visualization or intuition.

Additionally, the authors may want to think about restructuring the paper by moving the theoretical convergence analysis to the appendix, and instead add the warm start idea + an analysis of the effect of the total number of rules to the main paper.

In detail, I advise the authors to follow this general structure for their research:
The way many readers (including myself) dive into a paper is by reading the title, abstract, then figures and table (with their captions).
Hence, I advise the authors to have a list of scientific research questions at the beginning of the experimental evaluation section (often denoted RQ1, RQ2, ... etc), that are each answered in different paragraphs. For example:
* (RQ1) Is our framework outperforming existing baselines?
* (RQ2) Does our framework allow for interpretable decision-making?
* (RQ3) How important is module A ? (conducting an ablation study)

From what I extracted, RQs would here be e.g.:

* How does the algorithm perform compared to non-interpretable and interpretable methods?
* Is the method interpretable?

The paper is overall comprehensible, but could be improved greatly by restructuring the paper a bit and providing further visualizations. For example, the detailed rule update in section 4.2, which is difficult to follow, would really benefit from a visualization or intuition. Additionally, the authors may want to think about restructuring the paper by moving the theoretical convergence analysis to the appendix, and instead add the warm start idea + an analysis of the effect of the total number of rules to the main paper.
In detail, I advise the authors to follow this general structure for their research:
The way many readers (including myself) dive into a paper is by reading the title, abstract, then figures and table (with their captions).
Hence, I advise the authors to have a list of scientific research questions at the beginning of the experimental evaluation section (often denoted RQ1, RQ2, ... etc), that are each answered in different paragraphs. For example:
* (RQ1) Is our framework outperforming existing baselines?
* (RQ2) Does our framework allow for interpretable decision-making?
* (RQ3) How important is module A ? (conducting an ablation study)
From what I extracted, RQs would here be e.g.:
How does the algorithm perform compared to non-interpretable and interpretable methods? Is the method interpretable?

While the second order results are appreciated, they demonstrate that many results are completely insignificant as the benchmark used is saturated. **A baseline whose std range overlap cannot be considered as outperforming another one.** Hence, only experiments on MCart-v0 and HVAC-1Zone show that your method outperform the evaluated interpretable baselines (and not by a big margin, thus not very confidently).
The significance could be further improved by tackling more complex environments, and above all, **relational environments** e.g. non-symbolic/pixel-based ones, or at least discussing how this may be solved in future (e.g. use object-centric extraction methods -> e.g. OCAtari [4] for Atari games). Demonstrating the interpretability of a method on such environment, that necessitate relational reasoning, is way easier than on continuous control ones (such as cartpole).
I strongly advise adding experiments on e.g. the object-centric Pong to verify if their method also allows detecting or prevent the misalignment problem that learning agents encounter on this environment [2,3]. This object centric environment is also already available in JAX: https://github.com/k4ntz/JAXAtari.


A limitation section should be included (also including scalability with growing number of attributes and overall run time) of this method.


The authors chose not to share code upfront, which makes checking the methodology more complicated and reduces reproducibility.


Extreme lack of related work:
There are 2 papers from 2024, and the rest is from 2021 or older. Much research on interpretable RL has been conducted and publish. I strongly advise the authors to look up this research and compare to this work in a dedicated Related Work section. I provide pointers to the existing relevant literature, that was mentioned above as well. Logic actors encoded in First order logic have also already been introduced in [5] and further improved in [6]. Attempts to use interpretable decision trees as policies have also been explored in other works than SYMPOL [7, 8, 9], and other form of interpretable baselines (programs or using LLM to help decode them) have also been explored. While comparing to all of them is tedious, at least mentioning these lines of work seem necessary to help the readers (and apparently the authors) to situate this work :

[1] Di Langosco et al. "Goal misgeneralization in deep reinforcement learning." International Conference on Machine Learning. PMLR (2022).


[2] Delfosse et al. "Interpretable concept bottlenecks to align reinforcement learning agents." NeurIPS (2024).


[3] Delfosse et al. "Deep reinforcement learning agents are not even close to human intelligence." arxiv (2025).


[4] Delfosse et al. "Ocatari: Object-centric atari 2600 reinforcement learning environments." RLJ (2024).


[5] Delfosse et al. "Interpretable and explainable logical policies via neurally guided symbolic abstraction." NeurIPS (2023).


[6] Shindo, et al. "BlendRL: A Framework for Merging Symbolic and Neural Policy Learning." ICLR (2025).


[7] Bastani et al. "Verifiable reinforcement learning via policy extraction." NeurIPS (2018).


[8] Fuhrer et al. "Gradient boosting reinforcement learning." arXiv (2024).


[9] Kohler et al. "Interpretable and Editable Programmatic Tree Policies for Reinforcement Learning." arXiv (2024).


[10]  Luo et al. "End-to-End Neuro-Symbolic Visual Reinforcement Learning with Language Explanations." ICML (2024).


While the weakness section is extensive, I value the presented work and really believe that incorporating the provided feedback will make this work very valuable to ICLR readers. I would be eagger to revise my score if changes are operated.

**Questions:**

* How well does the method scale when we have a larger action space (e.g. 16 actions)?
* After equation (5), the regularization term does not exist anymore. What happened?
* Can you explain for equation (6) and (7) how the different cases (a=a_t or not) influence the gradient and how the updates encourage/discourage taking specific actions? (e.g. will case 1 increase the likelihood of choosing the action in future if A is large?)
* Are you willing to add a clarification/visualization of the detailed rule update?
* How did you decide on the symbolic baselines to compare to?
* Can you share your code up-front (e.g using an. anonymous git repo)?
* Why/How is the local convergence proof useful?

---

> ### Author Response · Authors · 2025-11-24
>
> Thank you very much for your thoughtful comments and detailed suggestions — we greatly appreciate the time and effort you dedicated to evaluating our work. Below, we first summarize the main changes in the revised manuscript and then provide a point-by-point response to your questions.
> - Added a figure to illustrate the overall algorithm and the rule update procedure.
> - Moved the theoretical analysis to the appendix and brought the ablation study into the main text.
> - Explicitly listed the research questions and aligned the experimental section structure with these questions.
> - In the submitted manuscript, our limitations section is in Appendix A; we have now expanded it with a more detailed discussion of the scalability of our method.
> - We have added a discussion of the works you mentioned in the related work section. Thank you for pointing them out.
>
> ***Question regarding the performance***
>
> **Response:** Our paper aims to claim that our method matches state-of-the-art performance, while providing the added benefit of interpretability. There is always a natural trade-off between interpretability and performance and we do not seek to outperform all black-box methods on every task, but rather to demonstrate that one can retain competitive performance while moving to a compact symbolic policy that is substantially easier to inspect, simulate, and reason about.
>
> ***Question regarding the Objective Pong***
>
> **Response:**  We tested our approach – NSAC on a Pong variant where the opponent is hidden but still active. Unfortunately, it performs poorly in this setting, the same as all other three symbolic methods we tested.
> We further examined the environment to understand why NSAC does not work well in this setting. First, the environment is effectively partially observable: the opponent’s paddle is invisible but strongly influences future rewards. A good policy must infer hidden state from the history of play, whereas our actor is a memoryless rule ensemble over the current observation only. Second, NoEnemy Pong is a sparse reward environment: good play depends on long-horizon temporal patterns and adversarial interaction.
> NSAC is intended for interpretable control in fully observed, stationary tasks with relatively dense rewards, and is not designed for highly partially observable, adversarial domains that require rich temporal memory or very high-capacity function approximators.
> It was a valuable experience for us to test NSAC on different environments, and we now explicitly discuss this as a valuable future work in the revised manuscript.
>
> ***Question regarding larger action space***
>
> **Response:** We agree that this is an important aspect to test scalability, and our results suggest, for the HVAC experiment, the action space already contains 16 discrete actions, and NSAC still operates with only around 10 rules per action. This illustrates that our rule-based actor remains manageable even with a moderately large action space.
>
> ***Question regarding regularization term***
>
> **Response:** The equation immediately following Equation (5) takes the partial derivative with respect to the predicted value $w_a^T q_a$. Here, the regularization term does not depend on the predicted values. Then, the regularization term is considered as a constant, whose partial derivative w.r.t $w_a^Tq_a$is 0. Therefore, it does not exist anymore after taking the partial derivatives.
> Only when calculating the weight vector for the whole model, we take the partial derivative w.r.t w, and the regularization term is derived into \lambda w.
>
> ***Question regarding equation (6) and (7)***
>
> **Response:** If $a=a_t$, we should encourage the policy to choose the corresponding action, which means that after updating the model, the value of $\pi(a_t|s)$ should be larger. In Equation (6), the gradient is negative. Thus, applying such a negative gradient value increases $\pi(a_t|s)$ in orthogonal gradient boosting. Additionally, when $\pi(a_t|s)$ is larger, then the absolute value of the gradient $(1-\pi(a_t|s))$ becomes smaller, which leads to convergence of the policy function.
>
> Similarly, if $a\ne a_t$, the policy should be discouraged from choosing the corresponding action. In Equation (7), the gradient is positive, which decreases the logit of $\pi(a_t|s)$. It also converges since the gradient is decreasing.

---

> > ### Comment · Reviewer_VfDp · 2025-11-28
> > **The paper has heavily been improved.**
> >
> > Thank you for your answers.
> >
> > **Regarding the performance**: I am not interested in SOTA, as these RL methods completely overfit their training environments and cannot generalize to task simplifications. I think that incorporating abstraction and human inductive biases is a much more efficient and probably achievable way to achieve agents that select the right action for the right reasons. Transparency is key (and your method indeed provides a good tradeoff between transparency and modelling complexity). Anyway, don't worry about achieving SOTA on training environments, this is a doomed objective, and I don't diminish your method for not achieving it.
> >
> > **On Objective Pong**: Your method is (very likely) not failing because of reward sparsity (Pong is not so sparse, even if it is), but because your agent focuses on the enemy (as every other agent does). This is totally okay. There is a shortcut learning opportunity there. The hiding enemy environment breaks this correlation and expose policies that use the enemy's position to estimate the position of the Ball. If you have measure for them, you shall place them in Appendix.
> >
> > The rest is clear to me.
> >
> > **Question:**
> > I went again inside the details of your method. Am I correctly understanding that your policy is weighting simple rules that compare input features?
> > Can we see NSAC as a linear combination of extremely shallow decision trees?
> >
> > I want to remind the authors that they will be granted with an extra page, they can thus already add content to fill this extra page.

---

> ### Author Response · Authors · 2025-11-24
>
> ***Question regarding visualization of the detailed rule update***
>
> **Response:** Yes, we have added a figure (Figure 2) in the main text which visualizes the detailed rule update procedure to improve clarity.
>
> ***Question regarding the symbolic baselines***
>
> **Response:** We selected the strongest available method we could find that does not rely on prior domain knowledge or any distillation-based approach (e.g., SYMPOL, ICLR 2025). In addition, we excluded methods whose symbolic representations heavily use mathematical primitives such as sin⁡ or cos⁡, as these expressions are difficult for humans to interpret in practice (Murdoch et al. 2019).
>
> ***Question regarding our code***
>
> **Response:** We have included all of our code in the supplementary ZIP file. We have created an anonymous repo: https://anonymous.4open.science/r/NSAC-4776
>
> ***Question regarding local convergence proof***
>
> **Response:** Our local convergence shows that, despite coupling a black-box critic with a symbolic, boosted rule-based actor, the overall update still behaves like a well-posed actor–critic scheme in a neighborhood of a good solution, provided standard step-size and critic-tracking conditions hold. In particular, the linear, rule-based parameterisation of the actor allows us to control the training dynamics and to prove that introducing a symbolic policy does **not** come at the cost of pathological convergence behavior. We have moved this part to the appendix C now.
>
> We hope these clarifications address your concerns. Please let us know if any further details would be helpful.

---

> ### Author Response · Authors · 2025-12-03
>
> Thank you for your suggestion and for being willing to increase the score based on our recent improvements. We truly appreciate the time and effort you have invested in our paper, and we are very encouraged by your positive view of our work.
>
> Below we provide a point-by-point response to the remaining questions, discussions, and suggestions you raised in the latest round of comments:
>
> ***Regarding the Question about Policy:***
>
> Yes, this interpretation is correct. Our policy is implemented as a linear combination of simple rules, where each rule is a conjunction of a few threshold tests on input features. In that sense, NSAC can indeed be viewed as a linear ensemble of extremely shallow decision trees. We chose this rule-based formulation rather than tree based one because deeper trees quickly lose interpretability and are not easily modular: individual branches are hard to compare, reuse, or analyze in isolation. In contrast, our rules remain small, modular components that can be inspected and combined transparently, while the linear aggregation still gives enough flexibility to approximate a strong policy.
>
> ## Discussions ##
>
> ***Regarding Performance***
>
> Thank you very much for sharing this perspective and we fully agree with your stance. Our main goal with this work is not to chase SOTA scores in the training environments (match is good enough for us), but exactly what the reviewer highlight: to move towards agents that “select the right action for the right reasons” by incorporating structure, abstraction, and human-inductive biases, while keeping the policy transparent. NSAC is explicitly designed to explore this transparency–complexity trade-off rather than to push raw performance as far as possible with opaque deep models.
>
> ***Regarding the Pong Environment***
>
> Thank you very much for pointing this out — the point is correct. We inspected the learned rule ensemble and computed a simple global feature-importance measure by summing the absolute coefficients of all rules in which each feature appears. This analysis shows that the policy places disproportionately high weight on the enemy’s vertical position y_e​, alongside the ball coordinates, while largely ignoring its own paddle state. In other words, as the reviewer anticipated, the agent is indeed using the enemy position as a shortcut signal. We now include this rule set and the corresponding feature-importance scores in the appendix and explicitly discuss this as an example of transparent-but-incorrect reasoning.
> These additions support the reviewer’s observation: the problematic policies are not failing because of sparse rewards, but because they exploit an easy shortcut via the enemy position. The advantage of our method is that this shortcut becomes clearly interpretable in the learned rules and the feature-importance analysis.
>
> ## Suggestions ##
> ***Regarding the first Figure:***
> Thanks for this suggestion. We have redone the graph as recommended and highlighted the potential issues arising from relying on a black-box model.
>
> ***Regarding the Figure 4:***
> We have updated Figure 4 (now shown as the full-page Figure 13 in Appendix M). We also include an environment rendering at the decision-shift point where p = - 0.48 and v = 0; when the MountainCar tracks with the goal flag on the right hill and the car renders near the bottom of the valley at P = −0.48. The policy’s decision depends on the sign of the velocity: for v > 0, the agent accelerates to the right to build speed toward the goal, whereas, for v < 0, it accelerates left to commit to a full swing back and gains momentum for the next climb.
>
> ***Regarding the Relational Reasoning Env***
>
> Thank you very much for pointing out these environments — this was extremely helpful. We have conducted an additional experiment on OCAtari Breakout and included the results in appendix S. Below is the performance for this new environment:
> | Environment | Q-table          | DQN              | A2C               | PPO              | SDSAC            | SACBBF           | Rainbow          | SYMPOL        | πaffine-D     | D-SDT        | NSAC  (Ours)    |
> |-|-|-|-|-|-|-|-|-|-|-|-|
> | OCAtari-breakout    | 41.23   | 127.23    | 344.23    | 389.21  | 342 | 267   | 235.76  | 227      | 23.86      | 34.26  | 312.76|
>
> PPO achieves the highest overall score on OCAtari-Breakout. Nonetheless, NSAC also attains strong performance, with an average score above 300 over 5 random seeds, making it competitive with neural baselines while clearly outperforming all other symbolic methods.

---

> > ### Author Response · Authors · 2025-12-03
> >
> > ***Regarding the Figure Captions:***
> >
> > Thank you very much for this suggestion regarding the structure of figure and table captions. We fully agree that the suggestion will make the paper easier to read and better connect the visuals to our claims. Due to the strict page limit for the submission version, we were not able to expand all captions to the desired level of detail without exceeding the space constraints. We have adjusted the captions as suggested, shown below, and we will incorporate them into the main text once space allows. Thank you again for the helpful feedback.
> >
> > Figure 2 : \caption{**NSAC learns an interpretable actor by combining a neural critic with a rule-based policy updated via orthogonal gradient boosting.** The diagram illustrates the full NSAC training loop, including critic updates, rule mining, rule replacement, and fully-corrective weight optimization steps for the symbolic actor. Blocks and arrows show how trajectories, advantages, and gradients are propagated through the system to iteratively refine the rule ensemble; additional algorithmic details are provided in Section 3 and Appendix C.}
> >
> > Table 1: \caption{**Our NSAC policy attains rewards that match or exceed deep RL baselines while clearly outperforming symbolic baselines across most environments.** The table reports mean episodic return (± standard deviation) for NSAC, classic deep RL methods (e.g., A2C, PPO, Rainbow), and symbolic approaches (SYMPOL, $\pi_\text{affine}^D$, D-SDT) on the Gym and HVAC tasks, showing that NSAC achieves comparable performance to strong neural baselines and markedly higher returns than tree-based policies, especially in more challenging environments, such as HVAC. Best results are highlighted in bold, with all numbers averaged over multiple random seeds; further training and evaluation details are given in Appendix~B.}
> >
> > Figure 4 : \caption{**The learned MountainCar rules recover the classic momentum-building strategy in a compact and interpretable form.** The figure visualizes the actor outputs over the $(p,v)$ state space for each action, where higher heatmap values indicate that an action is more strongly preferred in that region. Overlaid squares mark the domains where individual symbolic rules apply, revealing how simple local conditions on position and velocity piece together into the global rocking behavior required to reach the goal; additional visualizations and discussion are provided in Appendix~M.}
> >
> > Figure 5 : \caption{**The HVAC rule ensemble encodes intuitive, human-readable control patterns that balance comfort and energy use.** This figure shows heatmaps of the actor output over key HVAC state variables, where higher values correspond to a stronger preference for each heating–cooling action. The resulting patterns align with domain intuition (e.g., increasing heating when indoor temperatures are low and outdoor conditions are cold), illustrating how NSAC produces a policy that is both effective and easy to interpret; further qualitative examples and time-series analyses are included in Appendix~M.}

---

### Author Response · Authors · 2025-11-27

Thank you again to all reviewers for the thoughtful feedback and engagement with our work. We have carefully revised the manuscript and prepared responses that we hope address your comments and clarify our contributions. If there are any remaining questions or if further clarification would be helpful, we would be very happy to continue the discussion during the remaining time in the rebuttal period

Authors

---

> ### Author Response · Authors · 2025-12-03
> **Final Remark**
>
> We sincerely thank all reviewers for their thoughtful feedback and the Area Chair for coordinating the process.
> Our initial scores were:  2 (VfDp), 4 (gu5z), 4 (oca3), and 6 (2xmA).
>
> Following our rebuttals to all reviews on 24 Nov 2025, two reviewers - gu5z and VfDp - engaged extensively with us, agreed with our clarifications, and indicated that they would increase their scores. The other two reviewers - oca3 and 2xmA - did not respond to our rebuttal before it closed.
>
>
> - **gu5z increased the score from 4 to 6, as shown in the discussion,**
>
>
> - **VfDp raised the score from 2 to 6, and explicitly stated an intent to raise it to 8, as shown in the discussion,**
>
>
> - **oca3 (4) and 2xmA (6) did not respond.**
>
>
> **Our score distribution after discussion is: 6(8), 6, 4, 6 with the last two who didn't respond.**
>
>
> Below, we provide a summary of this discussion process and describe how we addressed the remaining concerns from reviewers who were unable to engage further during the discussion period.
>
> ***Reviewer gu5z*** engaged in a constructive post-rebuttal discussion with us (before 24 Nov 2025), mainly focusing on the interpretability of other baselines. In this exchange, we compared our approach with these baselines from an interpretability perspective and incorporated this discussion into Appendix Q. Following this, the reviewer raised their score from 4 to 6. As they noted in the final sentence of their last comment:
> - ***“Thank you for your additional work on the manuscript… For those reasons, I am increasing my score to 6.”***
>
> Reviewer gu5z also pointed out an unintentional error that was introduced during the formatting conversion of the appendix. We have since corrected this issue in the revised manuscript.
>
> ***Reviewer VfDp*** provided highly constructive feedback that helped us significantly improve the quality of our paper. During the discussion phase, they noted:
>
> - ***"The paper structure and content has been greatly updated. I will thus update my score to 6 (I am considering moving up to 8).”***
>
> - The initial score is 2
> - Note: Reviewer VfDp misplaced his comments under Reviewer oca3
>
> They also clarified:
> - ***“Because of the incident, I cannot modify my score. I would like to openly say that I had made my mind about the updated work before the incident. I personally don't care about leak. I stand by my reviews. However, I would like to update my score such that it reflects the current state of the authors' work. I hope that we will be able to do this soon.”***
>
> In addition to their score-related comments, Reviewer VfDp asked several helpful questions and provided detailed editorial suggestions, which we have addressed as follows:
> - *Clarification on Rule Structure:* We confirmed the reviewer’s interpretation is correct — our rules can indeed be viewed as extremely shallow decision trees, where tree depth is the main bottleneck for interpretability.
> - *Additional Example in Appendix:* We added a new example in the appendix for the MountainCar environment, and we provided a rescaled version of Figure 4 to improve clarity. We will enlarge this figure further in the main paper.
> - *Figure Naming Improvements:* As suggested, we provided improved figure captions and naming in the discussion to be incorporated into the main text (space permitting).
> - *OCAtari Pong Environment:* Following the reviewer’s request, we incorporated both the symbolic rule set and corresponding analysis for the Pong environment (OCAtari version).
> - *Other Relational Reasoning Environment:* We conducted additional experiments on the OCAtari-Breakout environment, as suggested by the reviewer. Our method achieves an average score above 300 (slightly lower than PPO, but the best in the symbolic methods) while retaining interpretable, rule-based policies. The details of this experiment are provided in Appendix S. We will incorporate these results into the main text if space permits.

---

> > ### Author Response · Authors · 2025-12-03
> > **Final Remark**
> >
> > **Reviewer oca3 (initial score 4)**, who did not participate in the discussion before it closed, raised two concerns in the initial review. Below is a point to point summary of our rebuttal regarding those issues:
> > - Reviewer oca3 questioned about the interpretability of NSAC, arguing that because the actor’s behavior is influenced by the critic during training—and the critic remains a black box—it is unclear whether the final policy can be meaningfully explained.
> > In response, we clarified that NSAC’s explanations focus on the actor’s policy at deployment time, which is when interpretability is most relevant. While the critic does play a role during training—providing advantage estimates and learning signals—it is not involved at all during inference. The actor’s decisions depend solely on the observed state and the learned rule ensemble, which is fully transparent and interpretable. Therefore, ***the explanations we provide—based on rule activations and their contributions to each action—faithfully reflect the actor’s decision-making process, regardless of how the rules were shaped during training.***
> > - The reviewer also asks about more complex actors: Since our actor is rule-based, increasing its complexity essentially means either (1) introducing a more intricate rule structure, or (2) increasing the number of rules. While both extensions are technically feasible within our framework, we intentionally avoid them to preserve the interpretability of the model. Larger ensembles or deeper rules would undermine the transparency that is central to NSAC's design. Instead, we deliberately keep the rule ensembles compact and human-readable. Importantly, we have also successfully applied this approach to complex, real-world tasks, such as multi-zone HVAC control, where it has demonstrated strong performance despite the high task complexity—further validating the practicality of our minimalist design.
> >
> > **Reviewer 2xmA (initial score 6)**, who also did not participate in the discussion before it closed, was quite positive about our contribution, noting that ***“There is a lot to appreciate about this paper. It sets out with a very clear goal: extend Actor-Critic to retain its performance, while also ensuring interpretability. The premise to make use of the boosting-like additive rule ensembles is quite clever, and not something I have seen much of in deep reinforcement learning.”*** They also raised several remaining concerns, mainly about editorial issues and presentation. We appreciate the reviewer’s suggestions and have revised the manuscript as follows:
> > - *Interpretability discussion (simulatability, modularity, low complexity):* We added a dedicated paragraph discussing these three aspects in the main text and further extended the comparison between our method and tree-based approaches in Appendix Q. We have corrected this in the revision: when shortening the paper to meet the page limit, we inadvertently removed the relevant discussion from the submitted version.
> > - *Improved exposition and figures:* We introduced a new figure (Figure 2) to illustrate the algorithm more clearly and rephrased several result descriptions to improve clarity (these edits were also aligned with suggestions from Reviewer VfDp). We also corrected the mischaracterized title to more accurately reflect the scope of our work.
> > - *Clarification on “Q-learning-like” updates:* The concern about a Q-learning-style update stems from a notation clash: in our manuscript, q denotes queries in the rule system, not Q-values in the RL sense. The policy updates themselves follow a standard A2C-style policy update, not a Q-learning update. We have clarified this in the revised text to avoid further misunderstanding.
> >
> >
> > ***In summary,*** this work presents an actor–critic framework that combines a neural critic with a rule-ensemble actor to address key challenges in reinforcement learning. Neural networks provide scalability and adaptability on the critic side, while the rule ensemble ensures transparent, interpretable decision-making for the actor. We develop a theoretical foundation for integrating rule ensembles into the Advantage Actor–Critic (A2C) framework, and show across several classic and complex HVAC environments that our method matches or exceeds the performance of representative deep RL and symbolic RL baselines, while remaining intrinsically interpretable and transparent.
> >
> > We are encouraged that reviewers recognize our work’s novelty, theoretical soundness, practical evaluation, and relevance to the community. We are also confident that our responses have adequately addressed all of the reviewer’s remaining concerns. We believe this consensus on our technical contributions, together with the clarified scope of the remaining points, will provide helpful context for the Area Chair when assessing the updated reviews.

---

### Meta-Review · Area_Chair_CEZV · 2026-01-02

**Summary:**

The paper introduces a novel interpretable reinforcement learning (RL) approach within the actor–critic framework. The core idea is to retain neural networks for the critic to ensure accurate value estimation, while replacing the neural actor with an interpretable policy representation. For interpretability, the authors adopt rule ensembles, which learn to select and weight multiple symbolic Boolean rules for each action given the current state. The advantage of each action is computed as the weighted sum of its associated rules, and the action with the highest advantage is selected. In addition to defining the methodology, the paper provides a local convergence proof and empirical evaluations assessing both performance and interpretability.

The paper received 4 reviews which initially leaned towards reject (3 being negative and 1 being positive). The major points of contention from the reviewers were as follows:

* The question regarding interepretability was raised by a couple of reviewers as the critic that influences the actor and thus directly impacts interpretability, is a blackbox.

* Lack of complex baselines since interpretable RL has now shown effectiveness in environments such as Atari games and thus need to be included in the empirical evidence.

* Overall structuring of the paper was deemed a bit insufficient by some reviewers.

**Reviewer Concerns:**

I think the overall rebuttal provided by the authors was pretty strong and the authors did a great job in answering almost all concerns along with rincorporating the said changes in the manuscript. This was evident in the increase in score by 2 of the reviewers. Overall, the answers about interpretability and overall structuring of the work was well answered and implemented. Although I do not see the related work being improved as the citations for several recent interpretable RL methods still seem to be missing despite of the fact that the authors have mentioned in the rebuttal that they have added the same. I hope that this is taken care of in the final version of the paper.

**Reviewer Scores:**

2 reviewers had already revised their scores as evident from the discussion and based on the rebuttal I think the text is adequate for the non-responsive reviewers (1 of which was already leaning positive) to shift their score upwards.

I hope that the authors add the related work and incorporate other specific changes in the final version of the paper. I recomment acceptance.

---

### Decision · Program_Chairs · 2026-01-26

Accept (Poster)